# Natural variation of *TBR* confers plant zinc toxicity tolerance through root cell wall pectin methylesterification

Kaizhen Zhong[1,2,3], Peng Zhang [1], Xiangjin Wei [1], Matthieu Pierre Platre [2], Wenrong He[2], Ling Zhang[2], Anna Małolepszy[2], Min Cao[2], Shikai Hu[1], Shaoqing Tang[1], Baohai Li [2,4] ✉, Peisong Hu [1,3] ✉ & Wolfgang Busch [2] ✉

Zinc (Zn) is an essential micronutrient but can be cytotoxic when present in excess. Plants have evolved mechanisms to tolerate Zn toxicity. To identify genetic loci responsible for natural variation of plant tolerance to Zn toxicity, we conduct genome-wide association studies for root growth responses to high Zn and identify 21 significant associated loci. Among these loci, we identify *Trichome Birefringence* (*TBR*) allelic variation determining root growth variation in high Zn conditions. Natural alleles of *TBR* determine TBR transcript and protein levels which affect pectin methylesterification in root cell walls. Together with previously published data showing that pectin methylesterification increase goes along with decreased Zn binding to cell walls in *TBR* mutants, our findings lead to a model in which *TBR* allelic variation enables Zn tolerance through modulating root cell wall pectin methylesterification. The role of TBR in Zn tolerance is conserved across dicot and monocot plant species.

Zinc (Zn) is an essential micronutrient for plants, however, its excess causes cellular damage. Consequently, plants have evolved tight regulatory mechanisms to maintain Zn homeostasis. However, once Zn concentration exceeds a threshold, plants will display toxicity phenotypes[1]. Excess Zn in the soil results in the inhibition of plant growth, leaf chlorosis, and yield reduction[2]. Zn content in agricultural soils ranges from 10 to 300 mg kg$^{-1}$ in soils of different areas of the world[3]. A Zn content between 30 and 200 µg Zn g$^{-1}$ of plant dry weight (DW) is required for normal growth in the majority of crops[4]. However, Zn content in surface soils can be contaminated with excessive Zn (up to 3000 mg kg$^{-1}$ Zn dry soil) due to human activities[5] and subsequently can easily build up in excess in plants (> 400 µg Zn g$^{-1}$ DW) to cause Zn toxicity[6]. Zn toxicity is prevalent in acidic and sludge amended soils, and there potentially limits crop production and quality.

Plants have developed tolerance mechanisms to keep Zn homeostasis when exposed to excess concentration of Zn in the environment. These include intracellular binding to Zn chelators[7], sequestration into the vacuole[8], efflux from root cells[9], reduction of transport to plastids[10], and cross-homeostasis between Iron (Fe) and Zn[11]. Some genes such as *HMA4* and *MTP1*, which are involved in Zn tolerance, showed higher gene expression in Zn hyperaccumulators *A. halleri* and *N. caerulescens* compared to *Arabidopsis thaliana*[12]. Root cell walls, the first interface between soil metals and plant cells, are another potential location for metal accumulation[13]. The modification of root cell wall components enhances the ability of the apoplast to accumulate excess metals, and that can contribute to preventing metals from entering the plant and thereby improve plant tolerance to metal toxicity in the soil[1,13]. The impact of such cell wall modifications is apparent in *ozs2* mutants that are defective in the gene encoding pectin methylesterase 3 (*PME3*) and that showed Zn hypersensitivity[14]. *dez* (*tbr; trichome birefringence*) mutants that show excessively methylesterified pectin display Zn hypersensitivity in the light[15] *and*

[1]State Key Laboratory of Rice Biology and Breeding, China National Rice Research Institute, Hangzhou, Zhejiang, China. [2]Plant Molecular and Cellular Biology Laboratory, Salk Institute for Biological Studies, La Jolla, CA, USA. [3]School of Agricultural Sciences, Jiangxi Agricultural University, Nanchang, Jiangxi, China. [4]MOE Key Laboratory of Environment Remediation and Ecological Health, College of Environmental and Resource Sciences, Zhejiang University, Hangzhou, Zhejiang, China. ✉e-mail: bhli@zju.edu.cn; hupeisong@caas.cn; wbusch@salk.edu

*TBR* has been previously identified to be involved in basal metal tolerance, which the metal tolerance level is thought to be common to all plants including *A. thaliana* as opposed to metal hypertolerance which is found only in specialist plants that are found on soils containing high, toxic levels of metals[16,17]. While the potential role of root cell wall composition on Zn binding and accumulation to avoid Zn toxicity in plants is evident, little is known about the genetic and molecular mechanisms that modulate root cell wall composition to avoid Zn toxicity.

Crops differ remarkably in their susceptibility to Zn toxicity. In acidic soils, most dicots are much more sensitive than most grasses. Leafy vegetables and the beet family are more sensitive to Zn toxicity than most other dicots[5]. Natural variation of plant tolerance to Zn toxicity is also widely observed at the intraspecies level, including *Arabidopsis*[11,18], rice[19–23], and *Brassica rapa*[24]. A total of 13 quantitative trait loci (QTL) contributing to Zn toxicity tolerance were identified via linkage mapping using biparental crosses[19,20] and 86 significantly associated loci were detected by genome-wide associate study (GWAS) using three genetically diverse populations[21–23] in rice, respectively. While genes that encode proteins such as MTP1, MTP3, ZIF1, HMA2, HMA3, HMA4, and PCR2 have central roles in basal Zn tolerance of *A. thaliana*[1], natural alleles underlying natural variation of Zn tolerance within plant species remain largely unknown. So far, *Ferric Reductase Defective 3* (*FRD3*) has been identified to be the underlying natural variation of the root growth responses to Zn toxicity tolerance in *Arabidopsis thaliana*, by regulating Fe and Zn translocation from the root to the shoot[11].

Here, making use of natural variation of root growth responses to high Zn in *Arabidopsis*, we map numerous genetic loci by GWAS. We identify *TBR* allelic variants as important determinants of root growth variation in high Zn conditions. We go on to characterize *TBR* allelic variation in this context and show that allelic variation of *TBR* determines transcript and protein expression levels of *TBR*, as well as root cell wall pectin methylesterification. As a *tbr* mutant had been previously identified to be hypersensitive to excess Zn and Zn-enhanced photomorphogenesis displayed altered pectin methylation and acylation affecting photomorphogenesis and going along with a decreased capacity to bind Zn in cell walls[15], our data give rise to a model in which *TBR* allelic variation enables Zn tolerance through pectin modification in root cell walls. Finally, we show that the role of TBR in Zn tolerance is conserved across dicot and monocot plant species.

## Results

### Natural variation of *Arabidopsis* root growth responses to Zn toxicity associates with several loci including *FRD3*

GWAS comprises an efficient method to identify genes and alleles associated with complex traits in diverse plant species[25–30]. Primary root growth is frequently used to evaluate metal tolerance[31–33] and has been used to assess Zn toxicity tolerance in plants[11,18]. To identify genes and their variants that underlie the natural variation of root growth responses to high Zn levels, we therefore cataloged root growth of *Arabidopsis thaliana* seedlings grown in control (½ MS) and high Zn conditions (300 μM) from day 1 to day 8 after germination. We screened 317 diverse accessions from the 1001 genome collection (Supplementary Data 1)[34] and observed extensive natural variation among accessions in root growth responses to high Zn (Supplementary Fig. 1 and Supplementary Data 2). The broad sense heritability of root length was from 36% to 60% in control conditions, and from 47% to 70% in high Zn conditions (Supplementary Table 1). Applying a mixed-model based GWAS approach that corrects for population structure for root length in Zn tolerance, we detected 21 loci exceeding the Benjamini Hochberg threshold of 5% from day 1 to day 8 (Supplementary Fig. 2 and Supplementary Data 3). Among the significantly associated loci, one was located within the known Zn tolerance gene

*FRD3*[11] (Fig. 1a, b). Fifty-one significantly associated SNPs were located throughout the upstream, coding, and downstream regions of *FRD3* (Supplementary Data 4). Consistent with *FRD3* being the causal gene in this region, it showed significantly different expression in high Zn conditions between accessions containing either the C-variant or T-variant at the top SNP (Chr3:2574560) (Fig. 1c and Supplementary Fig. 3). In contrast, the expression of the neighboring genes *AT3G08490* and *AT3G08500* was not significantly different between accessions with differing SNP variants (Fig. 1c). These results, as well as the known role of *FRD3* in Zn tolerance, strongly suggested that *FRD3* might be the causal gene in this region. Finally, we tested whether *FRD3* affects root growth in our high Zn conditions. For this, we measured the primary root length of *frd3* mutants in a high Zn medium. Compared to the Col-0 wildtype, the *frd3-3* mutant showed a significant reduction of root growth in response to high Zn according to the 2-way ANOVA ($p < 0.001$) while the genotype x treatment interaction was marginally significant for *frd3-7* ($p = 0.063$) (Fig. 1d, e, Supplementary Fig. 4 and Supplementary Table 2 and 3). Overall, these data, and the data published by Pineau et al. [11], indicate that our high Zn GWAS screen was able to identify genes that are involved in determining root growth responses to high Zn conditions. Based on this strong finding, we then set out to identify other potential genes involved in determining root growth response to Zn toxicity.

### Naturally occurring *TBR* alleles are associated with root growth variation in high Zn conditions

We further focused our attention on a significant peak located on chromosome 5 (Fig. 2a), which persistently was detected at all time points starting from day 4 (Supplementary Fig. 2). According to the top SNP (Chromosome 5; position 2070536), all the accessions were grouped into two groups. The T-allelic group was associated with Zn sensitive root growth and contained 19% of the accessions, while the A-allelic group was associated with Zn tolerant root growth and contained 81% of the accessions, including Col-0 in our GWAS panel (Supplementary Fig. 5a).

To identify the causal gene for Zn tolerance within the peak on chromosome 5, we first determined genes that were in proximity to the associated peak. We detected 2 genes, *AT5G06700* (*TBR*) and *AT5G06710* in the genome region covering a window of 10 kb around the top SNP (Fig. 2b). We first analyzed their gene expression. Publicly available organ and tissue-specific microarray expression data (https://bar.utoronto.ca/efp/cgi-bin/efpWeb.cgi)[35] showed that *TBR*, as well as *AT5G06710,* are expressed throughout the plant (Supplementary Fig. 5b–e). Publicly available root single-cell RNA sequencing data[36] showed that *TBR* is expressed in most root cell types, while *AT5G06710* is expressed in a subset of root cell types only (Supplementary Fig. 5f). To then test whether the response of the candidate genes in the allelic groups of accessions that were identified by GWAS was consistent with the causal involvement of one of the two candidate genes, we selected representative accessions from the T-allelic group (Eds-1, Eds-9 and TRÄ 01) and the A-allelic group (Vår2-6, Sim-1 and Fri 1). In addition, we also included Col-0 with an intermediate root length, which was from the A-allelic group. We found that the transcript level of *TBR* in Vår2-6, Sim-1, and Fri 1 that are from the A-allelic accessions under high Zn conditions was slightly but significantly higher than that of the T-allelic accessions (Col-0 which was the A-allelic group but didn't display a significantly higher expression level of *TBR* compared to the T-allelic accessions) (Fig. 2c and Supplementary Fig. 5g). In contrast, *AT5G06710* showed no significant difference between the two allelic groups of accessions both in control and high Zn conditions (Fig. 2c and Supplementary Fig. 5g). While the detected differences were at the lower end of changes that can be reliably detected via RT-qPCR, nevertheless, this analysis hinted towards *TBR* to be the best candidate gene within the proximity of the GWAS association for being causal for root growth variation in high Zn. Moreover, *TBR* was also in linkage

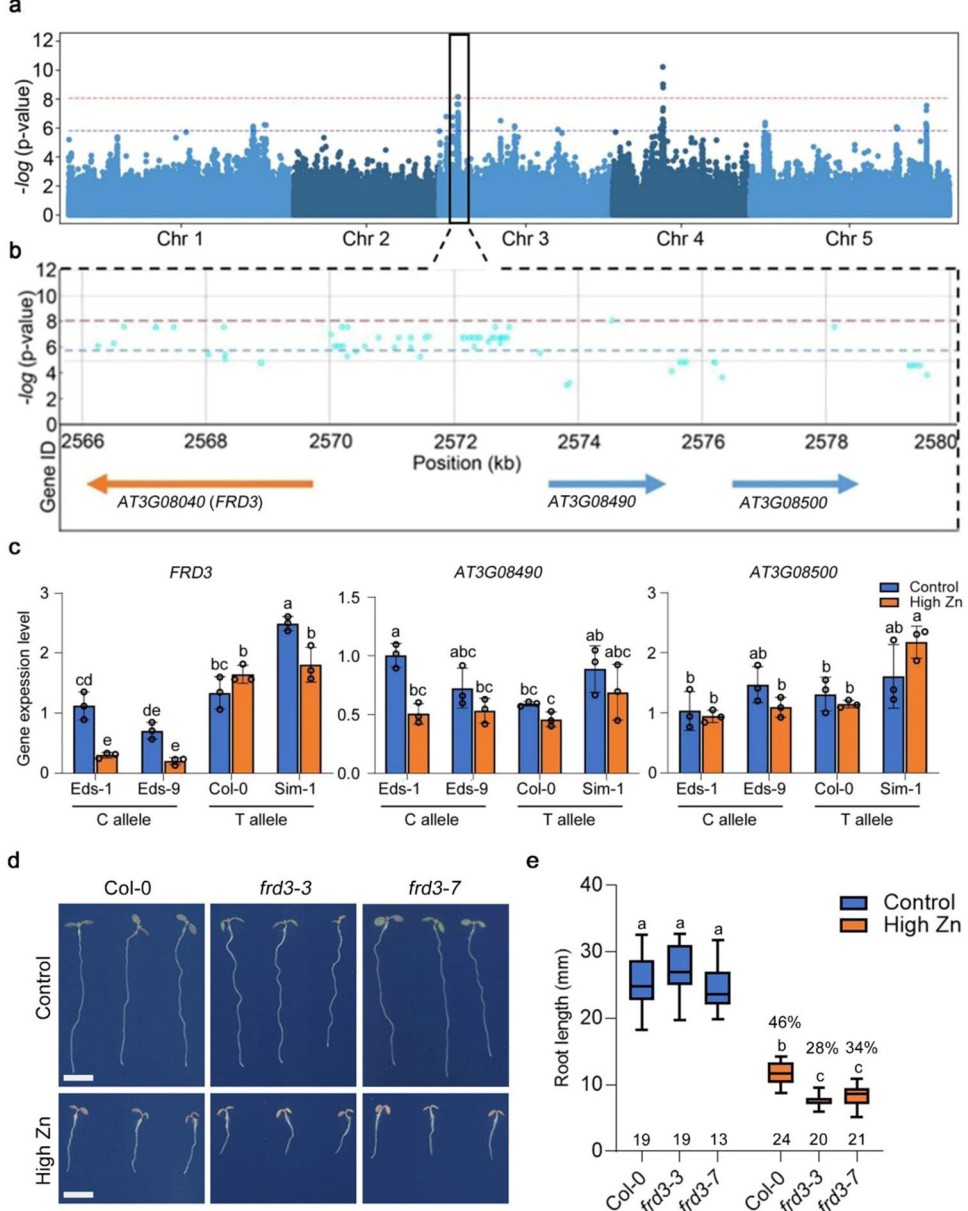

**Fig. 1 | Natural allelic variation of *FRD3* is associated with root growth responses to high Zn levels. a, b** Genome-wide Manhattan plot (**a**) and regional plot (**b**) of GWAS for the relative primary root length in high Zn conditions (root length high Zn / root length control) at 7 days after germination (DAG). *x*-axis: SNP position. *y*-axis: -log₁₀ *p*-value of association of the SNPs according to AMM mixed model analysis. Chromosomes are depicted in different colors. The horizontal blue dashed line corresponds to a 5% significance threshold after the Benjamini Hoch-berg correction and the red dashed line corresponds to a 5% significance threshold after Bonferroni correction. The black box in (**a**) indicates the significant GWA peak on Chromosome 3. Regional plot in (**b**) of the significant SNPs and candidate genes surrounding the significant peak on Chromosome 3. **c** Gene expression levels of *FRD3*, *AT3G08490*, and *AT3G08500* in roots of accessions with contrasting root responses to high Zn (Sensitive accession: Eds-1 and Eds-9; Tolerant accession: Col-

0 and Sim-1). Expression levels were normalized to the expression of Eds-1 in control conditions. Data are mean ± standard deviation (S.D.). Circles indicate a single biological replication, *n* = 3. Statistical analysis was performed using one-way ANOVA analysis with Tukey's HSD test (*p* < 0.05). **d** Representative images of *frd3* mutants (*frd3-3* and *frd3-7*) and their wildtype Col-0 seedlings grown on control and high Zn (300 μM) medium at 7 DAG. Scale bars: 5 mm. **e** Boxplots for root length of *frd3* mutants and their wildtype depicted in (**d**). The value on the top of each box in high Zn is the relative root length (root length in high Zn / root length in control). The number below each box indicates the number of replicates. Statistical analysis was performed using one-way ANOVA analysis with Tukey's HSD test (*p* < 0.05). For box plots, the horizontal line represents the median value, the lower and upper quartiles represent the 25th and 75th percentile, and the whiskers show the maximum and minimum values. Source data are provided as a Source Data file.

disequilibrium (LD) with the lead SNP of the GWAS peak (Supplementary Fig. 5h), and *TBR* had been shown to be involved in basal tolerance to excess Zn and repression of photomorphogenesis, as partial TBR loss-of-function caused decreased root growth under high Zn exposure and Zn-enhanced photomorphogenesis[15]. However, as the previous study addressed artificially induced mutations and not naturally occurring sequence variants, we still considered both genes

in proximity to the significantly associated SNP. We therefore obtained mutants for the two candidate genes (Supplementary Fig. 6). For TBR, we obtained two mutants: the *tbr-1* mutant in which TBR function is strongly impaired due to a G to A mutation in the 3ʳᵈ exon of the coding region, resulting a predicted amino-acid change of Gly to Glu at position 427[37], as well as the T-DNA line SALK_058509C, which hereafter was called *tbr-3* and which is most likely a loss of function allele as it

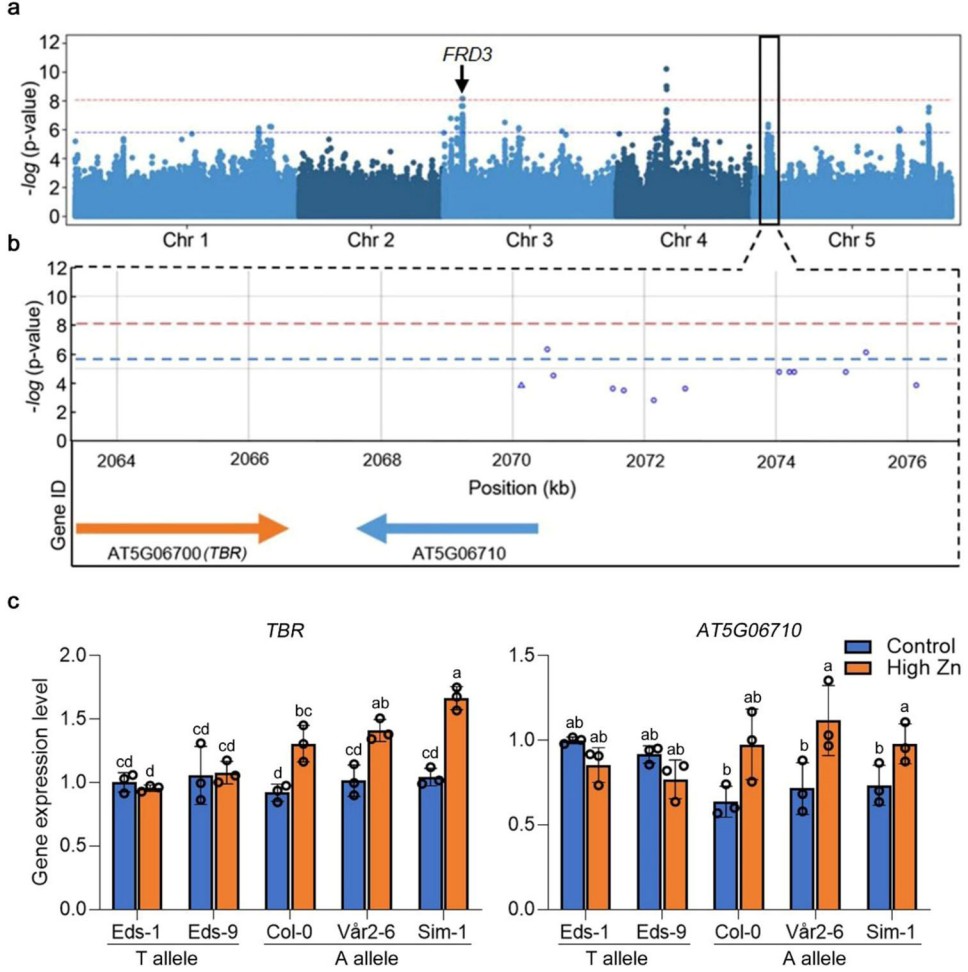

**Fig. 2 | Natural allelic variation of *TBR* is associated with root growth responses to high Zn levels. a, b** Genome-wide Manhattan plot (**a**) and regional plot (**b**) of GWAS for the relative primary root length in high Zn conditions (root length high Zn / root length control) at 7 DAG. *x*-axis: SNP position. *y*-axis: -log$_{10}$ *p*-value of association of the SNPs according to AMM mixed model analysis. Chromosomes are depicted in different colors. The horizontal blue dash line corresponds to a 5% significance threshold after Benjamini Hochberg correction and the red dash line corresponds to 5% significance threshold after Bonferroni correction. The black box in (**a**) indicates the significant GWA peak on Chromosome 5. **b** Regional plot of the significant SNPs and candidate genes surrounding the significant GWA peak on chromosome 5. **c** Gene expression levels of *AT5G06700* (*TBR*) and *AT5G06710* in roots of representative accessions (Sensitive: Eds-1 and Eds-9; Intermediate: Col-0; Tolerant: Vår2-6 and Sim-1). Expression levels were normalized to the expression of Eds-1 in control conditions. Data are mean ± S.D. Circles indicate a single biological replication, *n* = 3. Statistical analysis was performed using one-way ANOVA analysis with Tukey's HSD test (*p* < 0.05). For qRT-PCR analysis, the seedlings were grown directly on 1/2 MS medium for 5 DAG and then transferred to control and high Zn (300 μM) liquid medium for 12 h. Source data are provided as a Source Data file.

contains a T-DNA insertion in the coding sequence close to the 5'-UTR that most likely leads to no protein, or a truncated, or a non-functional protein (Supplementary Fig. 6a). For AT5G06710, we obtained the T-DNA insertion line SALKseq_062866 that has an insertion in the 3$^{rd}$ exon and therefore also most likely constitutes a loss of function mutant (Supplementary Fig. 6a). In addition to confirmation of the T-DNA insertion by genotyping, we also measured transcript levels at 3' downstream of the T-DNA insertions, which frequently led to alterations in the level of transcript. We found that in the cases of both T-DNA insertion mutants, the transcript was present at a much lower level than in wildtype (Supplementary Fig. 6b). We then determined root growth of these lines in ½ MS medium and high Zn conditions. The root growth of both *tbr* mutants was significantly more sensitive to high Zn. The *tbr-1* and *tbr-3* mutants displayed 75% and 58% root growth inhibition compared to control conditions, while Col-0 showed 36% inhibition when grown on a high Zn medium for 7 days (Fig. 3a, b). As we detected no obvious difference between wildtype and *tbr* mutants in leaves at high Zn soil (Supplementary Fig. 7), it appears that the sensitive phenotype of *tbr* mutants is root-specific. In contrast, the root growth of the *at5g06710* mutant in which expression of

*AT5G06710* was severely reduced (Supplementary Fig. 6b), was not significantly reduced compared to Col-0 in high Zn medium (Fig. 3a, b). Taken together, these data show that *TBR* is expressed in the root and required for root growth tolerance to high Zn as shown previously[15].

We next looked at the consequences of the *tbr* mutation on root growth at the cellular level. For this and the subsequent analyses, we utilized the *tbr-3* loss of function mutant, as the *tbr-1* mutation has a negative effect on *TBR* function, but it is not clear whether it is a simple loss of function mutant or a gain-of-function allele with a negative function. We acquired confocal microscopy images of the *tbr-3* mutant and Col-0 wildtype in control and high Zn conditions. While the length of the meristem zone and the mature cortical cell was not significantly reduced in *tbr-3* compared to wildtype under high Zn conditions (Supplementary Fig. 8), we found that *tbr-3* seedlings showed significant inhibition of the root elongation zone compared to wildtype under high Zn conditions (Fig. 3c, d). We next tested whether these phenotypes could be rescued by transgenic complementation of *TBR* in the *tbr-3* mutant (TBR_Col-0, the Col-0 *TBR* coding sequence driven by its own promoter from Col-0). Seedlings of four independent T3,

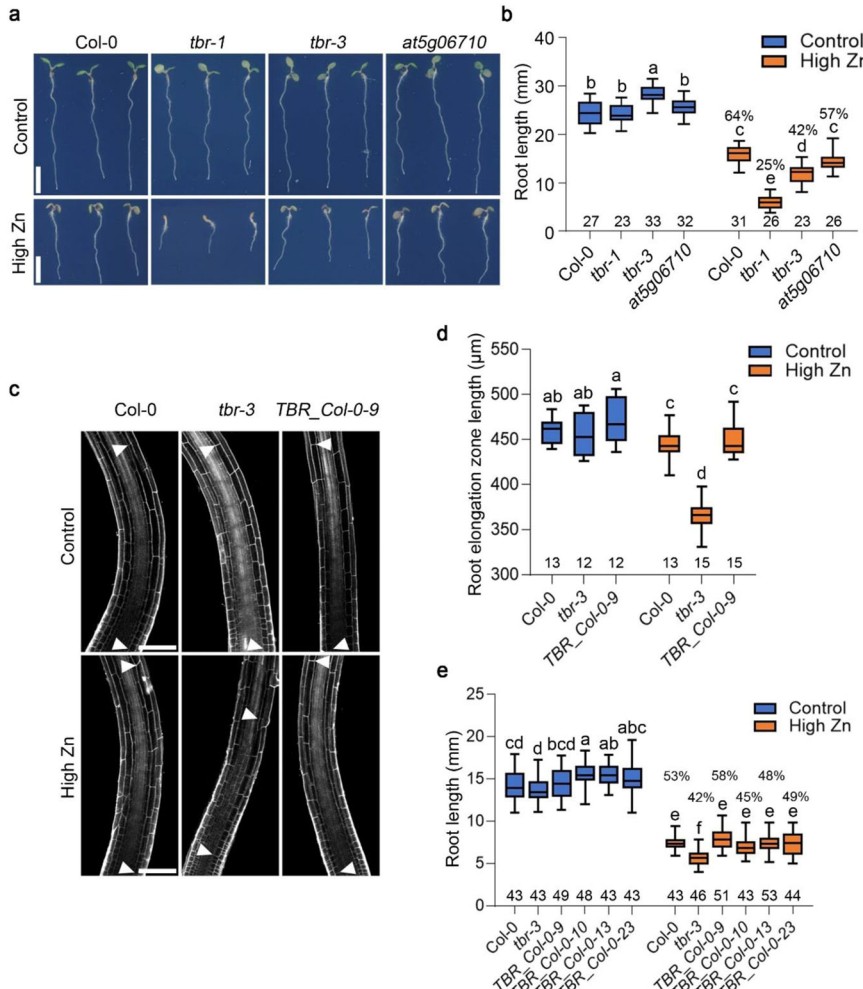

**Fig. 3 | *TBR* is required for root growth tolerance to high Zn levels. a** Pictures of seedlings of Col-0, *tbr* mutants, and the *at5g06710* mutant grown on control and high Zn (300 μM) medium at 7 DAG. Scale bars: 5 mm. **b** Boxplots for root length depicted in (**a**). The value on the top of each box in high Zn is the relative root length (root length in high Zn / root length in control). The number below each box indicates the number of replicates. Statistical analysis was performed using one-way ANOVA analysis with Tukey's HSD test ($p < 0.05$). **c** Confocal images of the root elongation zone of wildtype, *tbr-3* and *TBR_Col-0-9* complemented line at 4 DAG. The arrows indicate the border of the maturation zone and meristem zone. Scale bars: 100 μm. **d** Boxplots for root elongation zone length are depicted in (**c**). The number below each box indicates the number of replicates. Two times were repeated independently with similar results. Statistical analysis was performed using one-way ANOVA analysis with Tukey's HSD test ($p < 0.05$). **e** Boxplots for root length of *TBR_Col-0* T3 complemented lines at 4 DAG. The value on the top of each box in high Zn is the relative root length (root length in high Zn / root length in control). The number below each box indicates the number of replicates. Statistical analysis was performed using one-way ANOVA analysis with Tukey's HSD test ($p < 0.05$). For box plots, the horizontal line represents the median value, the lower and upper quartiles represent the 25th and 75th percentile, and the whiskers show the maximum and minimum values. Source data are provided as a Source Data file.

single insertion homozygous complementation lines showed the recovery of the root growth (Fig. 3e). We then measured the length of the root elongation zone in high Zn conditions in one of these lines and found that consistent with the restoration of root growth under high Zn, the length of the root elongation zone was also restored (Fig. 3c, d). Taken together, these results indicate that the root growth inhibition of *tbr* mutants in high Zn conditions is tied to a decreased size of the root elongation zone.

### *TBR* allelic variants cause root growth variation in response to high Zn via increased TBR expression level

To test whether allelic variation of *TBR* is causal for a portion of the observed variation in high Zn tolerance, we complemented the *tbr-3* mutant with alleles from two representative accessions. For this, we first fused the *TBR* promoter and 5′-UTR (2437 bp fragment upstream of the *TBR* translational start codon) to the *TBR* protein coding sequence (translation start codon to stop codon) from the T-allelic group accession (Eds-1) and the A-allelic group accession (Vår2-6) and

then transformed these constructs into the *tbr*-3 mutant. Consistent with the hypothesis that *TBR* variants determine root growth in high Zn conditions, five independent-homozygous lines (*TBR_Vår2-6*) from the A-allelic variant that was associated with increased root length fully rescued the *tbr-3* mutant phenotype in high Zn conditions and displayed longer roots in high Zn conditions than the T-allelic complementation lines (Fig. 4a). One of the T3 lines (*TBR_Eds-1-11*) that had been transformed with the T-allelic variant showed a significantly lower Zn tolerance than the *tbr-3* mutant, while the other two lines (*TBR_Eds-1-16* and *TBR_Eds-1-20*) showed increased root length compared to the *tbr-3* mutant (Fig. 4a). Consistent with the root growth phenotype, *TBR_Vår2-6* transgenic lines showed a significantly higher *TBR* transcript level than *TBR_Eds-1* transformed lines which was already apparent in control conditions (Fig. 4b). To determine whether *TBR* transcript regulation is associated with higher protein levels, we evaluated the TBR protein levels using confocal microscopy and western blot analysis in the *TBR* allelic complemented lines, where TBR was fused with mCITRINE and HA. Consistent with the transcript levels, we

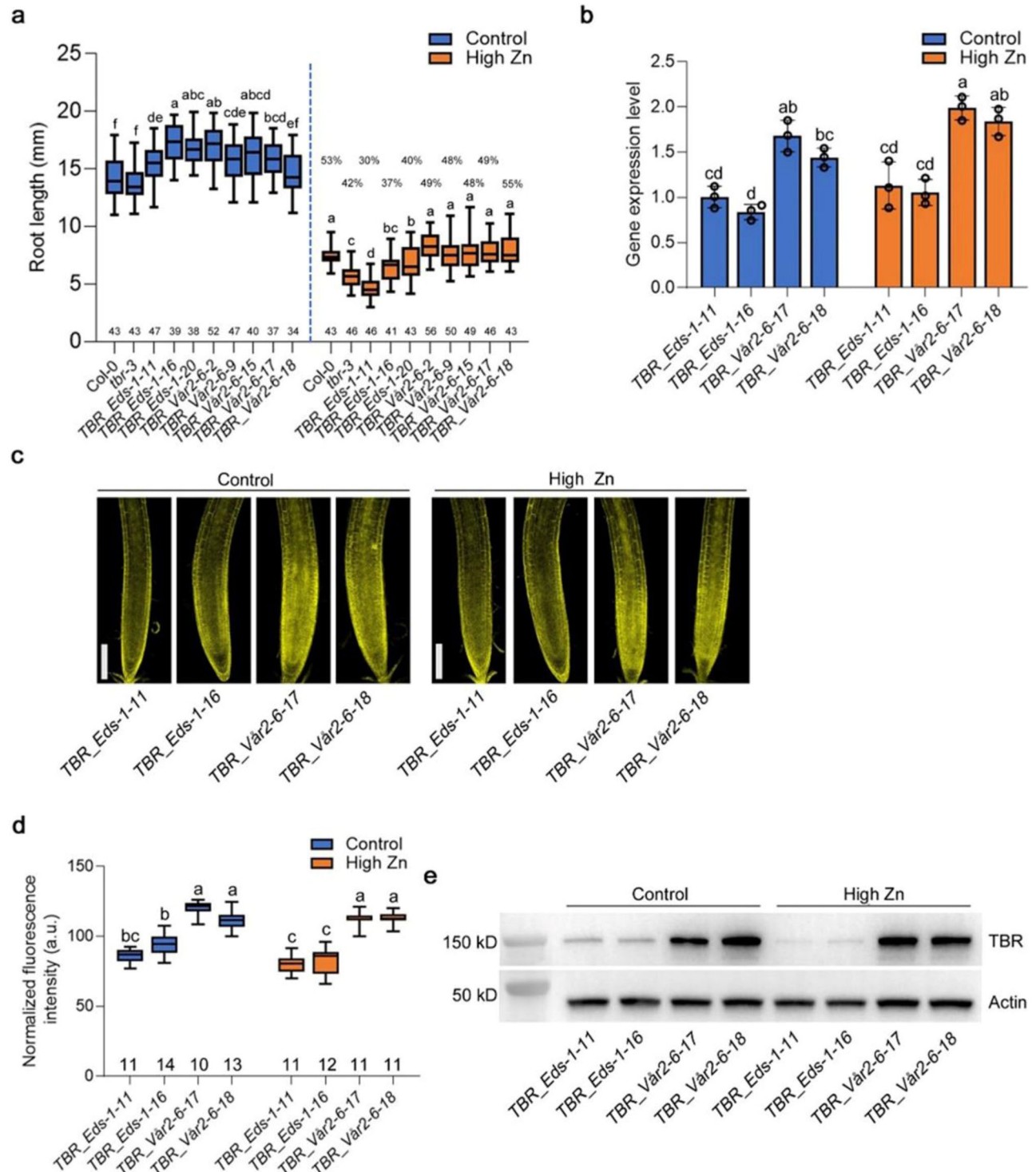

**Fig. 4 | Natural *TBR* variants cause variation of Zn toxicity tolerance. a** Boxplots for root length of T3 transgenic lines expressing *pTBR_Eds-1::TBR_Eds-1-HA-mCI-TRINE-HA* (*TBR_Eds-1*) and *pTBR_Vår2-6::TBR_Vår2-6-HA-mCITRINE-HA* (*TBR_Vår2-6*) grown on control and high Zn (300 μM) medium at 4 DAG. The value on the top of each box in high Zn is the relative root length (root length in high Zn / root length in control). The number below each box indicates the number of replicates. Statistical analysis was performed using one-way ANOVA analysis with Tukey's HSD test ($p < 0.05$). **b** *TBR* transcript expression in the roots of these transgenic lines. Expression levels were normalized to the expression of *TBR_Eds-1-11* in control conditions. Data are mean ± S.D. Circles indicate a single biological replication, $n = 3$. Statistical analysis was performed using one-way ANOVA analysis with Tukey's HSD test ($p < 0.05$). **c** Confocal images of TBR expression in the root tip at 6 DAG. mCITRINE: yellow signal. Scale bars: 100 μm. **d** Boxplots for fluorescence

intensity of T3 transgenic lines depicted in (**c**). a.u.: arbitrary units. The number below each box indicates the number of replicates. Two times were repeated independently with similar results. Statistical analysis was performed using one-way ANOVA analysis with Tukey's HSD test ($p < 0.05$). **e** Western blot analysis of TBR protein levels. Western blot of HA-tagged TBR was conducted using an anti-HA antibody. Actin was the protein loading control. Two times were repeated independently with similar results. For transcript expression and western blot analysis, the seeds were grown directly on 1/2 MS medium for 5 DAG and then transferred to control and high Zn (300 μM) liquid medium for 12 h. For box plots, the horizontal line represents the median value, the lower and upper quartiles represent the 25th and 75th percentile, and the whiskers show the maximum and minimum values. Source data are provided as a Source Data file.

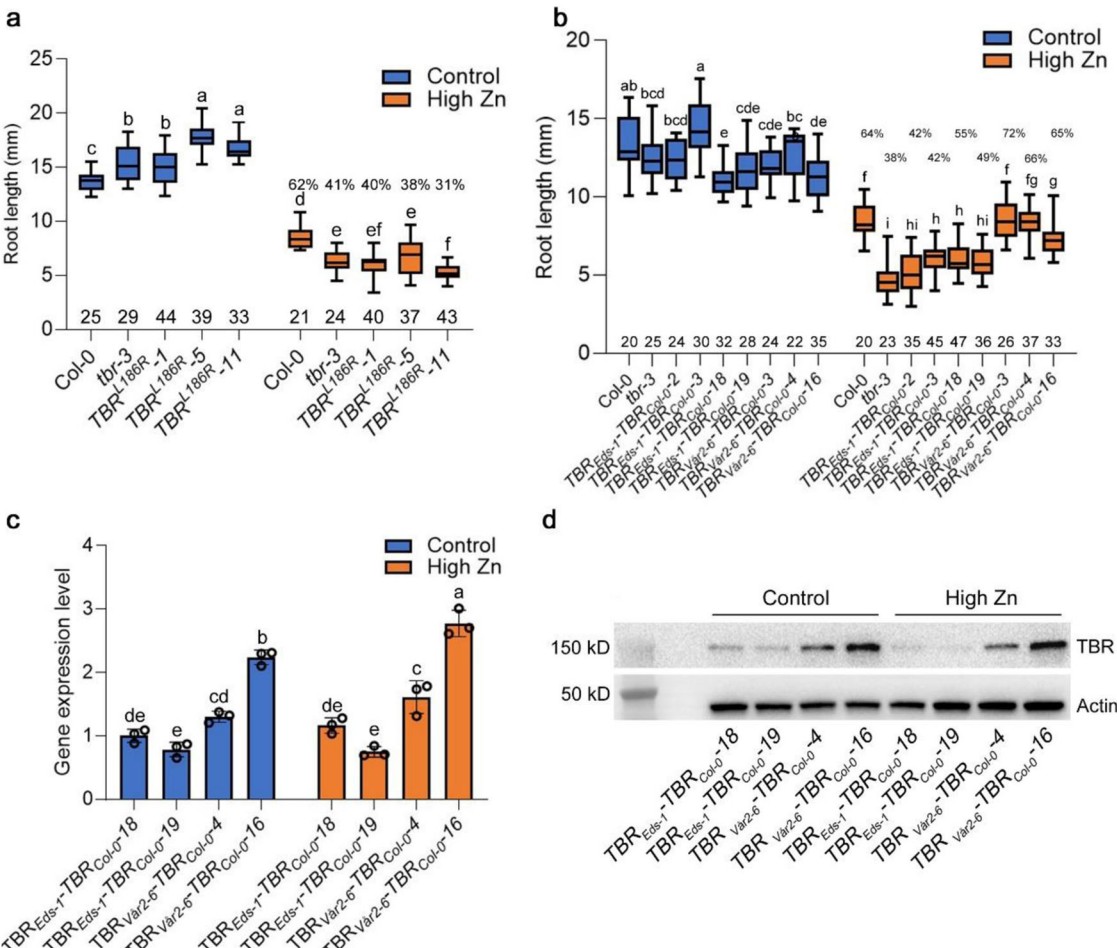

**Fig. 5 | Natural *TBR* variants underlie variation of Zn toxicity tolerance through sequence variants in the *TBR* promoter and/or 5′-UTR. a** Boxplots for root length of T3 transgenic lines from *pTBR::TBR^L186R^-HA-mCITRINE-HA* (*TBR^L186R^*) in control and high Zn (300 μM) conditions at 4 DAG. The value on the top of each box in high Zn is the relative root length (root length in high Zn / root length in control). The number below each box indicates the number of replicates. Statistical analysis was performed using one-way ANOVA analysis with Tukey's HSD test (*p* < 0.05). **b** Boxplots for root length of T3 transgenic lines from *pTBR_Eds-1::TBR_Col-O-HA-mCITRINE-HA* (*TBR_Eds-1-TBR_Col-O*) and *pTBR_Vår2-6::TBR_Col-O-HA-mCITRINE-HA* (*TBR_Vår2-6-TBR_Col-O*) at 4 DAG. The value on the top of each box in high Zn is the relative root length (root length in high Zn / root length in control). The number below each box indicates the number of replicates. Statistical analysis was performed using one-

way ANOVA analysis with Tukey's HSD test (*p* < 0.05). **c** *TBR* transcript expression in roots of *TBR_Eds-1-TBR_Col-O* and *TBR_Vår2-6-TBR_Col-O*. Expression levels were normalized to the expression of *TBR_Eds-1-TBR_Col-O-18* in control conditions. Data are mean±S.D. Circles indicate a single biological replication, *n* = 3. Statistical analysis was performed using one-way ANOVA analysis with Tukey's HSD test (*p* < 0.05). **d** Western blot analysis of TBR protein levels. Actin is the protein loading control. Western blot of HA-tagged TBR was conducted using an anti-HA antibody. Two times were repeated independently with similar results. For box plots, the horizontal line represents the median value, the lower and upper quartiles represent the 25th and 75th percentile, and the whiskers show the maximum and minimum values. Source data are provided as a Source Data file.

found that *TBR_Vår2-6* transgenic lines showed a higher accumulation of TBR protein than *TBR_Eds-1* transformed lines on both control and high Zn conditions (Fig. 4c–e and Supplementary Table 4). Taken together, these results indicate that *TBR* allelic variation significantly contributes to the natural variation of Zn toxicity tolerance via increased mRNA and protein expression levels of *TBR*.

## Allelic variation in the promotor and/or 5′-UTR of *TBR* modulates root growth responses to high Zn

After finding that *TBR* allelic variation was involved in the natural variation of root growth responses to Zn toxicity, we set out to identify the causal sequence polymorphisms. We, therefore, compared the polymorphism patterns of the two allelic groups that we had identified. When analyzing the sequence at the *TBR* locus, we found a SNP that caused non-synonymous amino-acid change among several members of the A-allelic group at position 2,070,324 (Supplementary Fig. 9, 10). While according to the whole genome resequencing data, it was not present in all A-allele accessions, due to potential whole-genome

sequencing inaccuracies and potential allelic heterogeneity, we considered the possibility that it contributed to the observed root phenotype variation. We, therefore, used site-directed mutagenesis to introduce the Arginine substitution at L186 in the Col-0 version of the *pTBR::TBR^L186R^-HA-mcitrine-HA* (*TBR^L186R^*) construct and, in this way generate the amino acid variant that was observed in the Zn tolerant A-allelic group. While the transformation of *tbr-3* with the Col-0 construct rescued root growth in high Zn conditions, root growth of *TBR^L186R^* lines was not rescued (Fig. 5a). This result suggests that the mutation of L186 observed in the natural alleles does not determine the observed natural variation. We then analyzed the promoter region of *TBR*, which contained several sequence variants (Supplementary Fig. 9). To corroborate these SNPs, we sequenced the sequence starting 2437 bp upstream of the translational start codon of the *TBR* gene in 3 T-allelic accessions and 3 A-allelic accessions. We found 7 SNPs in the promoter region and one SNP in the 5′-UTR that were different between T and A allelic accessions (Supplementary Fig. 11). One SNP was located in a putative transcription factor binding site for *AT1G14687*(*HB32*)

(Supplementary Table 5). To determine whether the allelic variation regulating root growth in Zn toxicity tolerance is attributable to the polymorphisms in the promoter, we fused the *TBR* promoter and 5'-UTR from T-allelic accession (Eds-1) and A-allelic accession (Vår2-6) to the *TBR* coding region of Col-0, and then transformed these constructs into the *tbr-3* mutant. The root length of all three independent $TBR_{Vår2-6}$-$TBR_{Col-O}$ variant lines fully recovered the *tbr-3* phenotype and was more tolerant to high Zn than the respective $TBR_{Eds-1}$-$TBR_{Col-O}$ variant lines (Fig. 5b). When we quantified transcript level via qRT-PCR, we found that the $TBR_{Vår2-6}$-$TBR_{Col-O}$-16 transgenic line showed 2.5 and 2.9-fold increased transcript level compared to $TBR_{Eds-1}$-$TBR_{Col-O}$ transformed lines in both control and high Zn conditions, while the other $TBR_{Vår2-6}$-$TBR_{Col-O}$ line ($TBR_{Vår2-6}$-$TBR_{Col-O}$-4) displayed about 1.5 and 1.7-fold increased transcript level compared to $TBR_{Eds-1}$-$TBR_{Col-O}$ transformed lines in high Zn conditions (Fig. 5c). Assessments of the protein level via western blotting supported an increased accumulation of TBR protein in both $TBR_{Vår2-6}$-$TBR_{Col-O}$ lines compared to $TBR_{Eds-1}$-$TBR_{Col-O}$ lines (Fig. 5d and Supplementary Table 6). Taken together, our results suggest that the sequence variation in the promoter region and/or 5'-UTR rather than the non-synonymous SNP in the coding region, causes the observed natural variation of root growth in high Zn conditions.

### *TBR* allelic variants influence pectin methylesterification in root cell walls to alleviate Zn toxicity

Since *TBR* is causal for the variation of root growth responses to Zn toxicity, we set out to investigate the underlying molecular mechanisms in roots. TBR plays a role in pectin *O*-acetylation, and this is associated with pectin modifications in the cell wall[15,37], and *tbr* mutants display increased levels of methylesterified pectin in etiolated seedlings[15]. To test whether the increased level of methylesterified pectin has a connection to the *tbr* mutant root phenotypes that we had observed in high Zn conditions, we measured pectin content and the degree of pectin methylesterification in roots. We found that the *tbr* mutants showed lower pectin content compared to wildtype in high Zn conditions (Supplementary Fig. 12a). However, the *tbr* mutants exhibited 2-fold higher levels of the degree of pectin methylesterification in roots compared to the wildtype when grown in high Zn medium (Supplementary Fig. 12b). This was mainly due to a decrease of pectin methylesterification in the wild type that didn't occur in the mutants. Overall, these results indicate that a decrease in the levels of high-methylesterified pectin may play an important role in Zn toxicity tolerance and this decrease is dependent on a functional *TBR* allele.

To test whether *TBR* allelic variants, which determine natural variation in root growth responses to Zn toxicity, affect pectin methylesterification, we measured the pectin methylesterification degree of the lines complemented with the different *TBR* alleles. Consistent with the model, that higher *TBR* expression that is conferred by *TBR* genetic variants leads to lower pectin methylesterification, the *TBR_Vår2-6* complemented lines displayed a lower degree of pectin methylesterification than the *TBR_Eds-1* complemented lines in high Zn conditions (Fig. 6a). Similar results were observed in *TBR* promoter allelic complementation, where the $TBR_{Vår2-6}$-$TBR_{Col-O}$ variant showed a lower degree of pectin methylesterification than the $TBR_{Eds-1}$-$TBR_{Col-O}$ transformed lines (Fig. 6b). In conjunction with the very robust, previously published data that an impairment of *TBR* function in the *tbr* mutant leads to a lower capacity to retain divalent $Zn^{2+}$ cations in the cell walls of etiolated seedlings compared to the wild-type in high Zn[15], our data suggest that under high Zn conditions sequence variation in the *TBR* promoter confers higher TBR expression levels that lead to lower pectin methylesterification that enable root tolerance to elevated Zn levels via Zn cell wall sequestration (Fig.6c).

### *TBR* is required for Zn toxicity tolerance in crop plants

The important role of *TBR* in pectin methylesterification in root cell walls and thereby to mitigate Zn toxicity, prompted us to explore whether the same mechanism might be relevant for other plant species. TBR shows high similarity to DUF231 protein homologs in other plant species[37,38]. To determine how well-conserved TBR is across other plant species, we conducted BLAST searches using the TBR protein sequence. We found many members of the Trichome Birefringence like (TBL) gene family containing a plant-specific DUF231 domain in monocots like *Oryza sativa* and other dicots such as *Lotus japonicus* (Supplementary Fig. 13). TBR shared a high identity with rice protein OsTBL31 (LOC_Os02g53380) and OsTBL32 (LOC_Os06g10560), and *Lotus japonicus* protein LjTBR (Lj1g3v4350070) (Supplementary Fig. 13, 14a).

We first tested whether the *Lotus japonicus* TBR homolog *LjTBR* is involved in Zn toxicity tolerance. For this, we obtained a mutant line in the first exon of *LjTBR* from the LORE1 insertion mutant collection (Supplementary Fig. 14b)[39]. Much like the *tbr* mutants in *Arabidopsis*, the root growth of *Ljtbr* lines was more sensitive to high Zn, and the root length of *Ljtbr* was reduced by 15% compared to Gifu (WT) (Fig. 7a, b). This result demonstrates that a *TBR* homolog is required for Zn tolerance in the legume *Lotus japonicus,* and TBR function for Zn tolerance maybe conserved throughout dicot plants.

We then set out to test whether TBR function is also conserved in rice, a member of the monocotyledons group. For this, we generated knockout lines of *OsTBL31* and *OsTBL32* using the CRISPR/Cas9 system (Supplementary Fig. 14b, c). Consistent with the root phenotype in *Arabidopsis*, both independent knockout lines of *OsTBL32* were more sensitive to high Zn compared to ZH11 (WT), while there was no significant difference in control conditions (Fig. 7c, d). However, a sensitive phenotype in high Zn was not observed in the *Ostbl31* mutants (Fig. 7d). These results demonstrate that the *TBR* homolog *OsTBL32* is required for Zn toxicity tolerance in rice. Taken together, our results indicate that the TBR function is required for root growth under elevated Zn levels throughout angiosperms. Collectively, these discoveries open the possibility of using genetic engineering to improve Zn toxicity tolerance into a wide variety of plant species.

## Discussion

We have found GWAS to be a powerful tool to identify genes associated with root growth responses to Zn toxicity. Our study detected 21 significantly associated loci for root growth responses to Zn toxicity (Supplementary Fig. 2 and Supplementary Data 3) and provides a resource for identifying genes involved in Zn tolerance in plants. We explored causal genes for only two of these loci, as outlined below, and causal genes for the other loci remain yet to be identified. This includes the most significant peak on chromosome 4. While we had tested multiple T-DNA insertion lines for several genes in proximity to the Chromosome 4 GWAS peak, we didn't find support for any of the respective candidate genes. However, this is not entirely surprising as such reverse genetics approaches are commonly hampered by genetic redundancy and robustness, and potential interactions of the genetic background and the tested genes. Further investigations, in particular for the region on chromosome 4, might be worthwhile. Nevertheless, and importantly, the validity of our GWAS approach was highlighted by the detection of a major association in the *FRD3* gene that already had been shown to be involved in natural variation of root growth responses to high Zn[11]. Interestingly, the previously described variants that had been linked to Zn tolerance variation through experiments with the two accessions Bay-0 and Sha were not very prevalent in our GWAS mapping panel (the 27-bp deletion in the promoter region observed in Bay-0 was present in 21.1% of the accessions and the 28-bp deletion observed in Sha in the promoter region was present in 12.6% of the accessions; the non-synonymous substitution N116S showed 10.7% allele frequency)[11]. In addition to these allelic variants, we identified many associated SNPs in the *FRD3* promoter in our GWAS (Supplementary Fig. 15, 16, and Supplementary Data 4). Some of these SNPs were located in binding sites for transcription factors

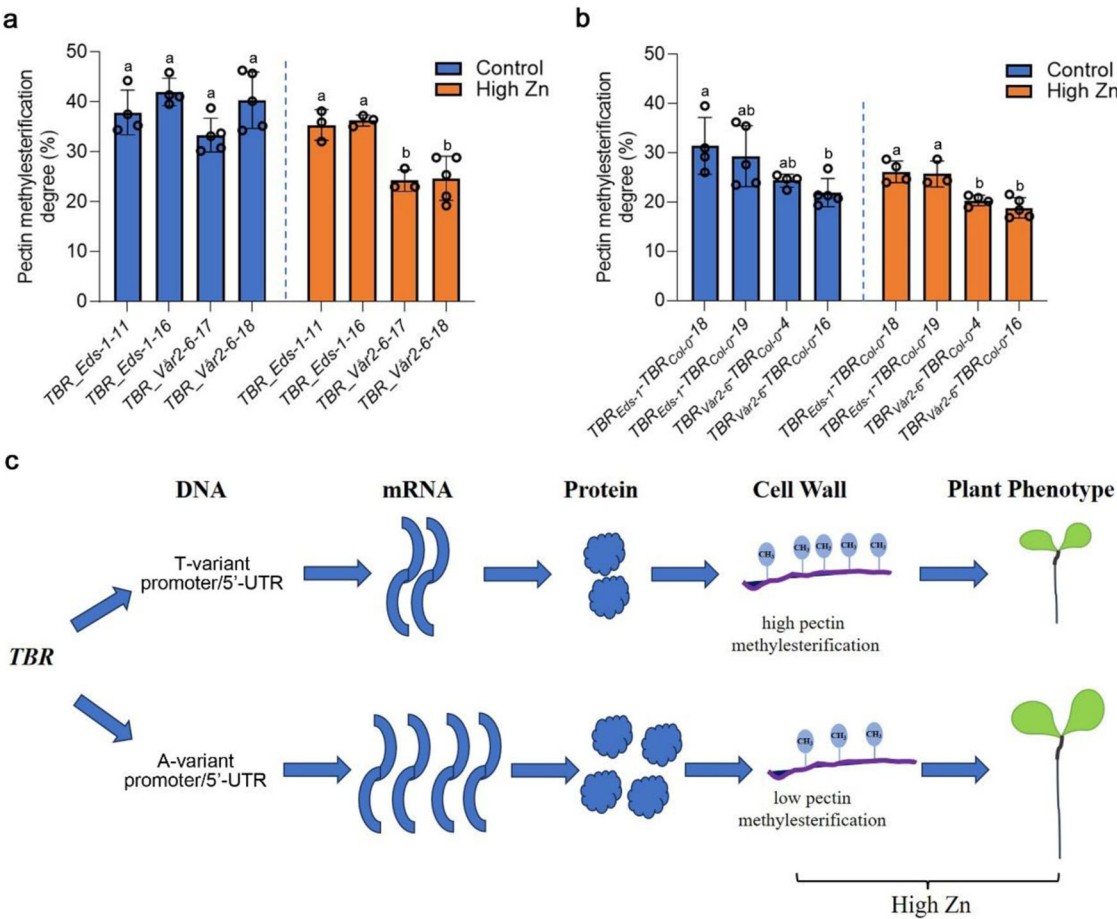

**Fig. 6 | *TBR* variants influence changes in pectin methylesterification and the mechanism by which *TBR* regulatory allelic variation confers root tolerance to elevated Zn levels. a** Pectin methylesterification degree in root cell walls of *TBR* allelic complemented lines. Data are mean ± S.D. Circles indicate a single biological replication, $n = 4, 4, 5, 5, 3, 3, 3, 5$. Statistical analysis was performed using one-way ANOVA analysis with Tukey's HSD test ($p < 0.05$). **b** Pectin methyl esterification degree in root cell walls of *TBR* promoter allelic complemented lines. For the analysis, the seedlings were grown directly on control and high Zn (200 µM) medium for 7 DAG and then harvested. Data are mean ± S.D. Circles indicate a single biological replication, $n = 4, 5, 4, 5, 4, 3, 4, 5$. Statistical analysis was performed using one-way ANOVA analysis with Tukey's HSD test ($p < 0.05$). **c** A proposed model illustrates how *TBR* regulatory allelic variation confers root tolerance to elevated Zn levels. *TBR* allelic variation in promoter and/or 5′-UTR induces higher mRNA levels and higher protein accumulation, which leads to lower pectin methylesterification that facilitates more Zn sequestration in root cell walls. Source data are provided as a Source Data file.

(Supplementary Data 5). For example, a SNP was in the binding site of STOP1 (2,570,777-2,570,792). STOP1 was previously shown to be necessary for *AtMATE* expression in aluminum tolerance[40]. Consistent with the root growth response to high Zn between these two groups of accessions, the transcript level in the T-allelic group accessions were significantly higher than in the C-allelic group accessions (Fig. 1c and Supplementary Fig. 3). These results indicated that *cis*-regulatory elements might also exist in the *FRD3* promoter region that control *FRD3* induction in response to Zn toxicity.

Even more importantly, we identified allelic variation in the *TBR* gene to be causal for root growth variation to high Zn. The *TBR* gene influences Zn homeostasis, and *tbr* mutants were previously shown to display Zn hypersensitivity in roots[15]. Our study expanded our comprehension of TBR's role by showing that *TBR* allelic transgenic lines displayed significantly different root growth in high Zn, and A-allelic transgenic lines showed more Zn toxicity tolerance than T-allelic transformed lines (Figs. 4, 5). It is interesting to note that one of the T-allelic complemented lines (*TBR Eds1-11*) showed a lower high-Zn root growth tolerance than the *tbr-3* loss of function mutant. This might indicate that in some circumstances low expression of *TBR* might render roots more sensitive than a full loss of function. Another intriguing observation from the complementation experiments was that the TBR protein containing an amino-acid substitution (L186R)

that was observed in the more tolerant A-allelic group of accessions did not rescue the *tbr-3* mutant root growth phenotype under high Zn conditions when driven by the Col-0 promoter (Fig. 5a). This might suggest that either a non-or less-functional version of the TBR protein might confer Zn-tolerance when expressed at higher levels (as conferred through the higher expressing A-allelic *TBR* variant promoters) or that the genetic background of A-allelic accession can confer full functionality to the TBR-L186R variant (e.g., by expressing certain variants of interacting proteins or genes in the pathway). Overall, further investigation into the molecular mechanism of TBR function will be very interesting and might illuminate interesting engineering strategies for TBR facilitated Zn tolerance. While it is not possible to conclude that the observed natural variation is a result of natural selection for Zn tolerance or similar stresses or a fortuitous fixation of alleles in this selfing species, it highlights a role of TBR in the variation of root growth responses to Zn toxicity and its genetic and molecular mechanisms. Previous studies have found instances in which allelic variation that affects transcript levels[30,32,41] or alters proteins[25,42] plays a critical role in trait variation. In line with this, we have found that the associated natural alleles of *TBR* confer distinct mRNA and protein expression levels that explain the ability of these alleles to confer Zn tolerance. This strongly suggests that regulatory SNPs are underlying the natural variation of root growth responses to Zn toxicity (Figs. 4, 5).

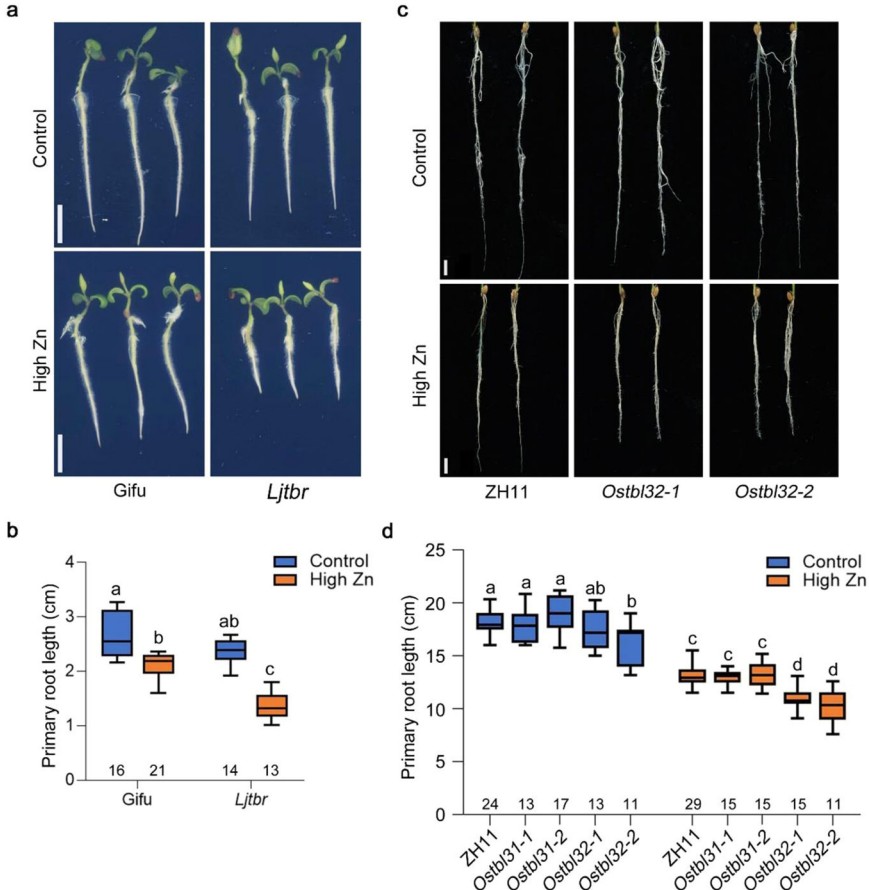

**Fig. 7 | TBR function is required for Zn toxicity tolerance in *Lotus japonicus* and rice. a** Representative pictures of root growth of Gifu and *Ljtbr* in control and high Zn (200 μM) medium for 8 days. Scale bars: 5 mm. **b** Boxplots for primary root length of Gifu and *Ljtbr* depicted in (**a**). The number below each box indicates the number of replicates. Statistical analysis was performed using one-way ANOVA analysis with Tukey's HSD test ($p < 0.05$). **c** Representative images of CRISPR/Cas9 knockout lines of *Ostbl32* and wildtype in control and high Zn (1000 μM)

hydroponic solution for 7 days. Scale bars: 10 mm. **d** Boxplots for primary root length of *Ostbl31*, *Ostbl32* mutants, and wildtype depicted in (**c**). The number below each box indicates the number of replicates. Statistical analysis was performed using one-way ANOVA analysis with Tukey's HSD test ($p < 0.05$). For box plots, the horizontal line represents the median value, the lower and upper quartiles represent the 25th and 75th percentile, and the whiskers show the maximum and minimum values. Source data are provided as a Source Data file.

Between the two groups of accessions for which we elucidated differences in transcript and protein levels using complementation experiments, we found seven SNPs in the *TBR* promoter region (2213 bp upstream of the transcriptional start site) and one SNP in the 5′-UTR. These SNPs might mediate the observed differences via *TBR* transcript levels or via joint effects on transcript expression and posttranscriptional processes such as transcript stability or translation rates. The higher expression level leads to TBR dependent lower pectin methylesterification (Fig. 6a, b). This had been shown to be accompanied by higher levels of Zn retained in the cell wall in etiolated seedlings[15]. These findings are in line with the that the modification of cell wall components can enhance metal binding ability in roots, especially in root cell walls in response to metal stress[15,43–46]. TBR encodes an *O*-acetyltransferase responsible for pectin acetylation[15,47,48]. The *tbr* mutant showed predominant alterations in pectin and pectin modification by Micro-Fourier-transformed infrared (μFTIR) spectra[15]. Furthermore, Sinclair et al. described that the *tbr* mutant exhibited an increased level of pectin methylesterification and a decreased level of pectin acetylation and Zn content in the cell wall[15]. Similarly, our study revealed that Zn exposure induced an increase in pectin methylesterification in root cell walls of *tbr* mutants. However, there are differences between Sinclair et al.'s findings and ours[15]. In Sinclair et al. [15], cell wall pectin methylesterification was slightly increased in Col-0 and *dez* (*tbr*) upon high Zn levels. In our study, we find a decrease of cell wall pectin methylesterification in Col-0 upon

high Zn conditions and an increase in *tbr* (Supplementary Fig. 12a, b). It seems most likely that these differences are due to the different growth conditions, seedling ages, or different tissues in the studies. Moreover, our study revealed that Zn sensitive T-allelic variant transgenic lines showed higher pectin methylesterification than Zn tolerant A-allelic variant transgenic lines (Fig. 6a, b). Consistent with the previously observed Zn enhanced photomorphogenesis[15], we found that the *tbr-3* mutant showed a shorter hypocotyl length compared to wildtype, and the hypocotyl length was rescued to almost wildtype levels through complementation with a functional *TBR* allele in high Zn conditions, while the hypocotyl length showed no significant difference in control conditions (Supplementary Fig. 17a). Moreover, T-allelic variant complementation lines displayed shorter hypocotyl lengths compared to the A-allelic variant complementation lines in etiolated seedlings (Supplementary Fig. 17b, c). These results indicate a similar mechanism of *TBR* variants in influencing basal tolerance to excess Zn and in suppression of photomorphogenesis most likely due to the effect that *TBR* variants have for structural modification of cell walls that lead to enhanced Zn sequestration and increased Zn tolerance.

Finally, *TBR*'s relevance for root tolerance to Zn toxicity is conserved in other species. We showed that TBR function is required for tolerating high Zn levels in the legume *Lotus japonicus* and also in the monocot rice (Fig. 7). Thus, *TBR* represents a prime target to develop crop varieties tolerant to Zn toxicity. Modification of the regulating

regions in the promoter might improve crop varieties for tolerance to Zn toxicity stress. Similar to the polymorphism patterns of *TBR* in *Arabidopsis*, rice also shows a large number of SNPs variation in the promoter region which included 43 SNPs and 5 INDELs of *OsTBL32* (http://ricevarmap.ncpgr.cn/v2/vars_in_gene/) (Supplementary Fig. 18). It would be interesting to test whether allelic variation in the cultivars show different root growth responses under high Zn conditions. If this was caused by specific regulatory variants, CRISPR/Cas9 technologies could then be efficiently utilized to engineer relevant varieties to increase Zn tolerance in rice, which might provide us with novel strategies for Zn biofortification in rice and for the remediation of Zn-contaminated soils.

## Methods

### Plant material and growth conditions

The 317 accessions of *Arabidopsis thaliana* were part of 1001 genome accessions (Supplementary Data 1). The *frd3-3*, *frd3-7*, *tbr-1* and *tbr-3* were from Col-0 background, respectively. *frd3-7* and *tbr-1* were described and characterized[37,49]; *frd3-3* is from Col-0 background as described in Delhaize[50] where it was named *man1* but then got renamed to *frd3-3* in Rogers and Guerinot[51]. The mutants of *frd3-7* (SALK_122235C), *tbr-1* (CS3741), *tbr-3* (SALK_058509C), as well as *at5g06710* (SALK_062866) were purchased from Arabidopsis Biological Resource Center (ABRC, OHIO, USA). The primers used for genotyping the mutants are shown in Supplementary Data 6.

*Arabidopsis thaliana* seeds were surface sterilized in opened 1.5 mL Eppendorf tubes in a sealed box for 1 h using chlorine gas generated from the mixture of 200 mL 6 % sodium hypochlorite and 3.5 mL 37% hydrochloric acid and then were stratified at 4 °C for 3 days in water and darkness. The seeds were placed on ½ × Murashige and Skoog (MS) agar plates (containing ½ × MS salt mixture, 1% (w/v) sucrose and 1% (w/v) agar, pH 5.7). High Zn medium was ½ × MS agar medium containing 300 μM ZnSO$_4$•7H$_2$O. This condition was used for GWAS screening and root phenotyping of *frd3* and *tbr* mutants. For cultivation in darkness, the seedlings were wrapped in 2 layers of aluminum foil. For further experiments, we used the other 1/2 MS media. The media contains 10.3 mM NH$_4$NO$_3$, 624 μM KH$_2$PO$_4$, 750 μM MgSO$_4$•7H$_2$O, 9400 μM KNO$_3$, 1500 μM CaCl$_2$, 2.5 μM KI, 50 μM H$_3$BO$_3$, 66 μM MnSO$_4$•H$_2$O, 15 μM ZnSO$_4$•7H$_2$O, 0.52 μM Na$_2$MoO$_4$•2H$_2$O, 0.5 μM CuSO$_4$•7H$_2$O, 0.52 μM CoCl$_2$•6H$_2$O, 50 μM Na-Fe-EDTA.

The *Lotus japonicus* mutant of *Ljtbr* (30009393) was ordered from Lotus Base (https://lotus.au.dk/)[39], and Gifu was a wild type control. The seeds were scraped vigorously with sandpaper for 2 min, sterilized in 0.5 % sodium hypochlorite for 18 min, and then rinsed with sterile water 4 times. Seeds were soaked with sterile water for 2 h, germinated on wet filter paper at 21 °C for 3 days, and then transferred to ½ × MS and high Zn (200 μM) medium. The primers used for genotyping are shown in Supplementary Data 6.

Rice seeds were soaked in water at 30 °C for 2 days in the dark, and then transferred to distilled water for another 2 days after germination. The seedlings were grown on control (1/2 Yoshida solution, pH 5.6) and high Zn (1000 μM) solution for 7 days. The solution was changed every 4 days. Seedlings were grown in the growth chamber under a cycle of 30 °C/16 h light and 28 °C/8 h dark and 75% humidity.

### Root phenotyping and GWAS

12 seedlings of 317 natural accessions were grown on control (½ MS) and high Zn (300 μM) conditions. The 12 seedlings of each accession were distributed on four plates (three seeds on each plate). Plant images were taken with 8 CCD flatbed scanners (Epson, Japan) from day 1 to day 8 after germination and then processed by the BRAT software[52]. GWAS was performed with root length in Zn tolerance (root length of high Zn relative to the control). GWAS was conducted

using 1001 full-sequence in a mixed-model that corrects for population structure on the GWA-portal (https://gwas.gmi.oeaw.ac.at/)[53]. Linkage disequilibrium (LD) was analyzed on the GWA-portal.

### Broad sense heritability calculation

The primary root length of all individuals from 317 natural accessions both on control and high Zn conditions were used to calculate the broad-sense heritability $h_B^2$

$$hB^2 = V_G/V_P \tag{1}$$

which is defined as the proportion of genetic variation ($V_G$) in phenotype variation ($V_P$).

### Plasmid construction and generation of transgenic plant lines

To evaluate *TBR* allelic function, we generate the plasmid of *pTBR::TBR-HA-mCITRINE-HA*. Firstly, the *TBR* promoter region and 5′-UTR (2437 bp fragment upstream of the *TBR* translational start codon) was PCR amplified using primers CP01 and CP02 from Eds-1 (T-allele), Vår2-6 and Col-0 (A-allele) genomic DNA. Gateway vector pDONR P4-P1r was amplified using primers CP03 and CP04, and then the linearized vector was recombined with the *TBR* promoter via Gibson assembly cloning (NEB). Next, the protein-coding sequence (from the translational start codon to stop codon, 1824 bp) of TBR was amplified using primers CP05 and CP06 from the cDNA of Eds-1, Vår2-6, and Col-0 and recombined with pDONR221. Finally, pDONR P4-P1r carrying *TBR* promoter, pDONR 221 carrying *TBR* coding sequence, and pDONR P2r-P3 carrying *HA-mCITRINE-HA* were recombined with Gateway destination vector pB7m34GW to obtain 3 constructs that contained the respective *TBR* promoter sequences and protein-coding sequences. To generate *pTBR::TBR_Col-0-HA-mCITRINE-HA* (the *TBR* allelic promoter-fusion construct), we recombined pDONR P4-P1r carrying *TBR* promoter sequence from two natural accessions (Eds-1 and Vår2-6), pDONR 221 carrying *TBR* protein coding sequence from Col-0, and pDONR P2r-P3 carrying *HA-mCITRINE-HA* with pB7m34GW vector. To generate *pTBR::TBR^{L186R}:HA-mCITRINE-HA* plasmid, a point mutation (G → T) was introduced into *TBR* by PCR amplified from pDONR221_Col-0 using primers CP05 and CP06 and then integrated the *TBR* promoter from Col-0 (in pDONR P4-P1) and *HA-mCITRINE-HA* (in pDONR P2r-P3) into pB7m34GW. All the constructs were transformed into *tbr-3* using the Agrobacterium tumefaciens GV3101 floral dipping method[54]. The transformants were selected on the appropriate antibiotic (glufosinate for mCITRINE). In the T1 generation, 24 independent lines were selected for each construct. From these, 5 independent transgenic lines were selected for further characterization according to the following criteria: (1) single insertion line (the T2 segregation ratio between sensitive and resistant followed the expected Mendelian distribution for a dominant single locus of 75% resistant and 25% sensitive); (2) a visible fluorescence signal of the mCITRINE fusion protein detected in the root by confocal microscopy (with no regard to the strength or the specific construct); (3) no obvious abnormal root growth phenotype. This selection was unbiased and random in the sense that the first 5 lines that displayed these properties were selected. In the T3, homozygous lines were selected using similar criteria as in the T2 generation, except that we selected 100% resistant homozygous lines. The cloning and sequence primers of transgenic lines were listed in Supplementary Data 6.

For *OsTBL* knockout lines, a single guide RNA of *OsTBL* was designed using the CRISPR-P tool (http://cbi.hzau.edu.cn/cgi-bin/CRISPR#opennewwindow) and then transferred into the pBGK032 binary vector. The *pBGK032-OsTBL* constructs were transformed into the ZH11 rice variety via Agrobacterium-mediated transformation. The genotypes of transgenic lines were tested by the primers in Supplementary Data 6.

## Quantitative real-time PCR

For qRT-PCR analysis, the seeds were grown directly on 1/2 MS agar medium for 5 DAG and then transferred to control and high Zn (300 μM) liquid medium for 12 h. Total RNA was extracted using a Spectrum Plant Total RNA Kit (Sigma). Total RNA was used as the template for first-strand cDNA synthesis with Maxima H Minus cDNA Synthesis Master Mix (Thermo Fisher). qRT-PCR was performed on a CFX384 Real-Time PCR Detection System (Bio-rad). Three biological replicates and two technical replicates were analyzed for each gene. Relative expression of genes was normalized using *EF1a* (*AT5G60390*) as an internal control. Primers used for qRT-PCR are shown in Supplementary Data 6.

## Western-blot analysis

The seedlings were grown directly on 1/2 MS medium for 7 days, followed by transferring them to control and high Zn (300 μM) hydroponic solution for 12 h. The roots (about 20 roots) were harvested and subjected to protein extraction. 100 μL protein extraction buffer was added to the samples and centrifuged at $15,133 \times g$ at 4 °C for 10 min. 50 μL supernatant with 12.5 μL loading buffer was transferred to a new tube and then boiled for 5 min. 10 μL samples were loaded and separated in a 5–25% gradient polyacrylamide gels. The following antibodies were used for immunoblotting: anti-HA-peroxidase, high-affinity clone 3F10 (1:2000, Catalog no.12013819001, Roche), anti β-Actin (1:4000, CW0264M, CWBIO), and goat anti-mouse IgG (H + L)-HRP conjugated (1:4000, CW0102). Two independent biological replicates were performed for the western blot.

## Microscopy

The seedlings of *Arabidopsis* were imaged with a Zeiss LSM 980 confocal microscope. mCITRINE was excited with a 488 nm laser power and emission was collected at 505–550 nm; propidium iodide (PI) dyes was excited with a 561 nm power and emission was collected at 590–660 nm. A Similar confocal setting was used when comparing fluorescence intensity.

## Measurements of cell length in roots

*Arabidopsis* seedlings were stained with 15 μM PI for 5 min and visualized with a Zeiss LSM 980 confocal microscopy. The meristem zone length was measured from the quiescent center to the first elongated cortex cell. The elongation zone length was measured from the onset of the rapid longitudinal cell expansion to the cessation. The mature cell length of fully elongated cells was measured in the cortex tissue layer[42].

## Root cell wall extraction

The root crude cell wall was extracted according to the method of Zhong and Läuchli[55]. First, roots were cut from the seedlings and ground with a mortar and pestle in liquid nitrogen. The powder was suspended in 75% ethanol and kept in an ice-cold water bath for 20 min. The samples were then centrifuged at $3320 \times g$ for 10 min, and the supernatant was discarded. The pellets were suspended and washed with acetone, methanol:chloroform (1:1, v/v), and methanol, respectively. Each treatment lasted 20 min and each supernatant was discarded after centrifugation. The final pellets were dried at 65 °C for the degree of pectin methyl esterification assay.

## Determination of the degree of pectin methylesterification

The pectin fraction was extracted from the cell wall by 5 mL 0.5% ammonium acetate buffer containing 0.1 % NaBH₄ (pH 4) at 100 °C for 1 h. The supernatant was the pectin. The degree of pectin methyl esterification was determined by the ratio of the methanol and uronic acid content[56]. The uronic acid content in the pectin fraction was assayed according to the method of Blumenkrantz and Asboe-Hansen using Galacturonic acid as a calibration standard[57]. Briefly, 0.2 mL of pectin extraction was incubated with 1.2 mL of sulfuric acid/tetraborate at 100 °C for 5 min. After cooling, 20 μL of 0.15% M-hydroxydiphenyl was added to the solution. And then absorbance at 520 nm was measured with a Beckman spectrophotometer after staying at room temperature for 20 min.

To determine pectin methylesterification, a colorimetric method was performed[58]. Methanol was quantified using alcohol oxidase (AO) and Purpald. The incubation solution contained pectin extraction, 200 mM phosphate buffer (pH 7.5), and 0.01 U μL−1 alcohol oxidase. After 10 min at 30 °C, 100 μL of 0.5 M NaOH solution containing 5 mg mL⁻¹ Purpald (4-amino-3-hydrazino-5-mercapto-1,2,4-triazole) was added. After an additional 30 minutes at 30 °C, H₂O was added to the samples for a final volume of 1.5 mL. Absorbance was determined at 550 nm. The pectin methylesterification degree of *TBR* allelic variant lines and *TBR* promoter allelic complemented lines was quantified using the Comin kit and protocol (GJJ-1-W, Comin, China).

## Phylogenetic analysis

Rice TBLs were identified based on the annotation of the rice genome database (Rice Genome Annotation Project, http://ricedata.cn). *Lotus japonicus* TBLs were identified based on the annotation of the lotus base (Lotus base, http://lotus.au.dk). A phylogenetic tree covering 46 TBLs in Arabidopsis, 66 TBLs in rice, and 57 TBL members in *Lotus japonicus* was generated using neighbor-joining with MEGA 10 software (1,000 bootstrap replications). As some TBL members showed low identity with TBR which prohibits estimation of the pairwise distance by MEGA 10, 4 TBL members for which the identity is lower than 30% were deleted, this included OsTBL7, OsTBL42, OsTBL62, and OsTBL66 in *Oryza sativa*.

## Quantification and statistical analysis

All values are presented as box plots or as means±standard deviation (S.D.). The number (n) of samples for each value is indicated in figures or figure legends. For box plots, the horizontal line represents the median value, the lower and upper quartiles represent the 25th and 75th percentile, and the whiskers show the maximum and minimum values. Significant differences between the two samples were analyzed with a two-tailed Student's *t*-test. Significant differences for multiple comparisons for single-point experiments were determined by one-way or two-way ANOVA with Tukey's HSD test ($p < 0.05$) as indicated in figure legends. Each experiment was repeated independently at least two times with consistent results.

## Reporting summary

Further information on research design is available in the Nature Portfolio Reporting Summary linked to this article.

# Data availability

All data supporting the findings of this study are available within the paper and its Supplementary Information files. Source data are provided with this paper.

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

## Acknowledgements

We thank all the lab members from the Busch and Hu labs for valuable discussions and comments. We thank Yintong Chen for carefully reading the manuscript and Takehiko Ogura for thoughtful discussion. We are grateful for Chory lab from Salk Institute for sharing the vectors of pDONR-P4P1R, pDONR221, pDONR P2r-P3_HA-mCITRINE-HA, and pB7m34GW. We thank Jie Xiong (Zhejiang Sci-Tech University), Chunquan Zhu (China National Rice Research Institute), and Hai Zhou (South China Agriculture University) for sharing the protocols for measuring pectin content and the degree of pectin methylesterification. The work is supported by start-up funds from the Salk Institute for Biological Studies and funds from the Hess Chair in Plant Science (W. Busch), the Zhejiang Provincial Natural Science Foundation of China (No. LDQ23C130001), the National Natural Science Foundation of China (No.32188102, No. 32071991), the Key Research and Development Program of Zhejiang Province (No. 2020R51007, No. 2021C02056-1) and the National Natural Science Found of China (No. 32172656, B. Li).

## Author contributions

W. B., B. L., P.H., and K.Z. conceived and designed all experiments. K.Z. performed the experiments on *Arabidopsis thaliana* and *Lotus japonicus*. P.Z. performed experiments on rice and analyzed rice data. W.H. contributed to *Arabidopsis* zinc tolerance screening and GWAS analysis. A.M. contributed to *Lotus* zinc tolerance experiments. L.Z. contributed to performing data analysis. M.C. contributed to performing western blot. M.P.P. contributed reagents and contributed to microscopy experiments. X.W., S.H., and S.T. contributed reagents and administrative assistance. K.Z., W.B., B.L., P.Z., and P.H. wrote the manuscript. W.B., P.H., and B.L. provided reagents, funding, and supervision.

## Competing interests

The authors declare no competing interests.
