## [Peer Review File · Nature Communications]

Natural variation of TBR confers plant zinc toxicity tolerance through root cell wall pectin methylesterificationReviewers' Comments:

Reviewer #1:

Remarks to the Author:

The authors assessed seedling tolerance to excessive Zn exposure in a panel of over 300 diverse *Arabidopsis thaliana* accessions based on root growth of plants grown on vertical agar plates containing 300 μM Zn^{2+} . They identify two quantitative trait loci (QTL) exceeding the strict Bonferroni corrected significance threshold, and several more that exceeded the less strict Benjamini Hochberg corrected significance threshold. The authors followed up on two of these QTL, and identified two genes underlying quantitative trait locus variation, the FRD3 gene (FERRIC REDUCTASE DEFECTIVE 3) and the TBR gene (TRICHOME BIREFRINGENCE). Genetic variation at the FRD3 gene had been implicated before to cause a QTL, for the TBR gene this was so far unknown. The functional relevance of the TBR gene in conferring Zn tolerance was studied, and revealed the gene to control the demethylesterification of pectin in wild-type plants. The actual molecular function of the TBR protein was not elucidated.

Both the identification of the FRD3 locus and of the TBR locus are important findings, as well as the establishment of the role of the TBR protein in controlling pectin demethylesterification. It is interesting to establish this based on natural genetic variation, and not on mutant analysis, as such information can be useful in development of breeding approaches to acquire Zn tolerant crops, where natural allelic variation will be needed to select on.

This is a very substantial piece of work, which will certainly be interesting to the field, a nice example of natural genetic variation for an agronomically important trait, found in a model species, with translational research into a crop (rice).

In general the manuscript is well written, though it could certainly benefit from a round of textual editing. I would like to see more clarity on the evidence that allelic variation of both genes is indeed causal to the association found in the GWAS, as specified below. Also the description of M&M and the legends to the (supplementary) figures, could be improved.

Major concerns

About the FRD3 QTL, since the alleles found in this analysis are different from the ones detected by Pineau et al., one would need to provide evidence that the allelic variation at this locus is causal to the identified association. Fig 1c and sup fig 3 suggest this to be due to promoter differences and transcript level differences. One could confirm causality by quantitative complementation, which would be preferred, but this will take considerable time and effort. Since this is a known Zn tolerance gene, though with novel allelic variation, it will be sufficient to demonstrate causality based on allelic differences in FRD3 transcription. To test this in only two accessions is fine for comparison of all genes in the region, but not to convincingly show the allelic difference resides in the promoter of FRD3 causing difference in transcription. Best would be to establish the transcript differences between the "G" and "T" alleles in (at least) five of the representative accessions of each allelic class. In addition, the FRD3 promoter sequences of the used accessions should be provided, not based on whole genome resequence data as shown in sup fig 3. This may identify any causal cis-elements, missing in the sensitive alleles, which would be an important finding, adding to the Pineau et al. paper. Sequencing of the promoters has been performed for the TBR gene in several accessions, but should be considered for the FRD3 gene too. In addition, authors should also discuss why the allelic variation found by Pineau is not found in the panel used for GWAS. I can imagine this is due to the allele frequency, but such will need to be verified and clarified.

There are several issues that will need some clarification about establishing the TBR gene to be underlying the association found on chromosome 5. It start with the observation (l. 141), that the TBR transcript level differs between allelic variants. However, Strangely enough, this is not obvious from fig

3. There, AT5G06710 shows to be more induced upon high Zn than TBR, although not in all accessions, and allelic groups, but it could be equally likely the causal gene. This needs better explanation. Reference is made to sup fig 4, but this figure lacks any explanation in the legend of what can be seen in the figure, so it is not obvious. Bit worrying here is that Sup Fig 4c does not show high root expression, and appears to be quite different from what is shown in Sup Fig 4d. A better legend, and better explanation of the interpretation is needed here.

Mutants were obtained for both TBR and AT5G06710 (l. 148), but I could not find any confirmation of their mutant genotypes. Are these indeed knock-out mutants? If so, how was this determined? I assume all the checks have been made, but as they are not shown, this will need to be clarified in the manuscript. The root growth phenotype of the AT5G06710 mutant is not shown, while the result is reported (l. 156), the results should be included, eg. in supp fig 5. It was not clear to me why authors chose to continue with *tbr-3* rather than *tbr-1*, while the latter had the strongest mutant phenotype. There is likely to be an obvious reason, so clarify this. Transgenic complementation of the *tbr-3* mutant is important (l. 166), but this experiment is poorly explained. Describe the construct used (TBR_Col-0) in the text and in fig 4. Also describe how many independent T3 lines are used for Fig 4d. I assume there would be at least three, but this is not clear. Very good to see that single homozygous transgenics are used. I did not get clear from M&M how this was established, add this to M&M. If only one single homozygous transgenic was used, best show the results of the other transgenics, with multiple inserts. For the complementation with the T- and A-allele TBR promoter and coding sequence constructs, only two independent transgenic lines are used. That is not much, especially since the Eds-1 transgenics show different phenotypes. What is the reason of the difference? There surely must have been more transformation events than just two! 10 would be more appropriate. Are the transgenes expressed at all? This needs explanation. I assume also wild-type Col-0 has been transformed with these constructs, as control, but this is not shown. Has this not been done, and if so, why not?

The difference in protein expression between the Eds-1 and Vår2-6 alleles is much more prominent than the difference in transcript expression (l. 186-188; fig 5), and this could be made more specific, as this is more convincing that the allelic differences in transcript expression. If the protein expression analysis was also performed on the TBREds-1-TBRCol-0 and TBR Vår2-6-TBRCol-0 transgenics, it would have been further proof that the allelic difference is solely due to the promoter differences, and not due to any signals controlling protein stability. Has this been done?

While the analysis of the non-synonymous SNP partially distinguishing the T- and A-alleles of TBR is interesting, it is not clear how this could explain the association, as not all A-allele genotypes carry this SNP. This could be made more clear.

The conclusion in l. 216-219 is not very strong and convincing, as the TBR expression in the original accessions of each of the two allelic classes, is not very different (fig 3c), and since only two independent transformants per allelic promoter-fusion construct have been examined. Has overexpression of the gene been tested to confirm this? Single copy transformants are used, but perhaps correlation analysis of TBR expression and Zn tolerance phenotype in T2 or T3 of all transformants, from several independent transformation events, would provide more convincing evidence that the difference in transcription of TBR between A and T- alleles is causal? Would such lines be available?

Some minor concerns:

l. 29 TBR in italics?

l. 55 Looks like a relevant reference on the role of PME in Arabidopsis Zn tolerance is missed: Weber et al., Plant J. 76:151-164. <https://onlinelibrary.wiley.com/doi/10.1111/tpj.12279>.

l. 103. There is no explanation as to why the strong association on chr 4 is not further analysed, though it is more significant, and appears to be narrower than the chr. 5 association on TBR. Please clarify this.

l. 153/fig 4. Why are roots of *tbr-3* longer in control conditions?
l. 173. What is exactly meant with 'TBR coding sequence'? From start to stop codon, or beyond?
M&M Plasmid construction (l. 369). Without reference to origin of vectors, certainly non-commercial ones, and reference to primers used, this cloning is hard to follow and even more difficult to reproduce. I assume primers in supp. data file 4 are used to amplify the TBR promoter and coding sequences, but this is not clear. Reference to *Agrobacterium* transformation method? It is not clear how the copy number of transgenics is determined.
Figure legends are not always clear.
Eg. Fig1c. What is meaning of blue and orange bars? When are roots sampled?
Fig 5. a. What is genetic background of T3 lines?
Supp fig 4. The legend to this figure is very short, for all panels.
Supp fig 5. What are TBR-Col-0 plants? Description of genotype is lacking in legend.

Reviewer #2:

Remarks to the Author:

The authors conducted a GWAS analysis to identify natural alleles of *A. thaliana* that affect zinc tolerance of root growth. They report the identification of natural variants of the *FRD3* and *TBR* genes. The authors provide support for the involvement of both genes through Zn-sensitive phenotypes of loss-of-function mutants. The authors further suggest that the *TBR* polymorphism is associated with expression differences resulting from causal sequence variation in the promoter region. The authors provide data to suggest that in *tbr* mutants, a lower pectin content and higher degree of pectin methylesterification lead to a decreased ability to bind and sequester Zn in cell walls, when compared to the wild type. Finally, the authors report that genetic lesions in *TBR* orthologues of crop species, rice and *Lotus japonicas*, also lead to Zn hypersensitivity.

Overall, the presented results appear to be largely sound (but see below) in general, and they may potentially extend existing knowledge on *TBR* protein functions to crop species (but see below). Yet, the major findings reported here are confirmatory, and they do not sufficiently advance the state-of-the-art concerning the biochemical or molecular mechanisms underlying the observed phenotypes associated with *TBR*. As the authors state in their manuscript, Pineau et al. (2012), for example, reported the role of *FRD3* in Zn tolerance. In addition, very importantly, Sinclair et al. (2017) reported the role of *TBR* in Zn and Ni tolerance upon cultivation of seedlings in the dark as well as in the light, which Sinclair et al. (2017) then associated with underlying changes in cell wall composition and metal-binding capability of the cell wall, as suggested again in this manuscript. Furthermore, Bischoff et al. (2010) also reported changes in pectin methylesterification in light-grown *tbr* seedlings.

Additional major comments

1. GWAS for Zn tolerance in *A. thaliana* was reported before. In addition, ecologically, there is no indication so far that any *A. thaliana* populations colonize soils high in zinc. The observed differences may not be the result of natural selection but of the fortuitous fixation of weakly deleterious alleles in this selfing species.
2. The reviewer is not convinced that the diagrams in Fig. 1c and 3c show sufficiently consistent differences in transcript levels between plants carrying Zn-sensitive and Zn-tolerant alleles of *FRD3* and *TBR*, respectively. In particular, in Fig. 3c, there is no significant increase in transcript levels in two out of three Zn-tolerance allele-carrying genotypes.
3. The protein family comprising *TBR* is rather large. Authors must include phylogeny of entire protein families of *A. thaliana*, soybean and rice in order to demonstrate the orthology between *AtTBR* and the genes of which functions are addressed here in soybean and rice. Alternatively, the authors must provide other properties or characteristics based on which they identified orthologues of *AtTBR* in rice and soybean among all pother members of the protein family.

Minor comments

4. The authors should discuss their findings also in the context of possible morphological cell wall

alterations in *A. thaliana*, soybean and rice. In *A. thaliana*, these were reported to be rather extensive, especially upon cultivation in high-zinc conditions.

5. Methods are generally insufficiently detailed and the basic characterization of all mutant and transgenic lines (min.: target gene/allele transcript levels) under the conditions examined here should be provided (e.g. not provided for those shown in Fig. 5g).

6. Lines: 28/29: "which had not been characterized associated with Zn toxicity tolerance": not correct (see above)

7. Line 86: TBR a "newly uncharacterized gene" – not correct

8. Line 120: "TBR as a novel root growth regulator under high Zn levels": not novel, not a regulator

9. How do the authors reconcile the small visible differences between alleles in Fig. 5c/d with the large differences in protein levels according to the immunoblot shown in Fig. 5e?

10. Also root lengths for low and high Zn should be shown for Fig. 5g (relative data do not correct statistically for variation under control conditions).

11. The methodology provided for root Zn data (Fig. 6c/d) does not follow established procedures and is thus unlikely to reflect the Zn pools the authors intended to address here.

Reviewer #3:

Remarks to the Author:

Key results In the present study, the authors conducted a GWAS for Zn tolerance in *Arabidopsis thaliana*, through the measurement of primary root length. It leads to the identification of new genetic loci in addition to an already described one, FRD3. They focused on the TBR locus, giving genetical proof to the role of cell wall in Zn tolerance. Although the role of cell wall in Zn tolerance has already been identified, few molecular targets were identified in contrast to the numerous ones characterized for the other processes involved in Zn tolerance in plants.

Validity In general, data interpretations were clear and accurate to me. Nevertheless, few points need to be qualified or clarified :

- I wonder the interest of focusing so much on FRD3 locus. For example, the figure 2 should be presented as additional data.

- I agree that the natural variation observed in the present study are distinct from the ones from Pineau et al. but the conclusion on causal natural variation in FDR3 is not accurate. Both protein function and expression are responsible for natural variation (line 284-287). In addition, many studies identified and focused on allelic variation responsible for expression variation: Satbhai et al 2017; Baxter et al 2010; Julkowska, 2016; etc... (line 295).

Significance I agree with the proposed findings and their significance for the topic of Zn tolerance in plants. This study identifies a key molecular player in the role of the root cell wall in Zn tolerance.

Data and methodology I found the process for validating the candidate gene appropriate and quality of data and presentations suitable expect for :

- o figure 1c : the color legend is missing

- o figure 1 legend : tolerant accessions instead of tolerance

- o figure 3C : the higher expression level of TBR in A-allelic accessions is only valid for one accession (Sim-1) and only 2 values exist for Vär2-6 expression. This should be improved.

- o figure 4d : the y axis is incomplete and does not correspond to values in the Excel file provided. In addition, inadequate unit is mentioned in Excel file.

- o in figure 4 : data on root growth for Col-0/tbr3/TBR_Col-0 are missing to compare with root elongation zone length.

- o figure 6c, d : there are only two values for Zn contents in trb-3 and TRB_Col-0

- o figures 2b, 4b: It would be easier to read if relative root growth was represented as done for GWAS and in figure 5 f, g

- o In methods, the information on plants for q-RTPCR experiment is missing.

Suggested improvements The manuscript could be improved with :

- Identification of motive responsible for expression variation in TBR promoter

- Search for TBR expression in the Zn hyper tolerant relative species *Arabidopsis halleri* or *Noccaea*

caerulescens as it is known that genes involved in Zn tolerance were constitutively and highly expressed in *N. caerulescens*.

- Localization and quantification of Zn in root cell wall in extreme accessions
- Precisions on nature and localization of SNP on the gene *FRD3* on supplementary figure 3 and table S2.

Clarity and context A better description of mechanisms involved in Zn binding to cell wall components and of known Zn tolerance processes would render the manuscript clearer. For example, the use of reviews on Zn tolerance would be useful to give better image on the different processes involved in Zn tolerance, and on the presence of hyperaccumulating plants (line 47-51). For example Tang et al 2022; Lin & Arts 2012.

REVIEWER COMMENTS

We thank the editor and three reviewers for taking time, effort and constructive criticisms and suggestions on our manuscript. We have revised our manuscript and responded to all points raised by the reviewers. We believe that our manuscript has greatly improved based on the reviewers' feedback.

Reviewer #1 (Remarks to the Author):

The authors assessed seedling tolerance to excessive Zn exposure in a panel of over 300 diverse *Arabidopsis thaliana* accessions based on root growth of plants grown on vertical agar plates containing 300 μM Zn^{2+} . They identify two quantitative trait loci (QTL) exceeding the strict Bonferroni corrected significance threshold, and several more that exceeded the less strict Benjamini Hochberg corrected significance threshold. The authors followed up on two of these QTL, and identified two genes underlying quantitative trait locus variation, the *FRD3* gene (*FERRIC REDUCTASE DEFECTIVE 3*) and the *TBR* gene (*TRICHOME BIREFRINGENCE*). Genetic variation at the *FRD3* gene had been implicated before to cause a QTL, for the *TBR* gene this was so far unknown. The functional relevance of the *TBR* gene in conferring Zn tolerance was studied, and revealed the gene to control the demethylesterification of pectin in wild-type plants. The actual molecular function of the TBR protein was not elucidated.

Both the identification of the *FRD3* locus and of the *TBR* locus are important findings, as well as the establishment of the role of the TBR protein in controlling pectin demethylesterification. It is interesting to establish this based on natural genetic variation, and not on mutant analysis, as such information can be useful in development of breeding approaches to acquire Zn tolerant crops, where natural allelic variation will be needed to select on.

This is a very substantial piece of work, which will certainly be interesting to the field, a nice example of natural genetic variation for an agronomically important trait, found in a model species, with translational research into a crop (rice).

In general the manuscript is well written, though it could certainly benefit from a round of textual editing. I would like to see more clarity on the evidence that allelic variation of both genes is indeed causal to the association found in the GWAS, as specified below. Also the description of M&M and the legends to the (supplementary) figures, could be improved.

We would like to thank the reviewer for their thorough review of the manuscript and the very helpful suggestions.

Major concerns

About the *FRD3* QTL, since the alleles found in this analysis are different from the ones detected by Pineau et al., one would need to provide evidence that the allelic variation at this locus is causal to the identified association. Fig 1c and sup Fig 3 suggest this to be due to promoter differences and transcript level differences. One could confirm causality by quantitative complementation, which would be preferred, but this

will take considerable time and effort. Since this is a known Zn tolerance gene, though with novel allelic variation, it will be sufficient to demonstrate causality based on allelic differences in *FRD3* transcription. To test this in only two accessions is fine for comparison of all genes in the region, but not to convincingly show the allelic difference resides in the promoter of *FRD3* causing difference in transcription. Best would be to establish the transcript differences between the "G" and "T" alleles in (at least) five of the representative accessions of each allelic class.

1) Thank you for the thoughtful suggestion. To confirm that the allelic variation in the promoter of FRD3 causes the difference in transcription under high Zn conditions, we have quantified the FRD3 gene expression in additional set of accessions. We selected 3 C-allelic accessions (TRÅ01, Vår2-6 and Näs 2) and 2 T-allelic accessions (Spro 1 and Bay-0. Bay is a tolerant parent in Pineau et al.) [we started the experiment with 3 accessions in each group, but one accession had poor germination and it was logistically challenging to set up another round of experiments as sending seeds from the USA to China is difficult at the moment]. The results were very clear and consistent with the pattern of FRD3 gene expression in Figure 1c, the expression of FRD3 showed a significant difference between the C-variant and T-variant accessions under high Zn conditions (Supplementary Fig. 3) in the revised manuscript (Lines 335-338).

2) We have revised "G" allele to "C" allele. This was a mistake while generating the figure. It has been corrected in the text (Line 105), Fig. 1c and Supplementary Fig. 3.

In addition, the *FRD3* promoter sequences of the used accessions should be provided, not based on whole genome re-sequencing data as shown in sup fig 3. This may identify any causal cis-elements, missing in the sensitive alleles, which would be an important finding, adding to the Pineau et al. paper. Sequencing of the promoters has been performed for the *TBR* gene in several accessions, but should be considered for the *FRD3* gene too.

Thank you for the suggestion. We have added the figures of the FRD3 promoter sequence and FRD3 protein coding sequence in supplementary Fig. 14 and supplementary Fig. 15. We found many associated SNPs variations on the promoter. Furthermore, we have analyzed Dapseq derived binding sites that some SNPs variations in the FRD3 promoter were located in binding sites for transcription factors (Supplementary Data 5).

In addition, authors should also discuss why the allelic variation found by Pineau is not found in the panel used for GWAS. I can imagine this is due to the allele frequency, but such will need to be verified and clarified.

Thank you for the suggestion. We modified the discussion as "For this, we identified several associated SNPs in the FRD3 promoter in our GWAS

(Supplementary Fig. 13-14; Supplementary Data 4). Some of these SNPs were located in binding sites for transcription factors (Supplementary Data 5). For example, a SNP (2,570,777-2,570,792) was in the binding site of *STOP1*. *STOP1* was previously shown to be necessary for *AtMATE* expression in aluminum tolerance³⁸. Consistent with the root growth response to high Zn between these two groups of accessions, the transcript level in the T-allelic group accessions were significantly higher than in the C-allelic group accessions (Fig. 1c; Supplementary Fig. 3). Interestingly, the previously described variants that had been linked to Zn tolerance variation through experiments with the two accessions Bay-0 and Sha were not very prevalent in our GWAS mapping panel (the 27-bp deletion in the promoter region observed in Bay-0 was present in 21.1% of the accessions and the 28-bp deletion observed in Sha in the promoter region was in 12.6% of the accessions; the non-synonymous substitution N116S showed 10.7% allele frequency)¹⁴.” in the revised manuscript (**Lines 330-344**).

There are several issues that will need some clarification about establishing the *TBR* gene to be underlying the association found on chromosome 5. It start with the observation (l. 141), that the *TBR* transcript level differs between allelic variants. However, Strangely enough, this is not obvious from fig 3. There, *AT5G06710* shows to be more induced upon high Zn than *TBR*, although not in all accessions, and allelic groups, but it could be equally likely the causal gene. This needs better explanation.

Thank you for pointing out this issue. Based on this feedback, we clarified and revised our results as follows:

*“To identify the most likely candidate of these two genes in proximity to the significantly associated SNP, we first analyzed their gene expression. Publicly available organ and tissue specific microarray expression data (<https://bar.utoronto.ca/efp/cgi-bin/efpWeb.cgi>)³² showed that *TBR*, as well as *AT5G06710* are expressed throughout the plant (Supplementary Fig. 4c, d, f, g). Root single cell RNA sequencing (<https://phytozome-next.jgi.doe.gov/tools/scrna/>) showed that *TBR* is expressed in most root cells, while *AT5G06710* is expressed in a subset of root cell types only (Supplementary Fig. 4e, h). To then test whether the response of the candidate genes in the allelic groups of accessions that were identified by GWAS was consistent with the causal involvement of one of the two candidate genes, we selected representative accessions from the T-allelic group (*Eds-1*, *Eds-9* and *TRÄ 01*) and the A-allelic group (*Vår2-6*, *Sim-1* and *Fri 1*). Additionally, we also included *Col-0* with an intermediate root length, which was from A-allelic group. We found that the transcript level of *TBR* in *Sim-1* and *Fri 1* that are from the A-allelic accessions was higher than that of the T-allelic accessions, while *Col-0* and *Vår2-6* showed no significant expression difference compared with T-allelic accessions in high Zn conditions (Fig. 3c; Supplementary Fig. 4i). While *AT5G06710* was induced by high Zn in several accessions, there was no significant difference between the two allelic groups of accessions in high Zn conditions (Fig. 3c; Supplementary Fig. 4i). While the expression data of *TBR**

was more consistent with it being the causal genes, the observation that AT5G06710 was also induced by high Zn, prompted us to test mutant phenotypes for the two candidate genes.” in the revised manuscript (Lines 136-156).

Reference is made to sup fig 4, but this figure lacks any explanation in the legend of what can be seen in the figure, so it is not obvious. Bit worrying here is that Sup Fig 4c does not show high root expression, and appears to be quite different from what is shown in Sup Fig 4d. A better legend, and better explanation of the interpretation is needed here.

Thank you for pointing out this issue. We have extensively modified the figure legend of Supplementary Fig. 4. Supplementary Fig. 4c is derived from a whole-plant microarray study in which they normalized expression over all tissues of the plant and used whole roots. Supplementary Fig. 4d is from the root tissue specific dataset and indicates the expression differences between different tissues within the primary root. As the figures are generated from different datasets (that have been normalized differently), it is not surprising that values can look quite different.

Mutants were obtained for both *TBR* and *AT5G06710* (l. 148), but I could not find any confirmation of their mutant genotypes. Are these indeed knock-out mutants? If so, how was this determined? I assume all the checks have been made, but as they are not shown, this will need to be clarified in the manuscript.

*Thank you for the suggestion. We have added the mutant characteristics of both *TBR* and *AT5G06710* in Supplementary Fig.5 and in the revised manuscript (Lines 156-162; Lines 167-169). *tbr-1* is a loss of function mutant due to a point mutation (G>A, causing a predicted Gly to Glu exchange at position 427, which will lead to a negative charge just in front of the DUF231 domain; this is described by Bischoff et al.,2010). *tbr-3* and *at5g06710* are knockdown mutants now shown in Supplementary Fig.5b.*

The root growth phenotype of the *AT5G06710* mutant is not shown, while the result is reported (l. 156), the results should be included, eg. in sup fig 5.

*The root growth phenotype of *at5g06710* mutant is shown in Fig. 4a, b.*

It was not clear to me why authors chose to continue with *tbr-3* rather than *tbr-1*, while the latter had the strongest mutant phenotype. There is likely to be an obvious reason, so clarify this.

*We now included our reasoning (Lines 172-175): “For this and the subsequent analyses, we utilized the *tbr-3* mutant as its effects were due to the reduced expression levels (and thus more like the observed natural variation of *TBR*), while the *tbr-1* mutant displayed an amino-acid change with more drastic, but less well*

understood consequences on TBR function.”

Transgenic complementation of the *tbr-3* mutant is important (l. 166), but this experiment is poorly explained. Describe the construct used (*TBR_Col-0*) in the text and in fig 4. Also describe how many independent T3 lines are used for Fig 4d. I assume there would be at least three, but this is not clear. Very good to see that single homozygous transgenics are used. I did not get clear from M&M how this was established, add this to M&M. If only one single homozygous transgenic was used, best show the results of the other transgenics, with multiple inserts.

1) We thank the reviewer for pointing this out. We added root growth data of four complemented lines of TBR_Col-0 in Fig. 4e and revised the description as “We next tested whether these phenotypes could be rescued by transgenic complementation of TBR in the tbr-3 mutant (TBR_Col-0, the Col-0 TBR coding sequence driven by its own promoter from Col-0). Seedlings of four independent T3, single insertion homozygous complementation lines showed the recovery of the root growth (Fig. 4e). We then measured the length of the root elongation zone in high Zn conditions in one of these lines and found that consistent with the restoration of root growth under high Zn, the length of root elongation zone was also restored (Fig. 4c, d)” in the text (Lines 181-188).

2) As TBR_Col-0 (pTBR::TBR-HA-mCITRINE-HA) belongs to allelic complemented transgenics, we described the plasmid construct method, as well as the determination of single insertion, homozygous T3 lines in lines 434-444 and lines 451-458.

For the complementation with the T- and A-allele *TBR* promoter and coding sequence constructs, only two independent transgenic lines are used. That is not much, especially since the *Eds-1* transgenics show different phenotypes. What is the reason of the difference? There surely must have been more transformation events than just two! 10 would be more appropriate. Are the transgenes expressed at all? This needs explanation.

1) We have added the root length data of three T-allelic complemented lines and five A-allelic transgenic lines in Fig. 5a and modified the description “Consistent with the hypothesis that TBR variants determine root growth in high Zn conditions, five independent-homozygous lines (TBR_Vår2-6) from the A-allelic variant that was associated with higher root length and fully rescued the tbr-3 mutant phenotype in high Zn conditions (Fig. 5a). In contrast, one of the T3 lines (TBR_Eds-1-11) that had been transformed with T-allelic variant showed a significantly lower Zn tolerance than the tbr-3 mutant, while the other two lines (TBR_Eds-1-16 and TBR_Eds-1-20) showed a similar root length to tbr-3 mutant (Fig. 5a)” in lines 197-204 in the text.

2) The reason of the TBR_Eds-1 transgenics show different phenotypes presumably due to the T-DNA insertion sites or seed germination.

3) 5 independent transgenic lines were selected in the T3 generation. All the Vår2-6 transgenes were expressed, but not all Eds-1 transgenes, therefore we included five and three lines in these analyses.

I assume also wild-type Col-0 has been transformed with these constructs, as control, but this is not shown. Has this not been done, and if so, why not?

We haven't transformed these constructs to Col-0 background. We chose to do it in the tbr mutant, as the expression of a functional allele might confound the analysis. We thought that for the purpose of our study to test whether TBR allelic variants cause root growth variation in response to high Zn, this seemed like the best experiment.

The difference in protein expression between the Eds-1 and Vår2-6 alleles is much more prominent than the difference in transcript expression (l. 186-188; fig 5), and this could be made more specific, as this is more convincing that the allelic differences in transcript expression. If the protein expression analysis was also performed on the *TBR_{Eds-1}-TBR_{Col-0}* and *TBR_{Vår2-6}-TBR_{Col-0}* transgenics, it would have been further proof that the allelic difference is solely due to the promoter differences, and not due to any signals controlling protein stability. Has this been done?

We thank the reviewer for raising the point.

1) *There might be multiple reasons for the visible difference. One reason might relate to the sensitivity of antibody staining. Another reason might be differences in growth conditions: For Fig. 5c/d, we placed the seeds directly on control and high Zn (300 µM) agar medium for 6 DAG and then calculated the fluorescence intensity. For the Western blot analysis, the seeds were placed on 1/2 MS medium for 5 DAG and then the seedlings were transferred to control and high Zn liquid medium for 12 h.*

2) *We performed the experiments of TBR transcript expression and TBR western blot in independent transformants of A and T allelic promoter-fusion construct. The results have been added to the revised manuscript (Lines 245-247). The results showed that *TBR_{Vår2-6}-TBR_{Col-0}* transgenic lines showed significantly higher transcription levels and higher accumulation of protein than *TBR_{Eds-1}-TBR_{Col-0}* transformed lines in high Zn conditions. Taken together, the allelic variation in the TBR promoter, which cause different gene expression and protein expression between *TBR_{Eds-1}-TBR_{Col-0}* and *TBR_{Vår2-6}-TBR_{Col-0}* transgenics, contributes to the natural variation of root growth in high Zn conditions (Fig. 6 c, d).*

While the analysis of the non-synonymous SNP partially distinguishing the T- and A-alleles of *TBR* is interesting, it is not clear how this could explain the association, as not all A-allele genotypes carry this SNP. This could be made more clear.

We have now clarified our reasoning for testing in the text: "When analyzing the

full genome sequence at the TBR locus, we found a SNP that caused non-synonymous amino-acid change among several members of the A-allelic group at position 2,070,324 (Supplementary Fig. 8, 9). While according to the whole genome resequencing data, it was not present in all A-allele accessions due to potential whole-genome sequencing inaccuracies and potential allelic heterogeneity, we considered the possibility that it contributed to the observed root phenotype variation.” (Lines 219-225).

The conclusion in l. 216-219 is not very strong and convincing, as the *TBR* expression in the original accessions of each of the two allelic classes, is not very different (fig 3c), and since only two independent transformants per allelic promoter-fusion construct have been examined. Has overexpression of the gene been tested to confirm this? Single copy transformants are used, but perhaps correlation analysis of *TBR* expression and Zn tolerance phenotype in T2 or T3 of all transformants, from several independent transformation events, would provide more convincing evidence that the difference in transcription of *TBR* between A and T- alleles is causal? Would such lines be available?

1) Thank you for suggesting the improvement. To address this, we added the TBR gene expression results of the third accession in these two allelic variants. The results were very clear and consistent with the pattern of TBR gene expression in Fig. 3c, as the TBR expression showed a significant difference between the T-variant (TRÄ 01) and A-variant (Fri 1) accession under high Zn conditions (Supplementary Fig. 4i).

2) We have now included the root length of four T-allelic and three A-allelic variant transformants of promoter-fusion construct in high Zn conditions which the result showed that the four independent T-allelic transformant lines were not able to complement the phenotype to WT levels, while the three independent A-allelic variant, were statistically indistinguishable from the WT in high Zn conditions (Fig. 6b). Furthermore, we also performed the experiments of TBR transcript expression and TBR western blot in these transgenic lines. The results showed that TBR_{Vär2-6}-TBR_{Col-0} transgenic lines showed significantly higher transcription levels and higher accumulation of protein than TBR_{E_{ds}-1}-TBR_{Col-0} transformed lines in high Zn conditions (Fig. 6c, d). Take together, the allelic variation in the TBR promoter, which causes different gene expression and protein expression between TBR_{E_{ds}-1}-TBR_{Col-0} and TBR_{Vär2-6}-TBR_{Col-0} transgenics, contributes to the natural variation of root growth in high Zn conditions. The results have been added to the revised manuscript (Lines 238-250).

Some minor concerns:

l. 29 *TBR* in italics?

Thanks for pointing out the issue. It has been modified accordingly.

l. 55 Looks like a relevant reference on the role of PME in Arabidopsis Zn tolerance is missed: Weber et al., Plant J. 76:151-164. <https://onlinelibrary.wiley.com/doi/10.1111/tpj.12279>.

Thank you for the suggestion. This paper has been added “The impact of such cell wall modifications is apparent in ozs2 mutants that are defect in the gene encoding pectin methylesterase 3 (pectin methylesterase 3, PME3) and that showed Zn hypersensitivity¹².” in the introduction (Lines 55-57).

l. 103. There is no explanation as to why the strong association on chr 4 is not further analyses, though it is more significant, and appears to be narrower than the chr. 5 association on *TBR*. Please clarify this.

We analyzed the significant peak and candidate genes in a window of 10 kb around the top SNP on chromosome 4. However, the T-DNA mutants of the genes in that regions that we checked for growth under high Zn conditions didn't show a phenotype. We therefore couldn't conclude much from these efforts so far.

l. 153/fig 4. Why are roots of *tbr-3* longer in control conditions?

We don't know why the tbr-3 mutant grows faster under control conditions. Root length in Fig. 4b was measured at day 7 after germination while others were measured at 4 DAG. As tbr-3 shows reduced growth compared to Col-0 in high Zn conditions (and this thus constitutes an even more pronounced difference), we don't believe it affects our conclusions.

l. 173. What is exactly meant with ‘*TBR* coding sequence’? From start to stop codon, or beyond?

The TBR coding sequence is given from the translational start codon to stop codon (which minus stop codon). The sentence in the manuscript has been modified to clarify this “For this, we first fused the TBR promoter to the TBR protein coding sequence (translation start codon to stop codon) from the T-allelic group accession (Eds-1) and the A-allelic group accession (Vår2-6) and then transformed these constructs into the tbr-3 mutant” in lines 194-197 and lines 439-441 in the method of plasmid construction.

M&M Plasmid construction (l. 369). Without reference to origin of vectors, certainly non-commercial ones, and reference to primers used, this cloning is hard to follow and even more difficult to reproduce. I assume primers in supp. data file 4 are used to amplify the *TBR* promoter and coding sequences, but this is not clear. Reference to *Agrobacterium* transformation method? It is not clear how the copy number of transgenics is determined.

We apologize for the confusion and have improved this now.

1) The vectors pDONR P4-P1r, pDONR 221, pDONR P2r-P3 and pB7m34GW are publicly available vectors and were shared by Chory lab from Salk Institute which we have now added in the acknowledgements.

2) To make it easy to follow, we have edited the method of plasmid construction “To evaluate function of TBR alleles, we generated the plasmid pTBR::TBR-HA-mCITRINE-HA. For this, the TBR promoter region (2437 bp fragment upstream of the TBR translational start codon) was PCR amplified using primers CP01 and CP02 from Eds-1 (T-allele), Vår2-6 and Col-0 (A-allele) genomic DNA. Gateway vector pDONR P4-P1r was amplified using primers CP03 and CP04, and then the linearized vector was recombined with TBR promoter via Gibson assembly cloning (NEB). Next, the protein coding sequence (from the translational start codon to stop codon, 1824 bp) of TBR was amplified using primers CP05 and CP06 from the cDNA of Eds-1, Vår2-6 and Col-0 and recombined with pDONR221. Finally, pDONR P4-P1r carrying the TBR promoter, pDONR 221 carrying TBR coding sequence and pDONR P2r-P3 carrying HA-mCITRINE-HA were recombined with Gateway destination vector pB7m34GW to obtain 3 constructs that contained the respective TBR promoter sequences and protein coding sequences. To generate pTBR::TBR_Col-0-HA-mCITRINE-HA (the TBR allelic promoter-fusion construct), we recombined pDONR P4-P1r carrying the TBR promoter sequence from two natural accessions (Eds-1 and Vår2-6), pDONR 221 carrying TBR protein coding sequence from Col-0, and pDONR P2r-P3 carrying HA-mCITRINE-HA with pB7m34GW vector. To generate pTBR::TBR^{L186R}:HA-mCITRINE-HA plasmid, a point mutation (G→T) was introduced into TBR by PCR amplified from pDONR221_Col-0 using primers CP05 and CP06, and then integrated the TBR promoter from Col-0 (in pDONR P4-P1) and HA-mCITRINE-HA (in pDONR P2r-P3) into pB7m34GW.” (Lines 434-451).

3) To make it easier to follow, we renamed the primers that were used to amplify TBR promoter, TBR coding sequence and TBR site-directed mutagenesis sequence in supplementary Data 6.

4) The reference for agrobacterium transformation method (Clough et al., Plant J, 1998) has been added as Reference No. 49 in the revised reference.

5) We have added the description for the determination of single insertion, homozygous T3 lines in the method of the revised manuscript (Lines 451-458).

Figure legends are not always clear.

Eg. Fig1c. What is meaning of blue and orange bars? When are roots sampled?

Fig 5. a. What is genetic background of T3 lines?

Supp fig 4. The legend to this figure is very short, for all panels.

Supp fig 5. What are *TBR-Col-0* plants? Description of genotype is lacking in legend.

These changes have been implemented in the revised manuscript according to the request of the reviewer.

1) We have added a color legend in Fig. 1c.

2) The root treatment and harvested time are described in the method section in the part of the quantitative real-time PCR in **lines 465-469**.

3) The genetic background of allelic transgenic complementation lines (TBR_Eds-1 and TBR_Vår2-6) in Fig. 5a was *tbr-3* mutant, which is described in **lines 194-197** in the revised manuscript.

4) We extended the figure legend of Supplementary Fig. 4 in Supplementary data.

5) TBR_Col-0 is one of the transgenic complementation of TBR in the *tbr-3* mutant. We have changed TBR_Col-0 to TBR_Col-0-9 in Fig.4c-d, Supplementary Fig. 6, 7 and 11 in the figures and figure legends.

Reviewer #2 (Remarks to the Author):

The authors conducted a GWAS analysis to identify natural alleles of *A. thaliana* that affect zinc tolerance of root growth. They report the identification of natural variants of the *FRD3* and *TBR* genes. The authors provide support for the involvement of both genes through Zn-sensitive phenotypes of loss-of-function mutants. The authors further suggest that the *TBR* polymorphism is associated with expression differences resulting from causal sequence variation in the promoter region. The authors provide data to suggest that in *tbr* mutants, a lower pectin content and higher degree of pectin methylesterification lead to a decreased ability to bind and sequester Zn in cell walls, when compared to the wild type. Finally, the authors report that genetic lesions in *TBR* orthologues of crop species, rice and *Lotus japonicas*, also lead to Zn hypersensitivity. Overall, the presented results appear to be largely sound (but see below) in general, and they may potentially extend existing knowledge on TBR protein functions to crop species (but see below). Yet, the major findings reported here are confirmatory, and they do not sufficiently advance the state-of-the-art concerning the biochemical or molecular mechanisms underlying the observed phenotypes associated with TBR. As the authors state in their manuscript, Pineau et al. (2012), for example, reported the role of *FRD3* in Zn tolerance. In addition, very importantly, Sinclair et al. (2017) reported the role of *TBR* in Zn and Ni tolerance upon cultivation of seedlings in the dark as well as in the light, which Sinclair et al. (2017) then associated with underlying changes in cell wall composition and metal-binding capability of the cell wall, as suggested again in this manuscript. Furthermore, Bischoff et al. (2010) also reported changes in pectin methylesterification in light-grown *tbr* seedlings.

We thank the reviewer for their comments and their helpful suggestions. However, we respectfully disagree with the assessment that our manuscript is mainly confirmatory. We have conducted the first GWAS for Zn tolerance and identified novel, common variants that underlie Zn tolerance in Arabidopsis. We show that expression differences conferred by these alleles lead to increased Zn tolerance (and now included additional work on the mechanism; see below). In addition, TBR's role in Zn tolerance is also conserved in two other species (monocot and dicot).

With regards to the previous work on TBR, we would like to note, that the work of Sinclair et al. (2017) was focused on photomorphogenesis and concluded that the dez mutation sensitized to high Zn. The dez mutant carries an intronic splice donor site G/A mutation that leads to the retention of the fourth intron in the mRNA (Figures S1A and S1B), thus resulting in an S/R amino acid exchange at position 491 of TBR, followed by several aberrant amino acids and a translational stop codon after position 496. While this is in line with our observations in the root, we clearly show that TBR can confer Zn tolerance, something that could not have been derived from the Sinclair et al or previous work, as loss of function or changes in protein phenotypes cannot be used to infer that an increase of gene product will lead to an increased phenotype. Moreover, there is nothing in this work that would

suggest anything about natural variation or the role of TBR (and which would be the orthologues) in other species.

However, given the reviewers' comments, we have expanded the manuscript. In this revised version of the manuscript, we have conducted and included additional analyses and experiments to investigate a potential function of the FRD3 alleles that we had identified and a deeper insight into the molecular mechanisms underlying the observed phenotypes associated with the TBR alleles.

We include an analysis on how the common alleles that we have found on FRD3, in addition to the variants that Pineau et al. had found, we also found a large number of associated SNPs in the FRD3 promoter (Supplementary Tab.4). We also included an analysis on SNPs and transcription factor binding sites. We found several SNPs in the transcription factor binding motifs (Supplementary Tab.5) in the revised manuscript.

For addressing the mechanisms by which TBR allelic variation modulates root tolerance to high Zn, we have now shown that TBR allelic variation in the TBR promoter leads to higher TBR expression (Fig. 6c). This in turn leads to higher protein levels of TBR (Fig.6d), which leads to a decrease in pectin methylesterification (Fig. 7c), which leads to more Zn bound in the root cell walls (Fig.7d). We can therefore trace the entire mechanism from the TBR allele through gene and protein expression to the effect on pectin methylesterification and Zn cell wall sequestration. To make this clear, we have added a model as Figure 8. We believe that our work allowed us to gain unprecedented insights, which we believe are not confirmatory but constitute a significant insight into the mechanisms by which TBR alleles facilitate higher Zn tolerance.

Additional major comments

1. GWAS for Zn tolerance in *A. thaliana* was reported before. In addition, ecologically, there is no indication so far that any *A. thaliana* populations colonize soils high in zinc. The observed differences may not be the result of natural selection but of the fortuitous fixation of weakly deleterious alleles in this selfing species.

*1) While there is a study that used recombinant inbred lines derived from a cross of two accessions to study Zn tolerance¹⁴. To our knowledge, there hasn't been a GWAS for Zn tolerance in *A. thaliana* reported.*

*2) We agree that it is not possible to conclude whether this is a result of natural selection for Zn tolerance, or similar stresses or a fortuitous fixation of alleles in this selfing species. However, as outlined in our previous response, we believe that our findings are novel and interesting even though we can't conclude much about the adaptive value of these alleles for *Arabidopsis*. We have addressed this now in the discussion (**Lines 347-349**).*

2. The reviewer is not convinced that the diagrams in Fig. 1c and 3c show sufficiently consistent differences in transcript levels between plants carrying Zn-sensitive and Zn-

tolerant alleles of *FRD3* and *TBR*, respectively. In particular, in Fig. 3c, there is no significant increase in transcript levels in two out of three Zn-tolerance allele-carrying genotypes.

1) To further confirm the results shown in Fig. 1c, we have quantified the *FRD3* gene expression in an additional set of accessions. We selected 3 C-allelic accessions (*TRÄ01*, *Vår2-6* and *Näs 2*) and 2 T-allelic accessions (*Spro 1* and *Bay-0* - Bay is a tolerant parent in Pineau et al.). These results were very clear and consistent with the pattern of *FRD3* gene expression in Fig. 1c, as the expression of *FRD3* showed a significant difference between the C-variant and T-variant accessions under high Zn conditions (Supplementary Fig. 3).

2) Thank you for pointing out this issue of Fig. 3c. Based on this feedback, we revised our results as follows:

“To identify the most likely candidate of these two genes in proximity to the significantly associated SNP, we first analyzed their gene expression. Publicly available organ and tissue specific microarray expression data (<https://bar.utoronto.ca/efp/cgi-bin/efpWeb.cgi>)³² showed that *TBR*, as well as *AT5G06710* are expressed throughout the plant (Supplementary Fig. 4c, d, f, g). Root single cell RNA sequencing (<https://phytozome-next.jgi.doe.gov/tools/scrna/>) showed that *TBR* is expressed in most root cells, while *AT5G06710* is expressed in a subset of root cell types only (Supplementary Fig. 4e, h). To then test whether the response of the candidate genes in the allelic groups of accessions that were identified by GWAS was consistent with the causal involvement of one of the two candidate genes, we selected representative accessions from the T-allelic group (*Eds-1*, *Eds-9* and *TRÄ 01*) and the A-allelic group (*Vår2-6*, *Sim-1* and *Fri 1*). Additionally, we also included *Col-0* with an intermediate root length, which was from A-allelic group. We found that the transcript level of *TBR* in *Sim-1* and *Fri 1* that are from the A-allelic accessions was higher than that of the T-allelic accessions, while *Col-0* and *Vår2-6* showed no significant expression difference compared with T-allelic accessions in high Zn conditions (Fig. 3c; Supplementary Fig. 4i). While *AT5G06710* was induced by high Zn in several accessions, there was no significant difference between the two allelic groups of accessions in high Zn conditions (Fig. 3c; Supplementary Fig. 4i). While the expression data of *TBR* was more consistent with it being the causal genes, the observation that *AT5G06710* was also induced by high Zn, prompted us to test mutant phenotypes for the two candidate genes.” in **lines 136-156**.

3. The protein family comprising *TBR* is rather large. Authors must include phylogeny of entire protein families of *A. thaliana*, soybean and rice in order to demonstrate the orthology between *AtTBR* and the genes of which functions are addressed here in soybean and rice. Alternatively, the authors must provide other properties or characteristics based on which they identified orthologues of *AtTBR* in rice and soybean

among all other members of the protein family.

Thank you for the suggestion. We have made the phylogeny of TBR protein families in 30 species and added the following test to the revised manuscript “To determine how well conserved TBR is across other plant species, we conducted BLAST searches using the TBR protein sequence. We found many putative orthologs in 30 plant species, including monocots like rice and other dicots such as Lotus japonicus (Supplementary Fig. 12a, b)” (Lines 298-302).

Minor comments

4. The authors should discuss their findings also in the context of possible morphological cell wall alterations in *A. thaliana*, soybean and rice. In *A. thaliana*, these were reported to be rather extensive, especially upon cultivation in high-zinc conditions.

We thank the reviewer for this suggestion. We have added the following test to the discussion “The cell wall is a preferential compartment for heavy metal accumulation. Modification of cell wall components can enhance heavy metal binding ability in root cell walls in many plant species⁴⁰⁻⁴³. Low-methylesterified pectins have been shown to bind divalent and trivalent metal cations and thereby contribute to forming a barrier limiting metal uptake and penetration into plant cells¹¹. Consistent with previous studies, our studies revealed that Zn exposure induced high levels of pectin de-methylesterification of root cell walls, which increased Zn binding capacity and Zn toxicity tolerance. It seems that this is a way that plants use to minimize the inhibition of root cell growth in high Zn conditions.” in the revised discussion part (Lines 359-367).

5. Methods are generally insufficiently detailed and the basic characterization of all mutant and transgenic lines (min.: target gene/allele transcript levels) under the conditions examined here should be provided (e.g. not provided for those shown in Fig. 5g).

We thank the reviewer for pointing out these issues.

1) We have included the basic characterization of all the mutants in the method of plant material and growth conditions (Lines 395-400). Moreover, we have described the gene structure of TBR and AT5G06710, and analyzed gene expression of the tbr mutants and at5g06710 in Supplementary Fig. 5.

2) To make it easier to follow, we have modified the method of plasmid construct, as well as the determination of single insertion, homozygous T3 lines in lines 434-458.

3) We have added TBR transcript level and protein level of TBR promoter allelic variants in Fig. 6c, d.

6. Lines: 28/29: “which had not been characterized associated with Zn toxicity tolerance”: not correct (see above)

We thank the reviewer for the suggestion. We have revised the sentence as “which was not known to be involved in determining root growth variation in high Zn conditions.” in the revised manuscript (Lines 28-29).

7. Line 86: TBR a “newly uncharacterized gene” – not correct

We thank the reviewer for the suggestion. The sentence has been changed to “Among them, a previously identified Zn tolerant gene, FRD3, and a gene, Trichome Birefringence (TBR) that was not known to be involved in determining root growth in high Zn conditions” in the revised manuscript (now lines 80-82).

8. Line 120: “TBR as a novel root growth regulator under high Zn levels”: not novel, not a regulator

We thank the reviewer for the improvement. We have modified the sentence “GWAS identify TBR as a novel locus to control root growth variation under high Zn levels” (now lines 119-120).

9. How do the authors reconcile the small visible differences between alleles in Fig. 5c/d with the large differences in protein levels according to the immunoblot show in Fig. 5e?

There might be multiple reasons for the visible difference. One reason might relate to the sensitivity of antibody staining and confocal microscope. Another reason might be differences in growth conditions: For Fig. 5c/d, we placed the seeds directly on control and high Zn (300 μ M) agar medium for 6 DAG and then calculated the fluorescence intensity. For the Western blot analysis, the seeds were grown on 1/2 MS medium for 5 DAG and then the seedlings were transferred to control and high Zn liquid medium for 12 h. However, since both independent measurements resulted in significant differences, we believe that our conclusions are warranted regardless of sensitivity differences between these independent approaches.

10. Also root lengths for low and high Zn should be shown for Fig. 5g (relative data do not correct statistically for variation under control conditions).

We thank the reviewer for this suggestion. We have modified the images using the root length data in control and high Zn conditions for site-mutagenesis lines and TBR-promoter allelic variants in Fig. 5f, g (now Fig. 6a, b). We also included relative root length value on the top of each box in high Zn in Fig. 6a, b.

11. The methodology provided for root Zn data (Fig. 6c/d) does not follow established procedures and is thus unlikely to reflect the Zn pools the authors intended to address here.

We believe this comment might have arisen due to our unclear description of the methods. We now have edited the method of Zn content measurement to avoid any confusion “To measure Zn accumulation in root cell walls and roots, seedlings of tbr mutants, their wildtype and allelic complemented lines were grown on control and high Zn conditions. Roots were harvested and rinsed with Milli-Q water, 10 mM CaCl₂ (5 min), 10 mM EDTA (5 min) and Milli-Q water, respectively.

*For Zn content in root cell walls, the cell wall precipitate was resuspended in 1 ml of 2 N HCl for 48 h with occasional shaking at room temperature. After centrifugation (13200 rpm, 10 min), the supernatant was diluted to a final volume of 5 ml with Milli-Q water⁵⁴; For Zn concentration in roots, the samples were dried in an oven at 85°C overnight. The dry samples were digested with 67% HNO₃ in a microwave oven using a temperature step gradient (120°C for 2 min, 160°C for 10 min and 190°C for 20 min) and then boiled in a heating block at 150°C. Samples were cooled to room temperature and subsequently diluted to a final volume of 10 ml with Milli-Q water. Zn concentration in roots was conducted by inductively coupled plasma mass spectrometry (Agilent 7800/7900ICP-MS). A blank sample and CRM rice, a certified reference sample (GBW10045a, Zn content is 12.4±1.2, ×10⁻⁶) were used to make the measurement precisely.” in the method part (**Lines 527-540**). The reference method for Zn content in root cell walls (Yang et al., Plant Physiology, 2011) has been added to the method and as Reference No.55 in the revised references.*

Reviewer #3 (Remarks to the Author):

Key results in the present study, the authors conducted a GWAS for Zn tolerance in *Arabidopsis thaliana*, through the measurement of primary root length. It leads to the identification of new genetic loci in addition to an already described one, *FRD3*. They focused on the *TBR* locus, giving genetical proof to the role of cell wall in Zn tolerance. Although the role of cell wall in Zn tolerance has already been identified, few molecular targets were identified in contrast to the numerous ones characterized for the other processes involved in Zn tolerance in plants.

Validity In general, data interpretations were clear and accurate to me. Nevertheless, few points need to be qualified or clarified:

We thank the reviewer for this assessment of our work and the thorough review.

- I wonder the interest of focusing so much on *FRD3* locus. For example, the figure 2 should be presented as additional data.

We thought that the FRD3 was a good proof of concept and illustrated the power of our GWAS approach to uncover genes that determine tolerance to Zn. Moreover, we have now included more comprehensive analysis on the additional FRD3 alleles and potential binding sites we have uncovered (Supplementary Data 4, 5). The chlorophyll content of frd3 mutants and their respective wildtypes in control and high Zn soil conditions have been added to Fig. 2c, d.

- I agree that the natural variation observed in the present study are distinct from the ones from Pineau et al. but the conclusion on causal natural variation in *FRD3* is not accurate. Both protein function and expression are responsible for natural variation (line 284-287). In addition, many studies identified and focused on allelic variation responsible for expression variation: Satbhai et al 2017; Baxter et al 2010; Julkowska, 2016; etc... (line 295).

1) Thank you for the correction. We revised this in the discussion part “For this, we identified several associated SNPs in the FRD3 promoter in our GWAS (Supplementary Fig. 13-15; Supplementary Data 4). Some of these SNPs were located in binding sites for transcription factors (Supplementary Data 5). For example, a SNP (2,570,777-2,570,792) was in the binding site of STOP1, STOP1 was previously shown to be necessary for AtMATE expression in aluminum tolerance, a member from MATE family³⁷. Consistent with the root growth response to high Zn between these two groups of accessions, the transcript level in the T-allelic group of accessions was significantly higher than in the C-allelic group accessions (Fig. 1c; Supplementary Fig. 3). Interestingly, the previously described variants that had been linked to Zn tolerance variation through experiments with the two accessions Bay-0 and Sha were not very prevalent in our GWAS mapping panel (the 27-bp deletion in the promoter region observed in Bay-0 was present in

21.1% of the accessions and the 28-bp deletion observed in Sha in the promoter region was in 12.6% of the accessions; the non-synonymous substitution N116S showed 10.7% allele frequency)¹⁴.” in the revised manuscript (**Lines 330-344**).

2) This sentence has been modified to “Previous studies have found instances in allelic variation that regulate transcription levels^{26, 28, 38} or alter proteins^{21, 39} play a critical role for trait variation. In line with this, we have found that the associated natural alleles of TBR confer distinct mRNA and protein expression levels that lead to decreased levels of a cell wall compound, which can bind Zn and sequester Zn in the root cell wall.” in line 295 (now **lines 350-355**).

Significance I agree with the proposed findings and their significance for the topic of Zn tolerance in plants. This study identifies a key molecular player in the role of the root cell wall in Zn tolerance. Data and methodology I found the process for validating the candidate gene appropriate and quality of data and presentations suitable expect for:

o figure 1c: the color legend is missing

Thanks for pointing this out. The color legends have been added to Fig. 1c.

o figure 1 legend: tolerant accessions instead of tolerance

Thank you for the correction. It has been modified.

o figure 3C: the higher expression level of TBR in A-allelic accessions is only valid for one accession (Sim-1) and only 2 values exist for Vår2-6 expression. This should be improved.

Thank you for pointing this out. Based on this feedback, we revised this section in the results as follows:

“To identify the most likely candidate of these two genes in proximity to the significantly associated SNP, we first analyzed their gene expression. Publicly available organ and tissue specific microarray expression data (<https://bar.utoronto.ca/efp/cgi-bin/efpWeb.cgi>)³² showed that TBR, as well as AT5G06710 are expressed throughout the plant (Supplementary Fig. 4c, d, f, g). Root single cell RNA sequencing (<https://phytozome-next.jgi.doe.gov/tools/scrna/>) showed that TBR is expressed in most root cells, while AT5G06710 is expressed in a subset of root cell types only (Supplementary Fig. 4e, h). To then test whether the response of the candidate genes in the allelic groups of accessions that were identified by GWAS was consistent with the causal involvement of one of the two candidate genes, we selected representative accessions from the T-allelic group (Eds-1, Eds-9 and TRÅ 01) and the A-allelic group (Vår2-6, Sim-1 and Fri 1). Additionally, we also included Col-0 with an intermediate root length, which was from A-allelic group. We found that the transcript level of TBR in Sim-1 and Fri 1 that are from the A-allelic accessions was higher than that of T-allelic accessions,

while Col-0 and Vår2-6 showed no significant expression difference compared with T-allelic accessions in high Zn conditions (Fig. 3c; Supplementary Fig. 4i). While AT5G06710 was induced by high Zn in several accessions, there was no significant difference between the two allelic groups of accessions in high Zn conditions (Fig. 3c; Supplementary Fig. 4i). While the expression data of TBR was more consistent with it being the causal genes, the observation that AT5G06710 was also induced by high Zn, prompted us to test mutant phenotypes for the two candidate genes.” in **lines 136-156**.

2) The third sample of Vår2-6 was destroyed when I extracted the RNA. However, I repeated this experiment for several times, Vår2-6 showed similar results in the TBR and AT5G06710 gene expression in different repeats. Furthermore, I added the gene expression results of the third accession in these two allelic variants. The results were very clear and consistent with the pattern of TBR gene expression in Fig. 3c, as the TBR expression showed a significant difference between the T-variant (TRÄ 01) and A-variant (Fri 1) accession under high Zn conditions (Supplementary Fig. 4i).

o figure 4d: the y axis is incomplete and does not correspond to values in the Excel file provided. In addition, inadequate unit is mentioned in Excel file.

Thank you very much for pointing out this issue. The y-axis has been revised to root elongation zone length. The data has been corrected in the Excel file of Fig. 4d in Source Data and the unit has been modified.

o in figure 4: data on root growth for Col-0/*tbr-3*/TBR_Col-0 are missing to compare with root elongation zone length.

1) Thank you for pointing out this issue. To address this, we added root growth data of four complemented lines of TBR_Col-0 in Fig. 4e and revised the description as “We next tested whether these phenotypes could be rescued by transgenic complementation of TBR in the *tbr-3* mutant (TBR_Col-0, the Col-0 TBR coding sequence driven by its own promoter from Col-0). Seedlings of four independent T3, single insertion homozygous complementation lines showed the recovery of the root growth (Fig. 4e). We then measured the length of root elongation zone in high Zn conditions in one of these transgenic lines and found that consistent with the restoration of root growth under high Zn, the length of root elongation zone was also restored (Fig. 4c, d).” in the text (**Lines 181-188**).

2) TBR_Col-0 is one of the transgenic complementation of TBR in the *tbr-3* mutant. We have changed TBR_Col-0 to TBR_Col-0-9 in Fig.4c, d, Supplementary Fig. 6, 7, 11 and in the figure legends.

o figure 6c, d: there are only two values for Zn contents in *trb-3* and TBR_Col-0

Thank you for pointing out the issue. We added the third value of Zn contents in

tbr-3 and TBR_Col-0-9 in Fig. 6 c, d (now Supplementary Fig. 11c, d).

- o figures 2b, 4b: It would be easier to read if relative root growth was represented as done for GWAS and in figure 5 f, g

We added the relative root length value on the top of each box in Fig. 2b, Fig. 4b, e, Fig.5a and Fig. 6a, b.

- o In methods, the information on plants for q-RT-PCR experiment is missing.

Thank you for pointing out the issue. We added the information on plants in the method section of quantitative real time PCR as “For qRT-PCR analysis, the seedlings were grown directly on 1/2 MS agar medium for 5 DAG and then transferred to control and high Zn (300 μ M) liquid medium for 12 h, except for the TBR and AT5G06710 gene expression in extreme accessions. For TBR and AT5G06710 gene expression, the seedlings were transferred to control and high Zn (300 μ M) agar medium for 1 d after growing on 1/2 MS agar medium for 5 DAG” (Lines 465-469).

Suggested improvements the manuscript could be improved with:

- Identification of motive responsible for expression variation in *TBR* promoter

Thank you for the suggestion. We have included an analysis of potential binding sites analysis of TBR promoter and found a SNP variation in the binding motif of AtHB32 according to the Cistrome database.

- Search for *TBR* expression in the Zn hyper tolerant relative species *Arabidopsis halleri* or *Noccaea caerulescens* as it is known that genes involved in Zn tolerance were constitutively and highly expressed in *N. caerulescens*.

Thank you for the comment. We haven't found and tested TBR expression in Arabidopsis halleri or Noccaea caerulescens. But we found two homologous genes of TBR (Ah6G07070 and Ah3G13430, which showed 93% and 66% identity, respectively) in Arabidopsis halleri (Supplementary Fig. 12a). The homologous in Arabidopsis halleri may involve in Zn tolerance as AtTBR and its homologous in rice and Lotus japonicus have clarified TBR function in response to Zn toxicity tolerance.

- Localization and quantification of Zn in root cell wall in extreme accessions

We agree with the reviewer that quantifying Zn content in the root cell walls in extreme accessions is convincing evidence to test our hypotheses. We have quantified Zn content in root cell walls of TBR allelic complementary lines and TBR

promoter allelic variants. The T-allelic variant contained lower amounts of Zn in root cell walls than A-allelic variant in high Zn conditions (Fig. 7b, d). These results could explain TBR allele function in Zn sequestration. The results have been described in the text of the revised manuscript (Lines 275-287).

- Precisions on nature and localization of SNP on the gene *FRD3* on supplementary figure 3 and table S2.

1) We made a mistake while copying the data of associated SNPs on the FRD3 gene. It has been corrected and added to Supplementary Data 4.

2) There were fifty-one associated SNPs located throughout the upstream, coding and downstream regions of FRD3. Localizing all these fifty-one SNPs on Supplementary Fig. 3 made the image more unclear (now Supplementary Fig. 13). So we only localized the SNPs position in Supplementary Tab. 2 (now Supplementary Data 4). We also added the reference allele and alternative allele of these SNPs in the Supplementary Data 4.

Clarity and context A better description of mechanisms involved in Zn binding to cell wall components and of known Zn tolerance processes would render the manuscript clearer. For example, the use of reviews on Zn tolerance would be useful to give better image on the different processes involved in Zn tolerance, and on the presence of hyperaccumulating plants (line 47-51). For example, Tang et al 2022; Lin & Arts 2012.

*Thank you for the suggestion. We have summarized the mechanisms of known Zn tolerance processes and involved in heavy metal binding to cell walls and added these excellent references “Plants have developed tolerance mechanisms to keep Zn homeostasis when exposed to excess concentration of Zn in the environment. These include control of uptake, intracellular binding to Zn chelators, efflux from the cell, sequestration into the vacuole, and cross-homeostasis between Iron (Fe) and Zn^{1, 7, 8}. Root cell walls, the first interface between soil metals and plant cells, are another potential location to sequester metals^{9, 10}. The modification of cell wall components enhances the ability of the apoplast to sequester excess heavy metals and that can contribute to prevent heavy metals from entering the plant and thereby improve plant tolerance to heavy metal toxicity in the soil^{1, 11}. The impact of such cell wall modifications is apparent in *ozs2* mutants that are defect in the gene encoding pectin methylesterase 3 (pectin methylesterase 3, PME3) and that showed Zn hypersensitivity¹².” in the introduction part (Lines 47-57).*

Reviewers' Comments:

Reviewer #1:

Remarks to the Author:

The authors have addressed nearly all my previous comments very well, and improved the manuscript considerably. I still have a few minor issues remaining, which the authors could consider to address.

- L 108, it seems to me that the reference to supplementary figure 3 should be made at the end of the sentence ending in L. 106: "... SNP (Chr3:2574560)."

- regarding the description of *tbr-3* and the mutant of At5g06710, the following needs to be considered. Both these mutants are T-DNA insertion lines. Normally, a T-DNA insertion in an exon, as is the case here, will make a non-functional protein due to loss of part of the transcript or the generation of chimeric transcripts which cause frameshifts in the ORF. Very rarely are T-DNA insertions in exons removed by splicing. If transcripts are found, they are often either from 5' or 3' of the T-DNA insert, not spanning it. It is not clear to me what is shown in sup fig 5b, but I expect detection of transcript from either 5' or 3' of the T-DNA insert for At5g06710 and 3' of the insert for *tbr-3*, but not spanning the insert. Such should be made clear.

Furthermore, the note should be added that these two alleles are likely to be loss-of-function alleles due to the position of the T-DNA in an exon. However, since the insertion on *tbr-3* is at the very 5' end of the protein coding region, and there is transcript found (probably from 3' of the insert), there is a very small probability that the resulting N-terminally truncated protein may have some function. However, I doubt it. Better will be to assume the insertion leads to a non-functional protein, even though some downstream transcript is found.

So, it seems unlikely that the reduced expression causes the *tbr-3* phenotype. Frankly, I expect *tbr-3* to be a full loss-of-function mutant, based on the position of the T-DNA. The *tbr-1* allele may very well not represent a loss-of-function allele, but a gain-of-function allele, albeit with a negative function, providing the stronger phenotype. After all, the *tbr-1* allele is transcribed and a full-length mutant protein is likely to be made, which is much less likely for the *tbr-3* allele. I suggest to alter the reasoning listed in lines 172-175, referring to the assumption that the *tbr-3* allele is likely to be a loss-of-function, while this is not clear for the *tbr-1* allele.

Finally, best not refer to 'heavy metals', as there currently is no proper definition of such. Best refer to 'metals' or 'trace elements'.

NB. some of the revised parts of the manuscript need a careful check for English.

Reviewer #2:

Remarks to the Author:

In a GWAS screen for hypersensitivity to excess Zn, the authors identify 21 genes (about 14 peaks), with the strongest peak on chromosome 4 that is not referred to further. The authors relate two of the further peaks to (groups of) putative causal variants at the *FRD3* and *TBR* locus of *Arabidopsis thaliana*. After a brief characterization of *FRD3* transcript levels and of Zn tolerance of several *frd3* mutants, the authors focus primarily on *TBR*. The *TBR* protein characterized in this publication is a putative polysaccharide O-acetyltransferase with a GDSL/SGNHH-like acyl/esterase family domain found in *PMR5* and *Cas1* proteins. It is also annotated as *POAT1* (PECTIN O-ACRETYLTRANSFERASE 1; Chiniquy et al., 2019; Zhong et al., 2023), based on the capability of the *TBR* protein of transferring O-acetyl groups from acetyl-CoA onto oligogalacturonides and the pectin component homogalacturonan. Natural *TBR* alleles of *Arabidopsis* are then functionally characterized in transgenic lines in this manuscript, relating increased promoter strength to elevated Zn tolerance. The authors

should detail more on previously characterized functions of TBR, especially in Zn tolerance and cell wall modification, as well as previously described Arabidopsis proteins contributing to basal Zn tolerance. Although not identified based on natural variation, these previously published results are highly relevant for this study. The reviewer has serious concerns in relation to one of the methods used, and has some other methodological issues. Sometimes the text does not match well with the data shown in the diagrams.

1. Based on my earlier request (11.), the authors have now added in the methods how they quantified cell wall-bound Zn (lines 527-533). They homogenized frozen plant material, then extracted the residue in 75% ethanol to subsequently isolate a cell wall fraction, from which they solubilized Zn using HCL, followed by the quantification of Zn in the supernatant. However, after homogenization of frozen tissues and thawing, or thawing in a liquid medium (75% ethanol), which destroys cells and mixes the content of internal and external sites, Zn that was localized in vacuoles as a major zinc storage pool, for example, is able to re-localize to cell wall components, and Zn bound only loosely to cell wall components can go into solution (also inside vacuoles or the cytosol). This method, although published, is entirely flawed and not at all state-of-the art in the plant metals field. To address cell-wall-bound Zn, non-destructive methods or the imaging of shock-frozen unthawed tissues must be used, as was done on wild-type and *tbr* roots grown under control and high-Zn conditions in Sinclair et al. (2017, shown in Fig. S4E-H). The reviewer thus has major doubts on the results shown in Fig. 7b and d, and Suppl. Fig. 11c). These results should be removed from the manuscript, and the authors should instead refer to Sinclair et al. (2017). The description of these results in lines 276 to 290, as well as discussion lines 353-355 and lines 367-369 must be changed/deleted accordingly.

2. The reviewer disagrees with the response of the authors to the reviewer's main comment. It seems that I did not make myself clear enough earlier (see Reviewer #2 earlier initial major comment, and subsequent comments 6, 7, and 8, and the authors' responses to these). Sinclair et al. (2017) showed and described in the text explicitly that root growth of the Arabidopsis thaliana partial loss-of-function mutants at the TBR locus (named *tbr* or *dez*) are hypersensitive to excess Zn, and they subsequently moved on to also examine the cell wall alterations in the mutant and its photomorphogenic phenotype. They showed multiple lines of evidence that both Zn exposure and the loss of TBR function lead to increased pectin methylesterification of cell walls and decreased amounts of Zn bound on cell walls, among other major cell wall and developmental alterations described in Sinclair et al. (2017). In more detail, the results text of Sinclair et al. (2017) begins: "In a genetic screen for Zn-hypersensitive mutants, we identified an EMS mutant ... (this is identified as a partial loss-of-function mutant of the TBR gene later in the same manuscript). Later in the results text of Sinclair et al. (2017): "The *dez* (= *tbr*) mutant was hypersensitive to Zn in the light, with clear inhibition of root elongation, but not of hypocotyl elongation, when compared to the wild-type (Figures S1D and S1F)." ("in the light" referring to ordinary day-night growth conditions in this context, as a common growth condition employed in Sinclair et al. (2017) was continuous darkness). Sinclair et al. (2017) also showed that Zn tolerance was restored in a complemented line (*dez* transformed with a genomic TBR construct). Supplemental Figures S1I and S1J of Sinclair et al. (2017) also showed hypersensitivity of root growth of *dez* = *tbr* to the other heavy metals Ni and Cd in the light. Sinclair et al. (2017) included a detailed characterization of the general Zn hypersensitivity of growth and development in *tbr* mutants, including the photomorphogenic phenotype of *dez* = *tbr*, with a shortened hypocotyl, and the exacerbation of the phenotype specifically under excess Zn conditions. Sinclair et al. (2017) demonstrated that the degree of pectin methylesterification is enhanced in the *dez* = *tbr* mutant, decreased pectin acetylation as well as decreased Zn binding to cell walls of the *tbr* mutant (Fig. 4, Fig. S4), similar to, but going beyond, some results reported in this manuscript (e.g. Fig. 7). The authors must make changes to their manuscript text to accurately reflect this previously published work. In detail, this entails:

(a) Abstract lines 28-29 (my former comment 6, and the authors' response): "...which was not known to be involved in determining root growth variation in high Zn conditions." This sentence is still wrong, or at least highly misleading, in the revision. The authors should insert here: "as reported by Sinclair et al. (2017) in mutants carrying a loss of function allele at the TBR locus (i.e., *tbr-1*, *dez*).

(b) Line 57: In addition to mentioning the previous publication on *ozs2* as a Zn-sensitive mutant altered in cell wall pectin, the authors must mention here that Sinclair et al (2017) reported that root elongation of *Arabidopsis* EMS mutants with partial loss of TBR function was hypersensitive to excess Zn and associated with decreased pectin acetylation, increased pectin methylesterification and decreased cell wall binding of Zn²⁺, by inserting a sentence in line 57, before the beginning of the sentence "While...".

(c) Line 81 (my former comment 7, and the authors' response): The authors write "... a gene, Trichome Birefringence (TBR) that was not known to be involved in determining root growth in high Zn conditions." This is not correct as mentioned above. The authors must mention also here that a *tbr* mutant was previously identified in a forward genetic screen for *Arabidopsis* mutants hypersensitive to excess Zn, and that its root growth was found to be hypersensitive to Zn (Sinclair et al., 2017).

(d) Line 119/120 (my former comment 8, and the authors' response): Title of this section again is not correct or at least misleading, given that partial TBR loss-of-function was published to cause decreased root growth in excess Zn conditions in Sinclair et al. (2017). This should be mentioned also here. Then, the authors can add that Sinclair et al. (2017) addressed artificially induced mutations and not a naturally occurring mutation as identified in this manuscript. In this way, if these two pieces of information are added, the text of this manuscript will no longer be misleading.

(e) Line 134/135: The authors write "Interestingly, TBR had been previously identified via an EMS mutant screening to regulate photomorphogenesis in high Zn conditions." The authors should modify this statement (as explained above):

"Interestingly, TBR had been previously identified via an EMS mutant screen conducted to identify genes required for basal tolerance to excess zinc *Arabidopsis* (31/Sinclair et al., 2017). Sinclair et al. (2017) reported additionally that TBR is necessary for the suppression of photomorphogenesis in the dark, with an exacerbated phenotype of the mutant in high-zinc conditions." (Photomorphogenesis in the dark was observed under both control and high-zinc conditions, but was stronger in high zinc).

3. My next major reservation relates to my previous comment no. 9 is on the designation and analysis of the TBR "promoter" as comprising 2437 bp upstream of the TBR translational start codon (line 234, line 435). While the reviewer appreciates the attempt of the authors to functionally validate candidate causal polymorphism, there are fundamental limitations to the data provided in the manuscript.

(a) The authors must discriminate between the segment that constitutes the promoter region (upstream of the transcriptional start site) and the contained segment that constitutes the 5'-UTR (downstream of the transcriptional start site and until the translational start codon). The promoter region can affect the rates of transcription and thus transcript levels, but it never affects protein levels to a substantially larger degree than it affects transcript levels. In this manuscript, however, protein levels appear to be affected to a much larger extent than transcript levels by the "promoter region" of A-allele individuals (Fig. 5e, compare Fig. 6c and 6d) – a finding of this manuscript that is even shown in the summarizing model (Fig. 8A) – so they do not see it as a consequence of differing experimental conditions or experimental sensitivities as suggested by their response. Since the authors show that none of the polymorphisms in the TBR coding sequence affect protein levels, the over-proportionately large protein levels shown in lines transformed with a TBR driven by an A-allele promoter in Fig. 6, for example, are either an artefact from making an HA-mCITRINE-HA fusion protein or from not including introns in the construct, or they are caused by upstream sequence that is not part of the promoter but instead part of the 5'-UTR. It would be state-of-the-art to address the effects of candidate causal polymorphisms experimentally in a manner that discriminates between the promoter region and the 5'-UTR instead of addressing both together as the promoter. Thus, the conclusion in lines 247 to 250 does not seem correct to the reviewer.

(b) To support the hypothesis that the causal polymorphisms in the promoter region lead to variation in transcript levels, the authors generate *tbr-3* lines transformed with TBR coding regions under the control of an allele A (Zn-tolerant) *Var2-6* and a T-allele (Zn-sensitive) *Eds-1* upstream region. However, gene expression can vary substantially between independently transformed lines. The way the authors selected the lines that they characterized (and the low number of lines worked with) may have led to mis-conclusions concerning relative promoter strengths (line 454-459): (i) Even if only a single T-DNA locus segregates, tandem insertions of two or more T-DNA copies are extremely

common. (ii) The authors chose lines with a desired fluorescence signal of the TBR-mCITRINE fusion protein. So in essence, what the authors do show here is that Zn tolerance in a *tbr* mutant background associate not only with intact TBR transcript levels (which was known before, Sinclair et al., 2017) but also with the amount of TBR protein produced, irrespective of the cause of differences in protein levels (Fig. 6; the phenotypic data in Fig. 5a is far less clear-cut based on statistics, line 201, where only part of the Var2-6 TBR promoter-containing lines are statistically significantly more Zn-tolerant than only part of the Eds-1 TBR promoter-containing lines) (this relates to my earlier comment 5).

4. Lines 112/113: The authors should alter the description of the results so that the text matches better with the diagrams. "Compared with wildtype, *frd3* mutants showed a significant reduction of root growth and severe chlorosis in response to high Zn (Fig. 2)". This statement is in disagreement with the data provided in Fig. 2b and d. *frd3-7* has significantly shorter roots and a trend for less chlorophyll than the corresponding wild type (Col-0) already under control conditions. Under high Zn conditions, the differences between *frd3-7* and the wild type are not increased for root length and chlorophyll. In addition, there is no significant difference between *frd3-3* and the corresponding wild type *gl-1* for root length at high Zn (Fig. 2c). In *frd3-3* chlorophyll is a bit down, but not statistically significantly in high Zn in *frd3-3*. However chlorophyll in *gl-1* is significantly down in high Zn compared to control in *gl-1*. Whether *gl-1* is the wild type corresponding to *frd3-3* is a bit unclear to me. The authors state this in their methods, but Rogers et al. (2004) write that *frd3-3* is *man1*, and Delhaize et al. (1996) who published *man1* write that *frd3-3* is in the Col-0 background. In any case, the data shown in this manuscript do not unequivocally support Zn hypersensitivity in *frd3* mutants, different from what is written in the manuscript text.

5. Lines 148-151, referring back to my previous comment 2: The authors state that A allele accessions had higher TBR transcript levels than T allele accessions. However, according to the shown relative transcript levels of TBR, there was no significant difference between accessions in control conditions. Moreover, there was no significant difference between two of the A allele accessions and the T-allele accessions in high Zn, either. Only transcript levels of the third A allele accession, Sim-1 were slightly higher (1.17 to 1.3-fold) than in the T-allele accessions under high-Zn conditions. Quantitatively, transcript level differences were minor and not comfortably within the detection range of differences by RT-qPCR (for such extremely small transcript level differences, ddPCR should be used to obtain reliable results).

Other issues:

6. Line 73 and following: The authors state "So far, only Ferric Reductase Defective 3 (FRD3) has been identified to be underlying natural variation of the root growth responses to Zn toxicity tolerance in *Arabidopsis thaliana*, by regulating Fe and Zn translocation from the root to the shoot". The reviewer feels that, although this is strictly speaking correct, the scope of this statement is artificially narrow and thus misrepresents the state of knowledge.

(a) Based on a number of mutants and publications, it is clear that MTP1, MTP3, HMA3, HMA2 and HMA4 and the corresponding protein functions, for example, have central roles in basal Zn tolerance of *A. thaliana*.

(b) Studies of *Arabidopsis*-related Zn-hypertolerant hyperaccumulator species (*A. halleri*, *N. caerulescens*) that exhibit extreme Zn tolerance and the alleles therein (often affecting the expression of Zn tolerance genes via cis-regulatory alterations) have provided major insights, with roles for HMA4, MTP1, and several other genes. This work should be mentioned either in the introduction or in the discussion.

(c) Candidate loci at a major peak on chromosome 4 identified in this submission should at least be mentioned in the results or discussion. Is this peak in the vicinity of HMA2, HMA3, HMA1, IRT1 (all on chromosome 4 and previously shown to affect Zn tolerance).

7. Fig. 7 title: The predicted function of TBR makes it extremely unlikely that this protein mediates pectin demethylesterification. According to a number of previous publications, the activity of TBR is highly likely to primarily affect pectin O-acetylation, and the effect on pectin methylesterification is

only a secondary effect caused by the highly homeostatic character of cell wall metabolism.

8. Line 261: "We found that the roots of *tbr* mutants showed lower pectin content compared to wildtype in high Zn conditions. "  refer here to Figure where this is shown

9. Lines 357 and following: Here the authors should write about the known relationship of TBR with pectin O-acetylation (Sinclair et al., 2017, Chiniqy et al., 2019; Zhong et al., 2023); in the present form this section in the discussion is easily misread.

10. Line 361: Sinclair et al (2017), who showed this very specifically for Zn and *tbr* mutants compared to the wild type, should be cited here, too.

11. Line 364: Change to "Zn exposure induced an increase in.."

12. Line 366: Discuss here also Sinclair et al (2017), who observed an opposing change in pectin methylesterification under exposure to excess Zn in the wild type.

13. The authors discuss roles exclusively for FRD3 and TBR in natural variation of Zn tolerance in *A. thaliana*, but a major peak was observed on chromosome 4, for which no gene is mentioned here, and TBR and FRD3 are only a set of 2 out of 21 loci the authors detected in this GWAS analysis to affect Zn tolerance. The discussion in lines 371-375 should thus be modified.

14. The discussion should mention the role of TBR in light-dependent development and the influence of Zn on it. Even if not analysed here, it is likely affect the natural accessions of *A. thaliana* studied here.

15. Referring back to my previous comment 3: As the TBR family has 46 members in *A. thaliana* (and 7 in the TBR-related group), I had in mind a more extensive phylogenetic tree covering at least the full TBL groups (all members) of Arabidopsis, rice and *Lotus japonicas* so that the reader can assess orthology.

REVIEWER COMMENTS

Reviewer #1 (Remarks to the Author):

The authors have addressed nearly all my previous comments very well, and improved the manuscript considerably. I still have a few minor issues remaining, which the authors could consider to address.

- L 108, it seems to me that the reference to supplementary figure 3 should be made at the end of the sentence ending in L. 106: "... SNP (Chr3:2574560)."

*Thank you for suggesting the improvement. The reference to supplementary figure 3 has been moved from line 108 to line 106 (now **line 117**).*

- regarding the description of tbr-3 and the mutant of At5g06710, the following needs to be considered. Both these mutants are T-DNA insertion lines. Normally, a T-DNA insertion in an exon, as is the case here, will make a non-functional protein due to loss of part of the transcript or the generation of chimeric transcripts which cause frameshifts in the ORF. Very rarely are T-DNA insertions in exons removed by splicing. If transcripts are found, they are often either from 5' or 3' of the T-DNA insert, not spanning it. It is not clear to me what is shown in sup fig 5b, but I expect detection of transcript from either 5' or 3' of the T-DNA insert for At5g06710 and 3' of the insert for tbr-3, but not spanning the insert. Such should be made clear.

Furthermore, the note should be added that these two alleles are likely to be loss-of-function alleles due to the position of the T-DNA in an exon. However, since the insertion on tbr-3 is at the very 5' end of the protein coding region, and there is transcript found (probably from 3' of the insert), there is a very small probability that the resulting N-terminally truncated protein may have some function. However, I doubt it. Better will be to assume the insertion leads to a non-functional protein, even though some downstream transcript is found.

So, it seems unlikely that the reduced expression causes the tbr-3 phenotype. Frankly, I expect tbr-3 to be a full loss-of-function mutant, based on the position of the T-DNA. The tbr-1 allele may very well not represent a loss-of-function allele, but a gain-of-function allele, albeit with a negative function, providing the stronger phenotype. After all, the tbr-1 allele is transcribed and a full-length mutant protein is likely to be made, which is much less likely for the tbr-3 allele. I suggest to alter the reasoning listed in lines 172-175, referring to the assumption that the tbr-3 allele is likely to be a loss-of-function, while this is not clear for the tbr-1 allele.

We thank the reviewer for this feedback. We have incorporated the suggestions in the text and altered Supplementary Fig.6b accordingly).

1) "We therefore obtained mutants for the two candidate genes (Supplementary Fig. 6). For TBR, we obtained two mutants: the tbr-1 mutant in which TBR function is strongly impaired due to a G to A mutation in the 3rd exon of the coding region ,

*resulting a predicted amino-acid change of Gly to Glu at position 427³³, as well as the T-DNA line SALK_058509C, which hereafter was called tbr-3 and which is most likely a loss of function allele as it contains a T-DNA insertion in the coding sequence close to the 5'-UTR that most likely leads to no protein, or a truncated, or a non-functional protein. For AT5G06710, we obtained the T-DNA insertion line SALKseq_062866 that has an insertion in the 3rd exon and therefore also most likely constitutes a loss of function mutant. In addition to confirmation of the T-DNA insertion by genotyping, we also measured transcript levels at 3' downstream of the T-DNA insertions, which frequently led to alterations in the level of transcript. We found that in the cases of both T-DNA insertion mutants transcript was present at a much lower level than in wildtype (Supplementary Fig. 6b)” in **lines 178-191**. 2) “For this and the subsequent analyses, we utilized the tbr-3 loss of function mutant, while the tbr-1 mutation has a negative effect but it is not clear whether it is a simple loss of function mutant” in **lines 204-206**.*

Finally, best not refer to 'heavy metals', as there currently is no proper definition of such. Best refer to 'metals' or 'trace elements'.

We thank the reviewer for pointing this out. It has been modified as metals accordingly.

NB. some of the revised parts of the manuscript need a careful check for English.

Thank you for the suggestion. We have carefully improved the language in the revised manuscript. These edits are shown with track changes.

Reviewer #2 (Remarks to the Author):

In a GWAS screen for hypersensitivity to excess Zn, the authors identify 21 genes (about 14 peaks), with the strongest peak on chromosome 4 that is not referred to further. The authors relate two of the further peaks to (groups of) putative causal variants at the FRD3 and TBR locus of Arabidopsis thaliana. After a brief characterization of FRD3 transcript levels and of Zn tolerance of several frd3 mutants, the authors focus primarily on TBR. the TBR protein characterized in this publication is a putative polysaccharide O-acetyltransferase with a GDSL/SGNHH-like acyl/esterase family domain found in PMR5 and Cas1 proteins. It is also annotated as POAT1 (PECTIN O-ACRETYLTRANSFERASE 1; Chiniquy et al., 2019; Zhong et al., 2023), based on the capability of the TBR protein of transferring O-acetyl groups from acetyl-CoA onto oligogalacturonides and the pectin component homogalacturonan. Natural TBR alleles of Arabidopsis are then functionally characterized in transgenic lines in this manuscript, relating increased promoter strength to elevated Zn tolerance. The authors should detail more on previously characterized functions of TBR, especially in Zn tolerance and cell wall modification, as well as previously described Arabidopsis proteins contributing to

basal Zn tolerance. Although not identified based on natural variation, these previously published results are highly relevant for this study. The reviewer has serious concerns in relation to one of the methods used, and has some other methodological issues. Sometimes the text does not match well with the data shown in the diagrams.

We thank the reviewer for the thorough review of our manuscript. We have addressed the concerns as outlined below, including the suggestion to include more detail previously characterized functions of TBR and address the expressed methodological concerns as outlined below.

1. Based on my earlier request (11.), the authors have now added in the methods how they quantified cell wall-bound Zn (lines 527-533). They homogenized frozen plant material, then extracted the residue in 75% ethanol to subsequently isolate a cell wall fraction, from which they solubilized Zn using HCL, followed by the quantification of Zn in the supernatant. However, after homogenization of frozen tissues and thawing, or thawing in a liquid medium (75% ethanol), which destroys cells and mixes the content of internal and external sites, Zn that was localized in vacuoles as a major zinc storage pool, for example, is able to re-localize to cell wall components, and Zn bound only loosely to cell wall components can go into solution (also inside vacuoles or the cytosol). This method, although published, is entirely flawed and not at all state-of-the art in the plant metals field.

To address cell-wall-bound Zn, non-destructive methods or the imaging of shock-frozen unthawed tissues must be used, as was done on wild-type and tbr roots grown under control and high-Zn conditions in Sinclair et al. (2017, shown in Fig. S4E-H). The reviewer thus has major doubts on the results shown in Fig. 7b and d, and Suppl. Fig. 11c). These results should be removed from the manuscript, and the authors should instead refer to Sinclair et al. (2017). The description of these results in lines 276 to 290, as well as discussion lines 353-355 and lines 367-369 must be changed/deleted accordingly.

We thank the reviewer for this important feedback and suggestions. We have taken this into account, removed these data from the main figures and pointed out the caveats of this commonly used methods very clearly (see below). We did not completely remove it (but pointing out the caveats of this method) as in Sinclair et al, Zn content in etiolated seedlings was measured (Fig. S4E), and Zn accumulation in hypocotyl cell wall was mapped by μ -XRF (Fig. S4F), not in roots. Because the Arabidopsis roots are softer and thinner than hypocotyls, μ -XRF may be still not sufficient to map Zn accumulation in cell walls in Arabidopsis roots. While having major limitations, the method we have utilized has been commonly used to extract the cell wall and then measure aluminum, phosphorus and cadmium in root cell wall (Gao et al., 2023, Journal of Integrative Plant Biology; Shi et al., 2015, Journal of Integrative Plant Biology; Zhu et al., 2015, Journal of Experimental Botany; Zhu et al., 2014, Plant Physiology; Zhu et al., 2012, Plant Cell). We hope that our current presentation is acceptable to the reviewer:

L301 ff: *“It had been previously shown that an impairment of TBR function in the tbr mutant leads to a lower capacity to retain divalent Zn²⁺ cations in the cell walls of etiolated seedlings compared to the wild-type in high Zn¹³. To corroborate this further in root tissues grown under our treatment conditions, we extracted root crude cell walls according to the method of Zhong and Läuchli³⁴ and quantified Zn concentration by inductively coupled plasma mass spectrometry. According to these measurements, tbr mutant roots contained lower amounts of Zn than wildtype roots in high Zn conditions, and Zn levels were rescued to almost wildtype levels through complementation with a functional TBR allele (Supplementary Fig. 12c, d). We would like to note that in our procedure, in contrast to the superior Micro-X-ray fluorescence spectroscopy method used by Sinclair et al¹³ for measuring Zn content in hypocotyl cell walls, due to the homogenization of frozen tissues and subsequent thawing during our procedure, Zn that is localized inside the cell might be able to re-localize to cell wall components, and Zn bound only loosely to cell wall components might go into solution. Despite these significant caveats, our results were consistent with the notion that root cell walls with lower pectin methylesterification like shown for those of the cell walls of etiolated seedlings¹³ can sequester increased amounts of Zn.”*

and

L329 ff: *“Again, we extracted root crude cell walls according to the method of Zhong and Läuchli³⁴ and quantified Zn levels (Supplementary Fig12e, g). As outlined above, this commonly used method might potentially over or underestimate Zn levels in cell walls. Nevertheless, according to this method, the Zn levels in samples from these complemented lines are in alignment with the model that TBR alleles can confer increased TBR expression and thereby reduce pectin methylesterification leading to enhanced Zn sequestration within the root cell wall matrix. Overall, our data, in conjunction with the very robust, previously published data in etiolated seedlings¹³, suggest that under high Zn conditions sequence variation in the TBR promoter confers higher TBR expression levels that lead to lower pectin methylesterification that facilitate Zn sequestration in root cell walls and thereby enable root tolerance to elevated Zn levels (Fig.6c).”*

and line 427ff in the discussion:

“The higher expression level leads to TBR dependent lower pectin methylesterification (Fig. 6a, b). This had been shown to be accompanied with higher levels of cell wall Zn sequestration capacity in etiolated seedlings¹³, and we have obtained data (albeit the method we were able to utilize might be confounded by Zn from other cell compartments) consistent with this also happening in roots (Supplementary Fig. 12e-g).”

2. The reviewer disagrees with the response of the authors to the reviewer’s main comment. It seems that I did not make myself clear enough earlier (see Reviewer #2

earlier initial major comment, and subsequent comments 6, 7, and 8, and the authors' responses to these). Sinclair et al. (2017) showed and described in the text explicitly that root growth of the *Arabidopsis thaliana* partial loss-of-function mutants at the TBR locus (named *tbr* or *dez*) are hypersensitive to excess Zn, and they subsequently moved on to also examine the cell wall alterations in the mutant and its photomorphogenic phenotype. They showed multiple lines of evidence that both Zn exposure and the loss of TBR function lead to increased pectin methylesterification of cell walls and decreased amounts of Zn bound on cell walls, among other major cell wall and developmental alterations described in Sinclair et al. (2017).

In more detail, the results text of Sinclair et al. (2017) begins: "In a genetic screen for Zn-hypersensitive mutants, we identified an EMS mutant ... (this is identified as a partial loss-of-function mutant of the TBR gene later in the same manuscript). Later in the results text of Sinclair et al. (2017): "The *dez* (= *tbr*) mutant was hypersensitive to Zn in the light, with clear inhibition of root elongation, but not of hypocotyl elongation, when compared to the wild-type (Figures S1D and S1F)." ("in the light" referring to ordinary day-night growth conditions in this context, as a common growth condition employed in Sinclair et al. (2017) was continuous darkness). Sinclair et al. (2017) also showed that Zn tolerance was restored in a complemented line (*dez* transformed with a genomic TBR construct). Supplemental Figures S1I and S1J of Sinclair et al. (2017) also showed hypersensitivity of root growth of *dez* = *tbr* to the other heavy metals Ni and Cd in the light. Sinclair et al. (2017) included a detailed characterization of the general Zn hypersensitivity of growth and development in *tbr* mutants, including the photomorphogenic phenotype of *dez* = *tbr*, with a shortened hypocotyl, and the exacerbation of the phenotype specifically under excess Zn conditions. Sinclair et al. (2017) demonstrated that the degree of pectin methylesterification is enhanced in the *dez* = *tbr* mutant, decreased pectin acetylation as well as decreased Zn binding to cell walls of the *tbr* mutant (Fig. 4, Fig. S4), similar to, but going beyond, some results reported in this manuscript (e.g. Fig. 7). The authors must make changes to their manuscript text to accurately reflect this previously published work. In detail, this entails:

(a) Abstract lines 28-29 (my former comment 6, and the authors' response): "...which was not known to be involved in determining root growth variation in high Zn conditions." This sentence is still wrong, or at least highly misleading, in the revision. The authors should insert here: "as reported by Sinclair et al. (2017) in mutants carrying a loss of function allele at the TBR locus (i.e., *tbr-1*, *dez*).

We thank the reviewer for pointing out the issue, providing us with the reasoning underlying these issues, and generously providing excellent suggestions. We have revised the abstract to reflect the important previous study. The revised section of the abstract reads now:

"Among these loci, we identify TBR allelic variation determining root growth variation in high Zn conditions. Natural alleles of TBR determine TBR transcript and protein levels which affect the level of pectin methylesterification in root cell walls. Together with previously published data showing that pectin

methylesterification increase goes along with decreased Zn binding to cell walls in TBR mutants, our findings lead to a model in which TBR allelic variation enables Zn tolerance through Zn sequestration in root cell walls.” (lines 26-32)

(b) Line 57: In addition to mentioning the previous publication on *ozs2* as a Zn-sensitive mutant altered in cell wall pectin, the authors must mention here that Sinclair et al (2017) reported that root elongation of Arabidopsis EMS mutants with partial loss of TBR function was hypersensitive to excess Zn and associated with decreased pectin acetylation, increased pectin methylesterification and decreased cell wall binding of Zn²⁺, by inserting a sentence in line 57, before the beginning of the sentence “While...”.

*Thank you for this very clear and helpful suggestion. We have added this to the text as follows: “TBR has been previously identified via an EMS mutant screening for basal tolerance to excess Zn and the *dez (tbr)* mutants display Zn hypersensitivity in the light¹³” in lines 59-61.*

(c) Line 81 (my former comment 7, and the authors’ response): The authors write “... a gene, Trichome Birefringence (TBR) that was not known to be involved in determining root growth in high Zn conditions.” This is not correct as mentioned above. The authors must mention also here that a *tbr* mutant was previously identified in a forward genetic screen for Arabidopsis mutants hypersensitive to excess Zn, and that its root growth was found to be hypersensitive to Zn (Sinclair et al., 2017).

*Thank you for this very clear and helpful suggestion. We have added this to the text as follows: “We identify TBR allelic variants as important determinants of root growth variation in high Zn conditions. We go on to characterize TBR allelic variation in this context and show that allelic variation of TBR determines transcription and protein expression levels of TBR, as well as root cell wall pectin methylesterification. As a *tbr* mutant had been previously identified to be hypersensitive to excess Zn and Zn-enhanced photomorphogenesis displayed altered pectin methylation and acylation affecting photomorphogenesis and going along with a decreased capacity to bind Zn in cell walls¹³, our data give rise to a model in which TBR allelic variation enables Zn tolerance through Zn sequestration in root cell walls.” (lines 86-94)*

(d) Line 119/120 (my former comment 8, and the authors’ response): Title of this section again is not correct or at least misleading, given that partial TBR loss-of-function was published to cause decreased root growth in excess Zn conditions in Sinclair et al. (2017). This should be mentioned also here. Then, the authors can add that Sinclair et al. (2017) addressed artificially induced mutations and not a naturally occurring mutation as identified in this manuscript. In this way, if these two pieces of information are added, the text of this manuscript will no longer be misleading.

We thank the reviewer for the suggestion. The title of this section has been changed

to “Naturally occurring TBR alleles are associated with root growth variation in high Zn conditions.” (now **line 132-133**)

We also added to this section (**lines 172-178**):

“Moreover, TBR was also in linkage disequilibrium (LD) with the lead SNP of the GWAS peak (Supplementary Fig. 5a) and TBR had been shown to be involved in basal tolerance to excess Zn and repression of photomorphogenesis, as partial TBR loss-of-function caused decreased root growth and Zn-enhanced photomorphogenesis¹³. However, as previous study addressed artificially induced mutations and not natural occurring sequence variants¹³, we still considered both genes in proximity to the significantly associated SNP.”

(e) Line 134/135: The authors write “Interestingly, TBR had been previously identified via an EMS mutant screening to regulate photomorphogenesis in high Zn conditions.” The authors should modify this statement (as explained above):

“Interestingly, TBR had been previously identified via an EMS mutant screen conducted to identify genes required for basal tolerance to excess zinc Arabidopsis (31/Sinclair et al., 2017). Sinclair et al. (2017) reported additionally that TBR is necessary for the suppression of photomorphogenesis in the dark, with an exacerbated phenotype of the mutant in high-zinc conditions.” (Photomorphogenesis in the dark was observed under both control and high-zinc conditions, but was stronger in high zinc).

We thank the reviewer for the improvement. We have modified the sentences and added to this section (**lines 172-176**):

“Moreover, TBR was also in linkage disequilibrium (LD) with the lead SNP of the GWAS peak (Supplementary Fig. 5a) and TBR had been shown to be involved in basal tolerance to excess Zn and repression of photomorphogenesis, as partial TBR loss-of-function caused decreased root growth and Zn-enhanced photomorphogenesis¹³.”

3. My next major reservation relates to my previous comment no. 9 is on the designation and analysis of the TBR “promoter” as comprising 2437 bp upstream of the TBR translational start codon (line 234, line 435). While the reviewer appreciates the attempt of the authors to functionally validate candidate causal polymorphism, there are fundamental limitations to the data provided in the manuscript.

(a) The authors must discriminate between the segment that constitutes the promoter region (upstream of the transcriptional start site) and the contained segment that constitutes the 5'-UTR (downstream of the transcriptional start site and until the translational start codon). The promoter region can affect the rates of transcription and thus transcript levels, but it never affects protein levels to a substantially larger degree than it affects transcript levels. In this manuscript, however, protein levels appear to be affected to a much larger extent than transcript levels by the “promoter region” of A-allele individuals (Fig. 5e, compare Fig. 6c and 6d) – a finding of this manuscript that is even shown in the summarizing model (Fig. 8A) – so they do not see it as a consequence of differing experimental conditions or experimental sensitivities as

suggested by their response. Since the authors show that none of the polymorphisms in the TBR coding sequence affect protein levels, the over-proportionately large protein levels shown in lines transformed with a TBR driven by an A-allele promoter in Fig. 6, for example, are either an artefact from making an HA-mCITRINE-HA fusion protein or from not including introns in the construct, or they are caused by upstream sequence that is not part of the promoter but instead part of the 5'-UTR. It would be state-of-the-art to address the effects of candidate causal polymorphisms experimentally in a manner that discriminates between the promoter region and the 5'-UTR instead of addressing both together as the promoter. Thus, the conclusion in lines 247 to 250 does not seem correct to the reviewer.

There might be technical explanations that underlie the seemingly more drastic differences in the western blots compared to the qPCR data. One reason for this might be that it is generally more difficult to accurately quantify protein levels than transcript levels. In fact, the fluorescence measurements as quantified by confocal microscopy which is shown in Figure 5d (now Fig.4d), suggest differences in line with the qPCR data.

We agree with the reviewer on the potential effect of 5'-UTRs and introns. We have therefore included in line 422 ff:

“Between the two groups of accessions for which we elucidated differences in transcript and protein levels using complementation experiments, we found seven SNPs in the TBR promoter region (2213 bp upstream of the transcriptional start site) and one SNP in the 5'-UTR. These SNPs might mediate the observed differences via TBR transcript levels or via joint effects on transcript expression and posttranscriptional processes such as transcript stability or translation rates.”
We have also edited the model in Figure 6c accordingly.

(b) To support the hypothesis that the causal polymorphisms in the promoter region lead to variation in transcript levels, the authors generate *tbr-3* lines transformed with TBR coding regions under the control of an allele A (Zn-tolerant) Var2-6 and a T-allele (Zn-sensitive) Eds-1 upstream region. However, gene expression can vary substantially between independently transformed lines. The way the authors selected the lines that they characterized (and the low number of lines worked with) may have led to misconclusions concerning relative promoter strengths (line 454-459): (i) Even if only a single T-DNA locus segregates, tandem insertions of two or more T-DNA copies are extremely common. (ii) The authors chose lines with a desired fluorescence signal of the TBR-mCITRINE fusion protein. So in essence, what the authors do show here is that Zn tolerance in a *tbr* mutant background associate not only with intact TBR transcript levels (which was known before, Sinclair et al., 2017) but also with the amount of TBR protein produced, irrespective of the cause of differences in protein levels (Fig. 6; the phenotypic data in Fig. 5a is far less clear-cut based on statistics, line 201, where only part of the Var2-6 TBR promoter-containing lines are statistically significantly more Zn-tolerant than only part of the Eds-1 TBR promoter-containing lines) (this relates to my earlier comment 5).

We agree with the reviewer that T-DNA insertions can be inserted in multiple locations and in multiple copies. We can exclude that these were inserted in multiple locations that can be differentiated by segregations ratios as we have only worked with lines that segregated in a mendelian manner. Formally we can't exclude multiple insertions in close proximity, however as we selected lines in T2 and T3 unbiasedly, we would expect this to occur only randomly. In this case there would not be a preference for one specific group/construct. Thus, multiple insertions or insertions in regions of the genome that confer higher/lower expression would generate more expression variation regardless of the group/construct and thus reduce the power to detect differences. It would seem very unlikely that our experimental design would selectively affect one set of constructs and not the other as the constructs are basically the same except for a handful of SNPs.

To make our unbiased selection criteria clearer, we added additional description in material and methods (line 548ff):

“In the T1 generation, 24 independent lines were selected for each construct. From these, 5 independent transgenic lines were selected for further characterization according to the following criteria: (1) single insertion line (the T2 segregation ratio between sensitive and resistant followed the expected Mendelian distribution for a dominant single locus of 75% resistant and 25% sensitive); (2) a visible fluorescence signal of the mCITRINE fusion protein detected in the root by confocal microscopy (with no regard to the strength or the specific construct); (3) no obvious abnormal root growth phenotype. This selection was unbiased and random in a sense of that the first 5 lines that displayed these properties were selected. In the T3, homozygous lines were selected using similar criteria as in the T2 generation except for that we selected 100% resistant homozygous lines.”

4. Lines 112/113: The authors should alter the description of the results so that the text matches better with the diagrams. “Compared with wildtype, frd3 mutants showed a significant reduction of root growth and severe chlorosis in response to high Zn (Fig. 2)”. This statement is in disagreement with the data provided in Fig. 2b and d. frd3-7 has significantly shorter roots and a trend for less chlorophyll than the corresponding wild type (Col-0) already under control conditions. Under high Zn conditions, the differences between frd3-7 and the wild type are not increased for root length and chlorophyll. In addition, there is no significant difference between frd3-3 and the corresponding wild type gl-1 for root length at high Zn (Fig. 2c). In frd3-3 chlorophyll is a bit down, but not statistically significantly in high Zn in frd3-3. However, chlorophyll in gl-1 is significantly down in high Zn compared to control in gl-1. Whether gl-1 is the wild type corresponding to frd3-3 is a bit unclear to me. The authors state this in their methods, but Rogers et al. (2004) write that frd3-3 is man1, and Delhaize et al. (1996) who published man1 write that frd3-3 is in the Col-0 background. In any case, the data shown in this manuscript do not unequivocally support Zn hypersensitivity in frd3 mutants, different from what is written in the manuscript text.

We thank the reviewer for pointing this out. We confirmed this in our reading of the literature and with Mary Lou Guerinot. We added this in the material and methods. We therefore have compared both lines to Col-0.

We have checked the root length data of frd3 mutants and Col-0, we found that we made a mistake when we copied the data. Now we have redrawn the graph of Figure 2b (now Fig. 1e) and placed the correct data to the source data. We also have added 2-way ANOVA. As we focused on root length under high Zn, we also removed the chlorophyll measurements in high Zn soils. We have now updated the section in this manuscript:

Line 123ff: “Compared to the Col-0 wildtype, frd3-3 mutant showed a significant reduction of root growth in response to high Zn according to the 2-way ANOVA (P-value < 0.001) while the genotype x treatment interaction was marginally significant for frd3-7 (P-value = 0.063) (Fig. 1d, e; Supplementary Fig. 4; Supplementary Tab.2, 3).”

5. Lines 148-151, referring back to my previous comment 2: The authors state that A allele accessions had higher TBR transcript levels than T allele accessions. However, according to the shown relative transcript levels of TBR, there was no significant difference between accessions in control conditions. Moreover, there was no significant difference between two of the A allele accessions and the T-allele accessions in high Zn, either. Only transcript levels of the third A allele accession, Sim-1 were slightly higher (1.17 to 1.3-fold) than in the T-allele accessions under high-Zn conditions. Quantitatively, transcript level differences were minor and not comfortably within the detection range of differences by RT-qPCR (for such extremely small transcript level differences, ddPCR should be used to obtain reliable results).

We performed transcription analysis in roots of these contrasting accessions which were grown on 1/2 MS medium for 5 DAG and then transferred to control and high Zn (300 μ M) liquid medium for 12 h (the treatment method for transcription analysis is the same as that for TBR allelic variant transgenic lines and promoter allelic variant transgenic lines). Three biological replicates with two technical replicates were analyzed for each gene. We found that the transcript level of TBR in Vår2-6, Sim-1 and Fri 1 that are from the A-allelic accessions was higher than that of the T-allelic accessions, while AT5G06710 showed no significant difference between the two allelic groups of accessions both in control and high Zn conditions. The results have been shown in Fig. 2c and Supplementary Fig. 5i.

We acknowledge that this is close to the comfortable qPCR detection limit and are grateful to the reviewer to point this out and we have therefore rephrased our text carefully as follows:

“To then test whether the response of the candidate genes in the allelic groups of accessions that were identified by GWAS was consistent with the causal involvement of one of the two candidate genes, we selected representative accessions from the T-allelic group (Eds-1, Eds-9 and TRÄ 01) and the A-allelic

group (*Vår2-6*, *Sim-1* and *Fri 1*). Additionally, we also included *Col-0* with an intermediate root length, which was from A-allelic group. We found that the transcript level of *TBR* in *Vår2-6*, *Sim-1* and *Fri 1* that are from the A-allelic accessions under high Zn conditions was slightly but significantly higher than that of the T-allelic accessions (*Col-0* which was in the A-allelic group but didn't display a significantly higher expression level of *TBR* compared to the T-allelic accessions). In contrast, *AT5G06710* showed no significant difference between the two allelic groups of accessions both in control and high Zn conditions (Fig. 2c; Supplementary Fig. 5i). While the detected differences were at the lower end of changes that can be reliably detected via RT-qPCR, nevertheless this analysis hinted towards *TBR* to be the best candidate gene within the proximity of the GWAS association for being causal for root growth variation in high Zn." (lines 157-172)

Other issues:

6. Line 73 and following: The authors state "So far, only Ferric Reductase Defective 3 (FRD3) has been identified to be underlying natural variation of the root growth responses to Zn toxicity tolerance in *Arabidopsis thaliana*, by regulating Fe and Zn translocation from the root to the shoot". The reviewer feels that, although this is strictly speaking correct, the scope of this statement is artificially narrow and thus misrepresents the state of knowledge.

(a) Based on a number of mutants and publications, it is clear that *MTP1*, *MTP3*, *HMA3*, *HMA2* and *HMA4* and the corresponding protein functions, for example, have central roles in basal Zn tolerance of *A. thaliana*.

(b) Studies of *Arabidopsis*-related Zn-hypertolerant hyperaccumulator species (*A. halleri*, *N. caerulescens*) that exhibit extreme Zn tolerance and the alleles therein (often affecting the expression of Zn tolerance genes via cis-regulatory alterations) have provided major insights, with roles for *HMA4*, *MTP1*, and several other genes. This work should be mentioned either in the introduction or in the discussion.

We thank the reviewer for highlighting a lack of clarity here. We edited the text so that it is more comprehensive:

*"Plants have developed tolerance mechanisms to keep Zn homeostasis when exposed to excess concentration of Zn in the environment. These include intracellular binding to Zn chelators, sequestration into the vacuole, efflux from root cells, reduction of transport to plastids and cross-homeostasis between Iron (Fe) and Zn^{1, 7, 8}. Some genes such as *HMA4* and *MTP1*, which are involved in Zn tolerance, showed higher gene expression in Zn hyperaccumulators *A. halleri* and *N. caerulescens* compared to *Arabidopsis thaliana*¹." (lines 47-53)*

We also clarified further in the part of the introduction the reviewer referenced:

*"While genes and encoding proteins such as *MTP1*, *MTP3*, *ZIF1*, *HMA2*, *HMA3*, *HMA4*, *PCR2* have central roles in basal Zn tolerance of *Arabidopsis thaliana*¹, and some genes such as *HMA4* and *MTP1*, which are involved in Zn tolerance, show increased gene expression in Zn hyperaccumulators *A. halleri* and *N. caerulescens* compared to *Arabidopsis thaliana*¹, natural alleles underlying*

natural variation of Zn tolerance within plant species remain largely unknown. So far, Ferric Reductase Defective 3 (FRD3) has been identified to be underlying natural variation of the root growth responses to Zn toxicity tolerance in Arabidopsis thaliana, by regulating Fe and Zn translocation from the root to the shoot¹⁵.” (lines 73-81)

(c) Candidate loci at a major peak on chromosome 4 identified in this submission should at least be mentioned in the results or discussion. Is this peak in the vicinity of HMA2, HMA3, HMA1, IRT1 (all on chromosome 4 and previously shown to affect Zn tolerance).

We thank the reviewer for this suggestion. We have added in the result part “Next, we analyzed the most significant peak and candidate genes in a window of 10 kb around the top SNP (6,817,712) on chromosome 4. While HMA1, HMA2, HMA3, and IRT1 are genes on chromosome 4 that have been previously implicated in Zn tolerance¹, none of these genes were in the vicinity of the peak (the closest of these genes is about 3.9 Mb away from the peak). Moreover, the T-DNA mutants of the genes in the region of the peak (AT4G11170, AT4G11175, AT4G11180 and AT4G11190) that we tested for a growth phenotype in high Zn conditions didn’t show a phenotype (data not shown).” (lines 133-140)

7. Fig. 7 title: The predicted function of TBR makes it extremely unlikely that this protein mediates pectin demethylesterification. According to a number of previous publications, the activity of TBR is highly likely to primarily affect pectin O-acetylation, and the effect on pectin methylesterification is only a secondary effect caused by the highly homeostatic character of cell wall metabolism.

We agree with the reviewer and have changed the title of Figure 7 (now Figure 6) to “Fig. 6 TBR variants cause changes in pectin methylesterification and the mechanism by which TBR regulatory allelic variation regulates root tolerance to elevated Zn levels.”

We also carefully checked the text and further edited

“TBR plays a role in pectin O-acetylation and this is associated with pectin modifications in the cell wall^{13, 33} and tbr mutants display increased levels of methylesterified pectin in etiolated seedlings¹³.” (lines 287-290)

8. Line 261: “We found that the roots of tbr mutants showed lower pectin content compared to wildtype in high Zn conditions. “  refer here to Figure where this is shown.

*The reference to figure of pectin content is in Supplementary Fig. 12a, which is now moved to **line 294**.*

9. Lines 357 and following: Here the authors should write about the known relationship

of TBR with pectin O-acetylation (Sinclair et al., 2017, Chiniquy et al., 2019; Zhong et al., 2023); in the present form this section in the discussion is easily misread.

*We thank the reviewer for the improvement. We have added the known relationship of TBR with pectin O-acetylation “TBR encodes an O-acetyltransferases responsible for pectin acetylation^{13, 44, 45}. The tbr mutant showed predominant alterations in pectin and pectin modification by Micro-Fourier-transformed infrared (μ FTIR) spectra¹³. Furthermore, Sinclair et al. described that the tbr mutant exhibited an increased level of pectin methylesterification and decreased level of pectin acetylation and Zn content in the cell wall¹³.” (now **lines 435-440**)*

10. Line 361: Sinclair et al (2017), who showed this very specifically for Zn and tbr mutants compared to the wild type, should be cited here, too.

*Thank you for pointing this out. The reference of Sinclair et al (2017) has been cited in line 361 (now **line 435**).*

11. Line 364: Change to “Zn exposure induced an increase in.”

*Thank you for the correction. It has been modified. (now **line 440-441**)*

12. Line 366: Discuss here also Sinclair et al (2017), who observed an opposing change in pectin methylesterification under exposure to excess Zn in the wild type.

We have added the following text to the discussion “Similarly, our study revealed that Zn exposure induced an increase in pectin methylesterification in root cell walls of tbr mutants. However, there are differences between Sinclair et al.’s findings and ours. In Sinclair et al., cell wall pectin methylesterification was slightly increased in Col-0 and dez (tbr) upon high Zn levels. In our study, we find a decrease of cell wall pectin methylesterification in Col-0 upon high Zn conditions and an increase in tbr (Supplementary Fig. 12a, b). It seems most likely that these differences are due to the different growth conditions or seedling ages in the studies.” (lines 440-447)

13. The authors discuss roles exclusively for FRD3 and TBR in natural variation of Zn tolerance in *A. thaliana*, but a major peak was observed on chromosome 4, for which no gene is mentioned here, and TBR and FRD3 are only a set of 2 out of 21 loci the authors detected in this GWAS analysis to affect Zn tolerance. The discussion in lines 371-375 should thus be modified.

To address this, we added:

“Our study detected 21 significantly associated loci for root growth responses to Zn toxicity (Supplementary Fig. 2; Supplementary Data 3) and provides a resource for identifying genes involved in Zn tolerance in plants. We explored causal genes

for only two of these loci as outlined below and causal genes for the other loci remain yet to be identified. This includes the most significant peak on chromosome 4. While we had tested multiple T-DNA insertion lines for several genes in the proximity to the Chromosome 4 GWAS peak, we didn't find support for any of the respective candidate genes. However, this is not entirely surprising as such reverse genetics approaches are commonly hampered by genetic redundancy and robustness, and potential interactions of the genetic background and the tested genes. Further investigations, in particular for the region on chromosome 4, might be worthwhile.” (lines 376-386)

14. The discussion should mention the role of TBR in light-dependent development and the influence of Zn on it. Even if not analysed here, it is likely affect the natural accessions of *A. thaliana* studied here.

We thank the reviewer for the helpful suggestions. We have quantified hypocotyl length both in TBR allelic complemented lines and TBR promoter allelic transformed lines cultivated on control and high Zn medium in continuous darkness. Consistent with the mechanism that TBR allelic variation determines root growth in high Zn, T-allelic variant lines showed shorter hypocotyl length compared to A-allelic variant lines in etiolated seedlings both in TBR allelic complemented lines and TBR promoter allelic variants (Supplementary Fig.16).

We have discussed these two mechanisms in the discussion as

*“Consistent with the previously observed Zn enhanced photomorphogenesis¹³, we found that the *tbr-3* mutant showed a shorter hypocotyl length compared to wildtype, and the hypocotyl length was rescued to almost wildtype levels through complementation with a functional TBR allele in high Zn conditions, while the hypocotyl length showed no significant difference in control conditions (Supplementary Fig.16a). Moreover, T-allelic variant complementation lines displayed shorter hypocotyl lengths compared to the A-allelic variant complementation lines in etiolated seedlings (Supplementary Fig.16b, c). These results indicate a similar mechanism of TBR variants in regulating basal tolerance to excess Zn and in suppression of photomorphogenesis most likely due to the effect that TBR variants have for structural modification of cell walls that lead to enhanced Zn sequestration and increased Zn tolerance.” (lines 451-461)*

15. Referring back to my previous comment 3: As the TBR family has 46 members in *A. thaliana* (and 7 in the TBR-related group), I had in mind a more extensive phylogenetic tree covering at least the full TBL groups (all members) of Arabidopsis, rice and *Lotus japonicas* so that the reader can assess orthology.

*We have made a phylogenetic tree covering of 46 TBL members in Arabidopsis, 66 TBL members in *Oryza sativa* and 52 TBL members in *Lotus japonicus*, which is shown in Supplementary Fig. 13a. As some TBL members show low identity with TBR which prohibits estimation of the pairwise distance by MEGA 6, some TBL*

members for which the identity is lower than 30% were deleted, this included TBL12 and TBL 20 in Arabidopsis, OsTBL 7, OsTBL42, OsTBL62 and OsTBL66 in *Oryza sativa*.

Figure legend: Phylogenetic analysis of TBR homologous protein from Arabidopsis, *Oryza sativa* and *Lotus japonicus*. MEGA 6 was used to construct a neighbor-joining phylogeny (bootstrap replications = 1,000). Asterisks represent TBR, LjTBR, OsTBL31 and OsTBL32.

Reviewers' Comments:

Reviewer #1:

Remarks to the Author:

This reviewer thanks the authors for addressing the reviewer's concerns appropriately. I have no further comments.

Reviewer #2:

Remarks to the Author:

The larger issues that I had previously raised concerning discrepancies between the results obtained and the manuscript text, misrepresentation of the literature, as well as the methodological issues, have now been clarified by describing/citing the appropriate literature, by describing results in more detail and mentioning caveats and limitations in a more balanced way.

In the final modified version of the manuscript I have two remaining issues:

1. The phylogenetic tree: (a) The authors mention bootstrapping, but the no bootstrap values are given on the branches. (b) The labels of the tree are impossible to read for me in the Suppl. File. (insufficient resolution) and should be improved. (c) Finally, the authors should verify their phylogenetic tree. I did a quick Blast comparison between the Lj sequence that is drawn right next to TBR, and it has lower % identity (49%) and lower query coverage (0.56) compared to TBR than TBL1 (65%, 0.99) has. Even TBL2 has 40% identity (0.83 coverage)(http://aramemnon.uni-koeln.de/clist_view.ep?GeneID=336&GeneClusterID=20423). (d) This makes me wonder which part of the TBL proteins the tree might have been constructed on (or the full lengths?).
2. The authors should replace each statement of "TBR regulates" by "TBR influences" or "TBR affects".

Reviewer #4:

Remarks to the Author:

Zhong and collaborators report that natural variation in TBR gene underlies variation in zinc sensitivity in *Arabidopsis thaliana* and that the role of this gene is conserved in rice and *Lotus japonicus*. Using GWAS for root growth response to excess zinc, the authors identify 21 candidate loci among which they further study two. In agreement with a previous publication (Pineau et al., 2012), they confirm that FRD3 is involved in natural variation in zinc tolerance. Furthermore, they show that variation in the expression TBR (Trichome Bi Refringence), encoding an O acetyl transferase, also accounts for natural variation in zinc sensitivity. Loss of TBR function or decreased expression of TBR is associated with increase in cell wall pectin methyl esterification. The authors propose that this decreases cell wall zinc binding capacity leading to increased zinc sensitivity.

This study corroborates a previous publication establishing the involvement TBR in zinc tolerance and further show that TBR accounts for natural variation in zinc tolerance. Most but not all conclusions are adequately supported by the data. The text is clear, although sometimes lacking precision, especially in the figure legends.

Major points

1) The method used to determine the concentration of zinc associated to cell wall is irrelevant. First homogenizing tissues before purifying cell wall and then measuring zinc in the cell wall fraction cannot provide an accurate determination of the amount of zinc associated to the cell wall because zinc is bound to the cell wall components by ionic interactions. These are sensitive to pH which is disturbed upon homogenization. During this process, the zinc present in other compartments is also redistributed. Accordingly, the amount of zinc measured in the cell wall closely follows the total amount of zinc measured in whole roots (supp figure 12).

The way the results obtained with this method are presented is twisted. The results are presented as flawed but still shown in the supplemental data and the conclusion is used in the title. Leaving these

results and the title as they are will set another precedent that other authors will cite to justify using an irrelevant method. The authors must either demonstrate this conclusion in a convincing way or completely omit this approach and the corresponding conclusions (including in the title), as well as the discussion. Imaging zinc by X-ray fluorescence in flash frozen roots would be a valid, although challenging, method. Another possibility would be to measure how much zinc can be desorbed from intact roots at low temperature in the different genotypes. An alternative interesting approach could be to measure the zinc binding capacity of cell wall purified from the different genotypes, either after removing endogenous zinc or using exchange with a zinc isotope.

2) Although the data generally support a simple relationship between the level of functional TBR and zinc tolerance, there are a few discrepancies hinting at a more complex scenario:

- the phenotype of the *tbr-1* mutant is much more severe than that of the *tbr-3*, which is considered as a complete loss of function. The *tbr-1* allele is not further studied on the basis that the effect of the mutation is not clear. However, it suggests a possible dominant negative effect of the *tbr-1* mutation.
- in figure 4a, one of the "complemented" lines TBR Eds1-11 is also more sensitive to zinc than the *tbr-3* loss of function mutant. This suggests that low expression of TBR may make the plant more sensitive to zinc than a complete absence of this protein.
- in figure 5a, the authors show that the L186R mutant of TBR is unable to complement *tbr-3*, suggesting that the mutant protein is not functional. The authors also state that the L186R mutation is found in some A alleles (the ones that confer increased tolerance). This suggests that expression of a non-functional TBR protein sometimes confer increased tolerance. These intriguing results certainly deserve further investigation or at least discussion.

Minor points

- 1) figure 3c and 3d show that the elongation zone is shorter in *tbr-3*. Does it result in shorter mature/fully elongated root cells?
- 2) the result section and the figure legend should indicate which antibody was used for the immunoblots.
- 3) the immunoblots in figures 4e and 5d should be quantified to show the variability between replicates.
- 4) It seems that gene expression levels in figures 4b and 5c are normalized to the first column, but this is not indicated in the legend.
- 5) In figures 4b and 5c, what would be the endogenous expression level in *col-0*? This would be useful to relate expression levels to complementation shown in 4a and 5b.
- 6) In the legend of figures 4a, 5a and 5b, it should be stated more clearly that the percentage indicated on the top of each high zinc box corresponds to the decrease relative to root length in control conditions.
- 7) In supplementary figure 12, it is surprising that roots accumulate about 10 times more zinc when exposed to 200 microM zinc for 7 days than when exposed to 150 microM zinc for

REVIEWER COMMENTS

Reviewer #1 (Remarks to the Author):

This reviewer thanks the authors for addressing the reviewer's concerns appropriately. I have no further comments.

Reviewer #2 (Remarks to the Author):

The larger issues that I had previously raised concerning discrepancies between the results obtained and the manuscript text, misrepresentation of the literature, as well as the methodological issues, have now been clarified by describing/citing the appropriate literature, by describing results in more detail and mentioning caveats and limitations in a more balanced way.

In the final modified version of the manuscript I have two remaining issues:

1. The phylogenetic tree: (a) The authors mention bootstrapping, but the no bootstrap values are given on the branches. (b) The labels of the tree are impossible to read for me in the Suppl. File. (insufficient resolution) and should be improved. (c) Finally, the authors should verify their phylogenetic tree. I did a quick Blast comparison between the Lj sequence that is drawn right next to TBR, and it has lower % identity (49%) and lower query coverage (0.56) compared to TBR than TBL1 (65%, 0.99) has. Even TBL2 has 40% identity (0.83 coverage)(http://aramemnon.uni-koeln.de/clist_view.ep?GeneID=336&GeneClusterID=20423). (d) This makes me wonder which part of the TBL proteins the tree might have been constructed on (or the full lengths?).

We have made a phylogenetic tree using the full length of TBL protein, which contains the bootstrap values. The phylogenetic tree covers 46 TBL members in Arabidopsis, 66 TBL members in Oryza sativa and 57 TBL members in Lotus japonicus, which is shown in Supplementary Fig. 13. As some TBL members show low identity with TBR which prohibits estimation of the pairwise distance by MEGA 10, 4 TBL members for which the identity is lower than 30% were deleted, this included OsTBL 7, OsTBL42, OsTBL62 and OsTBL66 in Oryza sativa.

We did Blast on the website of phytozome for Arabidopsis and Oryza sativa and Lotus base for Lotus japonicus. TBL1 showed 64% identity and 75% positives (<https://phytozome-next.jgi.doe.gov/blast-results/983313>) and Lj1g3v4350070 presented 60.5% identity and 73.7% positive (https://lotus.au.dk/blast/#Query_1_hit_68).

2. The authors should replace each statement of "TBR regulates" by "TBR influences" or "TBR affects".

We thank the reviewer for pointing this out. It has been modified in the text and figure legends accordingly.

Reviewer #4 (Remarks to the Author):

Zhong and collaborators report that natural variation in TBR gene underlies variation in zinc sensitivity in *Arabidopsis thaliana* and that the role of this gene is conserved in rice and *Lotus japonicus*. Using GWAS for root growth response to excess zinc, the authors identify 21 candidate loci among which they further study two. In agreement with a previous publication (Pineau et al., 2012), they confirm that FRD3 is involved in natural variation in zinc tolerance. Furthermore, they show that variation in the expression TBR (Trichome Bi Refringence), encoding an O acetyl transferase, also accounts for natural variation in zinc sensitivity. Loss of TBR function or decreased expression of TBR is associated with increase in cell wall pectin methyl esterification. The authors propose that this decreases cell wall zinc binding capacity leading to increased zinc sensitivity. This study corroborates a previous publication establishing the involvement TBR in zinc tolerance and further show that TBR accounts for natural variation in zinc tolerance. Most but not all conclusions are adequately supported by the data. The text is clear, although sometimes lacking precision, especially in the figure legends.

We would like to thank the reviewer for their thorough review of the manuscript and the very helpful suggestions.

Major points

1) The method used to determine the concentration of zinc associated to cell wall is irrelevant. First homogenizing tissues before purifying cell wall and then measuring zinc in the cell wall fraction cannot provide an accurate determination of the amount of zinc associated to the cell wall because zinc is bound to the cell wall components by ionic interactions. These are sensitive to pH which is disturbed upon homogenization. During this process, the zinc present in other compartments is also redistributed. Accordingly, the amount of zinc measured in the cell wall closely follows the total amount of zinc measured in whole roots (supp figure 12).

The way the results obtained with this method are presented is twisted. The results are presented as flawed but still shown in the supplemental data and the conclusion is used in the title. Leaving these results and the title as they are will set another precedent that other authors will cite to justify using an irrelevant method. The authors must either demonstrate this conclusion in a convincing way or completely omit this approach and the corresponding conclusions (including in the title), as well as the discussion. Imaging zinc by X-ray fluorescence in flash frozen roots would be a valid, although challenging, method. Another possibility would be to measure how much zinc can be desorbed from intact roots at low temperature in the different genotypes. An alternative interesting approach could be to measure the zinc binding capacity of cell wall purified from the different genotypes, either after removing endogenous zinc or using exchange with a zinc isotope.

We thank the reviewer for these important suggestions, helpful explanations and guidance. As we currently don't have the capacity to experimentally address this and considering that other authors would cite to justify using an irrelevant method, we have now completely deleted this approach, the related images in supplementary figure 12 and the conclusions (including in the title), as well as the discussion and the description of this protocol in the Method section.

2) Although the data generally support a simple relationship between the level of functional TBR and zinc tolerance, there are a few discrepancies hinting at a more complex scenario:

- the phenotype of the *tbr-1* mutant is much more severe than that of the *tbr-3*, which is considered as a complete loss of function. The *tbr-1* allele is not further studied on the basis that the effect of the mutation is not clear. However, it suggests a possible dominant negative effect of the *tbr-1* mutation.

We thank the reviewer for pointing this out and now have specified this:

*Line 206ff: "For this and the subsequent analyses, we utilized the *tbr-3* loss of function mutant, as the *tbr-1* mutation has a negative effect on TBR function, but it is not clear whether it is a simple loss of function mutant or a gain-of-function allele with a negative function."*

- in figure 4a, one of the "complemented" lines TBR Eds1-11 is also more sensitive to zinc than the *tbr-3* loss of function mutant. This suggests that low expression of TBR may make the plant more sensitive to zinc than a complete absence of this protein.

We thank the reviewer for this suggestion, we added to the discussion:

*Line 410ff: "It is interesting to note that one of the T-allelic complemented lines (TBR Eds1-11) showed a lower high-Zn root growth tolerance than the *tbr-3* loss of function mutant line. This might indicate that in some circumstances low expression of TBR might render roots more sensitive than a full loss of function."*

- in figure 5a, the authors show that the L186R mutant of TBR is unable to complement *tbr-3*, suggesting that the mutant protein is not functional. The authors also state that the L186R mutation is found in some A alleles (the ones that confer increased tolerance). This suggests that expression of a non-functional TBR protein sometimes confer increased tolerance.

These intriguing results certainly deserve further investigation or at least discussion.

We thank the reviewer for this keen observation! While we don't have the capacity to experimentally address this, we have added this to the discussion:

Line 413ff: "Another intriguing observation from the complementation experiments was that the TBR protein containing an amino-acid substitution (L186R) that was observed in the more tolerant A-allelic group of accessions did

not rescue the tbr-3 mutant root growth phenotype under high Zn conditions when driven by the Col-0 promoter (Fig 5a). This might suggest that either a non-or less-functional version of the TBR protein might confer Zn-tolerance when expressed at higher levels (as conferred through the higher expressing A-allelic TBR variant promoters) or that the genetic background of A-allelic accession can confer full functionality to the TBR-L186R variant (e.g. by expressing certain variants of interacting proteins or genes in the pathway). Overall, further investigation into the molecular mechanism of TBR function will be very interesting and might illuminate interesting engineering strategies for TBR facilitated Zn tolerance.”

Minor points

1) figure 3c and 3d show that the elongation zone is shorter in tbr-3. Does it result in shorter mature/fully elongated root cells?

We have measured root length of meristem zone and mature cortical cells. The length of meristem zone and the mature cortical cell was not significantly reduced in the tbr-3 compared to wildtype under high Zn conditions (Supplementary Fig. 8).

2) the result section and the figure legend should indicate which antibody was used for the immunoblots.

Thank you for the suggestion. We have modified the result section as “we evaluated the TBR protein levels by confocal microscopy of TBR fused with mCITRINE and western blot with an anti-HA antibody” in lines 243-244.

and added in the figure legend of figure 4e and 5d “Western blot of HA-tagged TBR was conducted using an anti-HA antibody”.

3) the immunoblots in figures 4e and 5d should be quantified to show the variability between replicates.

Thank you for the improvement. We have quantified the immunoblots in figures 4f and 5e and added in L281ff:

“When we quantified transcription level via qRT-PCR, we found that the TBR_{vär2-6}-TBR_{Col-0-16} transgenic line showed a drastically increased transcription level compared to both TBR_{Eds-1}-TBR_{Col-0} transformed lines in both control and high Zn conditions, while the other TBR_{vär2-6}-TBR_{Col-0} line (TBR_{vär2-6}-TBR_{Col-0-4}) displayed a statistically significant increased transcript level compared to TBR_{Eds-1}-TBR_{Col-0} transformed lines in high Zn conditions (Fig. 5c). Assessments of the protein level via western blotting supported an increased accumulation of TBR protein in both TBR_{vär2-6}-TBR_{Col-0} lines compared to TBR_{Eds-1}-TBR_{Col-0} lines (Fig 5d, e).”

As we described before, there might be technical explanations that underlie the seemingly more drastic differences in the western blots compared to the qPCR data. One reason for this might be that it is generally more difficult to accurately quantify protein levels than transcript levels. In fact, the fluorescence measurements as quantified by confocal microscopy which are shown in Figure 4d, suggest differences in line with the qPCR data.

4) It seems that gene expression levels in figures 4b and 5c are normalized to the first column, but this is not indicated in the legend.

We thank the reviewer for the improvement. We have added to the figure legend of figures 1c, 2c, 4b and 5c.

5) In figures 4b and 5c, what would be the endogenous expression level in col-0? This would be useful to relate expression levels to complementation shown in 4a and 5b.

We haven't tested the endogenous expression level in Col-0 together with allelic complemented lines. However, we compared gene expression levels in roots of representative accessions (Sensitive: Eds-1 and Eds-9; Intermediate: Col-0; Tolerant: Vår2-6 and Sim-1) in figure 2c. The results were depicted as "To then test whether the response of the candidate genes in the allelic groups of accessions that were identified by GWAS was consistent with the causal involvement of one of the two candidate genes, we selected representative accessions from the T-allelic group (Eds-1, Eds-9 and TRÅ 01) and the A-allelic group (Vår2-6, Sim-1 and Fri 1). Additionally, we also included Col-0 with an intermediate root length, which was from A-allelic group. We found that the transcript level of TBR in Vår2-6, Sim-1 and Fri 1 that are from the A-allelic accessions under high Zn conditions was slightly but significantly higher than that of the T-allelic accessions (Col-0 which was in the A-allelic group but didn't display a significantly higher expression level of TBR compared to the T-allelic accessions)" in lines 158-167.

6) In the legend of figures 4a, 5a and 5b, it should be stated more clearly that the percentage indicated on the top of each high zinc box corresponds to the decrease relative to root length in control conditions.

We thank the reviewer for pointing out the issue. We have revised it as "The value on the top of each box in high Zn is the relative root length (root length in high Zn / root length in control)" in the legend of figures 4a, 5a and 5b.

7) In supplementary figure 12, it is surprising that roots accumulate about 10 times more zinc when exposed to 200 microM zinc for 7 days than when exposed to 150 microM zinc for 14 days.

We think that the reasons may be due to differences in sample harvest time and Zn concentration in the medium. For Zn content in roots and root cell walls of Col-0, tbr-3 and TBR_Col-0-9, 14 DAG seedlings cultivated in control and 150 μ M Zn medium were harvested. For Zn content in allelic complemented lines, 7 DAG seedlings in control and 200 μ M Zn medium were harvested.

Reviewers' Comments:

Reviewer #2:

Remarks to the Author:

I agree with the acceptance of the manuscript for publication, provided that the following changes are implemented (despite what was stated in the authors' response letter, some changes I had highlighted earlier were not made in the manuscript):

1. Suppl. Fig. 12c/d has now gained substantially more importance, given that the methodologically flawed cell wall Zn data have been removed (the reviewer agrees strongly with the removal). In the methods, the authors say that they washed the roots in 10 mM CaCl₂ for 5 mins and then 10 mM EDTA for 5 mins. Such kind of washes are state-of-the-art for removing cell wall-bound zinc, although they were shorter in this manuscript than is usually done. In any case, at least part of and potentially all of the cell wall-bound Zn were removed here from roots prior to the analysis of Zn concentrations in plant tissues shown in Suppl. Fig. 12c/d, with generally little control over this. In the opinion of the reviewer, the interpretation of the results shown in Suppl. Fig. 12c/d (as reflecting cell wall-bound Zn) in the results and discussion sections are not permissible here. The reviewer suggests that the authors remove all interpretation of these results as reflecting cell wall-bound Zn (lines 323 to 331, line 442/443, and ... roots.) and refer merely to ref. 11 concerning the question of Zn binding ability of the cell walls. In this context it remains peculiar that minor differences in age and growth conditions lead to such dramatic differences in Zn concentrations in roots (between Suppl. Fig. 12c and d). The reviewer suggests that the authors add a sentence on this in the discussion.
2. Line 54: In addition to reference 1, the reviewer believes that a minimum would be to cite doi: 10.1038/nature06877 for HMA4, and possibly also 10.1534/genetics.106.064485 and 10.1016/j.febslet.2005.06.046 for MTP1, as well as <https://doi.org/10.1016/j.febslet.2007.04.010> for the comparisons of gene expression changes in *Noccaea* and *A. halleri*. Merely citing a rather derived review seems insufficient here.
3. The authors use the term "basal tolerance" in lines 61, 75 and 175. This term should be defined at first use (line 61). ("Basal metal tolerance" is the metal tolerance level common to all plants including *A. thaliana*, and the term is used to discriminate this from naturally selected metal hypertolerance which is found only in specialist plants that are found on soils containing high, toxic levels of metals. This was a modification in reference 11 of the terms coined in 10.1007/s004250000458, further detailed here 10.1016/j.bbamcr.2012.05.016)
4. L. 176: "decreased root growth"; add after this here: "under high-Zn exposure". Else it would be inaccurate.
5. Line 204: "required for root growth tolerance to high Zn"; please add at the end of this sentence here (by expanding the sentence: ", as shown previously¹¹". Else this remains too obscure in the context of the way reference 11 is referred to in the earlier part of the manuscript.
6. Line 243/244: fused with both mCitrine and HA – else this is hard to understand.
7. Line 245: "transcript" (not: "transcription"), again line 282. Please check also for the same issue elsewhere in the text.
8. Line 255-256: when analyzing the full genome sequence at the TBR locus"  rephrase this sentence. It is confusing: either the authors analyzed the full genome sequence or they analyzed the sequence at the TBR locus.
9. Line 283: "... showed a drastically increased...". This is not true. Depending on which transgenic lines are compared, the difference is slight to moderate, but never drastic. The authors should merely write "x- and y-fold increased..." or "on average x-fold increased..."
10. Line 405: "The TBR gene is a major regulator..." Replace by: "The TBR gene influences..."
11. L. 446: "...transferase" (not "...s")
12. L. 456/7: The authors should add the information here that different tissues were examined, too.
13. Line 468: "in regulating". Replace by: "... influencing..."
14. Line 405 "The TBR gene is a major regulator in Zn homeostasis and *tbr* mutants were previously shown to display Zn hypersensitivity in roots¹¹." Please replace as follows (TBR is NOT a regulator): "The TBR gene is known to influence Zn homeostasis, and *tbr* mutants were previously shown to display Zn hypersensitivity in roots¹¹."

15. Suppl Fig. 17 Title: Replace "TBR regulates" by "TBR influences"

Reviewer #4:

Remarks to the Author:

I thank the authors for taking into account my comments. I do not have any additional comment on the revised version. Congratulations for this nice piece of work.

REVIEWERS' COMMENTS

Reviewer #2 (Remarks to the Author):

I agree with the acceptance of the manuscript for publication, provided that the following changes are implemented (despite what was stated in the authors' response letter, some changes I had highlighted earlier were not made in the manuscript):

1. Suppl. Fig. 12c/d has now gained substantially more importance, given that the methodologically flawed cell wall Zn data have been removed (the reviewer agrees strongly with the removal). In the methods, the authors say that they washed the roots in 10 mM CaCl₂ for 5 mins and then 10 mM EDTA for 5 mins. Such kind of washes are state-of-the-art for removing cell wall-bound zinc, although they were shorter in this manuscript than is usually done. In any case, at least part of and potentially all of the cell wall-bound Zn were removed here from roots prior to the analysis of Zn concentrations in plant tissues shown in Suppl. Fig. 12c/d, with generally little control over this. In the opinion of the reviewer, the interpretation of the results shown in Suppl. Fig. 12c/d (as reflecting cell wall-bound Zn) in the results and discussion sections are not permissible here. The reviewer suggests that the authors remove all interpretation of these results as reflecting cell wall-bound Zn (lines 323 to 331, line 442/443 “, and ... roots.”) and refer merely to ref. 11 concerning the question of Zn binding ability of the cell walls. In this context it remains peculiar that minor differences in age and growth conditions lead to such dramatic differences in Zn concentrations in roots (between Suppl. Fig. 12c and d). The reviewer suggests that the authors add a sentence on this in the discussion.

We thank the reviewer for these important suggestions and helpful explanations. We have now completely removed this approach, the related images in supplementary figure 12, as well as all interpretation of the results that reflects cell wall bound Zn in roots in the results and the discussion section. As all the results that reflecting cell wall-bound Zn are removed, it would be weird to add a sentence on this in the discussion.

2. Line 54: In addition to reference 1, the reviewer believes that a minimum would be to cite doi: 10.1038/nature06877 for HMA4, and possibly also 10.1534/genetics.106.064485 and 10.1016/j.febslet.2005.06.046 for MTP1, as well as <https://doi.org/10.1016/j.febslet.2007.04.010> for the comparisons of gene expression changes in *Noccaea* and *A. halleri*. Merely citing a rather derived review seems insufficient here.

We thank the reviewer for these important suggestions, we added the references in the introduction part in lines 49-53.

3. The authors use the term “basal tolerance” in lines 61, 75 and 175. This term should be defined at first use (line 61). (“Basal metal tolerance” is the metal tolerance level common to all plants including *A. thaliana*, and the term is used to discriminate this from

naturally selected metal hypertolerance which is found only in specialist plants that are found on soils containing high, toxic levels of metals. This was a modification in reference 11 of the terms coined in 10.1007/s004250000458, further detailed here 10.1016/j.bbamcr.2012.05.016)

*We thank the reviewer for pointing this out and now have specified this:
Lines 59ff: “dez (tbr; trichome birefringence) mutants that show excessively methylesterified pectin display Zn hypersensitivity in the light¹⁵ and TBR has been previously identified to be involved in basal metal tolerance, which the metal tolerance level is thought to be common to all plants including A. thaliana as opposed to metal hypertolerance which is found only in specialist plants that are found on soils containing high, toxic levels of metals^{16, 17}”*

4. L. 176: “decreased root growth”; add after this here: “under high-Zn exposure”. Else it would be inaccurate.

We thank the reviewer for pointing this out. “under high Zn exposure” has been added in line 168 of the text.

5. Line 204: “required for root growth tolerance to high Zn”; please add at the end of this sentence here (by expanding the sentence: “, as shown previously¹¹”. Else this remains too obscure in the context of the way reference 11 is referred to in the earlier part of the manuscript.

We thank the reviewer for pointing this out. We have added “as shown previously¹⁵” in line 196.

6. Line 243/244: fused with both mCitrine and HA – else this is hard to understand.

Thank you for the improvement. We have revised it as “we evaluated the TBR protein levels using confocal microscopy and western blot analysis in the TBR allelic complemented lines, where TBR was fused with mCITRINE and HA” in lines 235-237.

7. Line 245: “transcript” (not: “transcription”), again line 282. Please check also for the same issue elsewhere in the text.

We thank the reviewer for pointing this out. It has been modified in the text accordingly.

8. Line 255-256: when analyzing the full genome sequence at the TBR locus”  rephrase this sentence. It is confusing: either the authors analyzed the full genome sequence or they analyzed the sequence at the TBR locus.

Thank you for the improvement. This sentence has been modified as “When analyzing the sequence at the TBR locus” in lines 248-249.

9. Line 283: “... showed a drastically increased...”. This is not true. Depending on which transgenic lines are compared, the difference is slight to moderate, but never drastic. The authors should merely write “x- and y-fold increased...” or “on average x-fold increased...”

Thank you for the improvement. It has been modified as “When we quantified transcript level via qRT-PCR, we found that the TBR_{vâr2-6}-TBR_{Col-0-16} transgenic line showed 2.5 and 2.9-fold increased transcript level compared to both TBR_{Eds-1}-TBR_{Col-0} transformed lines in both control and high Zn conditions, while the other TBR_{vâr2-6}-TBR_{Col-0} line (TBR_{vâr2-6}-TBR_{Col-0-4}) displayed about 1.5 and 1.7-fold increased transcript level compared to TBR_{Eds-1}-TBR_{Col-0} transformed lines in high Zn conditions (Fig. 5c)” in lines 274-280.

10. Line 405: “The TBR gene is a major regulator...” Replace by: “The TBR gene influences...”

We thank the reviewer for pointing out the issue. It has been modified in the text accordingly (now line 386).

11. L. 446: “...transferase” (not “...s”)

We thank the reviewer for pointing this out. It has been modified in the text accordingly (now line 425).

12. L. 456/7: The authors should add the information here that different tissues were examined, too.

Thank you for the suggestion. We added the information as “It seems most likely that these differences are due to the different growth conditions, seedling ages, or different tissues in the studies” in lines 435-437.

13. Line 468: “in regulating”. Replace by: “... influencing...”

We thank the reviewer for pointing this out. It has been modified in the text accordingly (now line 447).

14. Line 405 “The TBR gene is a major regulator in Zn homeostasis and tbr mutants were previously shown to display Zn hypersensitivity in roots11.”: Please replace as follows (TBR is NOT a regulator): “The TBR gene is known to influence Zn homeostasis, and tbr mutants were previously shown to display Zn hypersensitivity in roots11.”

We thank the reviewer for pointing this out. As this comment is the same as No. 10, we have replaced it as “The TBR gene influences Zn homeostasis, and tbr mutants were previously shown to display Zn hypersensitivity in roots¹⁵” in lines 386-387 of the text.

15. Suppl Fig. 17 Title: Replace “TBR regulates” by “TBR influences”

We thank the reviewer for pointing this out. The title has been modified as “TBR influences hypocotyl growth in high Zn” of Supplementary figure17.